# A Staphylococcus pro-apoptotic peptide induces acute exacerbation of pulmonary fibrosis

Corina N. D'Alessandro-Gabazza[1,2,3,18], Tetsu Kobayashi[4,18], Taro Yasuma[1,2,5,18], Masaaki Toda[1,2,18], Heejin Kim[3,18], Hajime Fujimoto[4], Osamu Hataji[6], Atsuro Takeshita[1,5], Kota Nishihama[5], Tomohito Okano[4], Yuko Okano[1,5], Yoichi Nishii[6], Atsushi Tomaru[4], Kentaro Fujiwara[4,6], Valeria Fridman D'Alessandro[1], Ahmed M. Abdel-Hamid[3], Yudong Ren[3,7], Gabriel V. Pereira [3,8], Christy L. Wright[9], Alvaro Hernandez[9], Christopher J. Fields [9], Peter M. Yau[9], Shujie Wang[10], Akira Mizoguchi[10], Masayuki Fukumura[11,12], Junpei Ohtsuka[11,12], Tetsuya Nosaka [12], Kensuke Kataoka[13], Yasuhiro Kondoh[13], Jing Wu[14], Hirokazu Kawagishi [14,15], Yutaka Yano[5], Roderick I. Mackie[3,8,16], Isaac Cann[3,7,8,16,17 ✉] & Esteban C. Gabazza[1,2,3 ✉]

Idiopathic pulmonary fibrosis (IPF) is a chronic and fatal disease of unknown etiology; however, apoptosis of lung alveolar epithelial cells plays a role in disease progression. This intractable disease is associated with increased abundance of *Staphylococcus* and *Streptococcus* in the lungs, yet their roles in disease pathogenesis remain elusive. Here, we report that *Staphylococcus nepalensis* releases corisin, a peptide conserved in diverse staphylococci, to induce apoptosis of lung epithelial cells. The disease in mice exhibits acute exacerbation after intrapulmonary instillation of corisin or after lung infection with corisin-harboring *S. nepalensis* compared to untreated mice or mice infected with bacteria lacking corisin. Correspondingly, the lung corisin levels are significantly increased in human IPF patients with acute exacerbation compared to patients without disease exacerbation. Our results suggest that bacteria shedding corisin are involved in acute exacerbation of IPF, yielding insights to the molecular basis for the elevation of staphylococci in pulmonary fibrosis.

---

[1] Department of Immunology, Mie University Faculty and Graduate School of Medicine, Edobashi 2-174, Tsu, Mie 514-8507, Japan. [2] Center for Intractable Diseases, Mie University, Edobashi 2-174, Tsu, Mie 514-8507, Japan. [3] Carl R. Woese Institute for Genomic Biology (Microbiome Metabolic Engineering), University of Illinois at Urbana–Champaign, Urbana, IL, USA. [4] Department of Pulmonary and Critical Care Medicine, Mie University Faculty and Graduate School of Medicine, Edobashi 2-174, Tsu, Mie 514-8507, Japan. [5] Department of Diabetes and Endocrinology, Mie University Faculty and Graduate School of Medicine, Edobashi 2-174, Tsu, Mie 514-8507, Japan. [6] Respiratory Center, Matsusaka Municipal Hospital, Tonomachi 1550, Matsusaka, Mie 515-8544, Japan. [7] The School of Molecular and Cellular Biology, University of Illinois at Urbana–Champaign, Urbana, IL, USA. [8] Department of Animal Science, University of Illinois at Urbana–Champaign, Urbana, IL, USA. [9] W.M. Keck Center for Functional and Comparative Genomics, University of Illinois at Urbana–Champaign, Urbana, IL, USA. [10] Department of Neural Regeneration and Cell Communication, Mie University Graduate School of Medicine, Tsu, Mie 14101, Japan. [11] BioComo Incorporation, Komono, Mie 510-1233, Japan. [12] Department of Microbiology and Molecular Genetics, Mie University Graduate School of Medicine, Tsu, Mie 514-8507, Japan. [13] Department of Respiratory Medicine and Allergy, Tosei General Hospital, 160 Nishioiwake-cho, Seto, Aichi 489-8642, Japan. [14] Research Institute of Green Science and Technology, Graduate School of Agriculture, Shizuoka University, 836 Ohya, Shizuoka 422-8529, Japan. [15] Green Chemistry Research Division, Research Institute of Green Science and Technology, Shizuoka University, 836 Ohya, Suruga-ku, Shizuoka 422-8529, Japan. [16] Division of Nutritional Sciences, University of Illinois at Urbana–Champaign, Urbana, IL, USA. [17] Department of Microbiology, University of Illinois at Urbana–Champaign, Urbana, IL, USA. [18] These authors contributed equally: Corina N. D'Alessandro-Gabazza, Tetsu Kobayashi, Taro Yasuma, Masaaki Toda, Heejin Kim. ✉email: icann@illinois.edu; gabazza@doc.medic.mie-u.ac.jp

diopathic pulmonary fibrosis (IPF) is the most frequent form of idiopathic interstitial pneumonitis characterized by a chronic, progressive, and fatal clinical outcome[1,2]. The prognosis of IPF is worse than many types of malignancy, with a life expectancy for patients following diagnosis of the disease being only 2–3 years[3,4]. Repetitive injury and/or apoptosis of lung epithelial cells, excessive release of profibrotic factors and enhanced lung recruitment of extracellular matrix-producing myofibroblasts play critical roles in the disease pathogenesis[2,5]. Recent evidence suggests that the lung microbiome plays a causative role in IPF, with increased lung bacterial burden being associated with acute exacerbation of the disease and high mortality rate[6]. The relative abundance of lung microbes of the *Staphylococcus* and *Streptococcus* genera has also been associated with acceleration of the clinical progression of IPF[7]. However, the role of these bacteria in the pathogenesis of pulmonary fibrosis remains unclear. The capacity to culture the bacteria associated with fibrotic tissues and elucidation of their phenotypic characteristics would be ideal in clearly identifying the organisms involved in the pathogenesis of IPF; however, to date there is no report of bacterial isolates that are relevant to disease pathogenesis.

In a previous report, we demonstrated that the lung fibrotic tissue from IPF patients and from transforming growth factor (TGF)β1 transgenic (TG) mice with lung fibrosis is characterized by an enrichment of halophilic bacteria, and a subsequent report has substantiated our observation[8,9]. These results led us to hypothesize that the fibrotic tissue is a salty microenvironment, and that the hypersaline condition of the lung fibrotic tissue facilitates the growth of bacteria that release factors that play a role in disease pathogenesis and its acute exacerbation.

In this study, we use a halophilic medium to enrich for *Staphylococcus* strains from lung fibrotic tissue samples originating from TGFβ1 TG mice. We then show that the culture supernatants of one of the bacterial strains, *S. nepalensis* strain CNDG, contain a pro-apoptotic peptide that induces apoptosis of lung epithelial cells. We further reveal that the peptide, designated corisin, is a component of a transglycosylase conserved in diverse members of the genus *Staphylococcus*, and that intratracheal instillation of mice with established lung fibrosis either with corisin or the corisin-encoding *S. nepalensis* strain CNDG leads to acute exacerbation of the disease. Finally, the enhanced detection of corisin in human IPF patients with acute exacerbation compared to patients without disease exacerbation suggests that bacteria carrying and shedding the pro-apoptotic peptide are involved in acute exacerbation of pulmonary fibrosis and therefore provide molecular insights underlying the association of the staphylococci with the worsening stage of pulmonary fibrosis.

## Results

**The fibrotic lung tissue is a salty microenvironment**. TGFβ1 is considered the most important mediator of IPF. Therefore, here we used TG mice with lung fibrosis induced by lung overexpression of human TGFβ1, as previously reported[8,10–12]. Similar to the disease in humans, these TGFβ1 TG mice spontaneously develop pulmonary fibrosis characterized by a predominant and progressive scarring process, fatal outcome and typical lung histopathological findings (diffuse collagen deposition, honeycomb cysts, and fibroblast foci-like areas)[8,11]. We used a line of TGFβ1 TG mice without fibrosis that express the human transgene but not the protein as controls[8,13]. To interrogate the hypothesis that the lung fibrotic tissue is a salty microenvironment, we measured the Na$^+$ content of lung fibrotic tissues from TGFβ1 TG mice with lung fibrosis[8], by allocating TGFβ1 TG and wild-type (WT) mice in groups by computed tomography (CT)-

based fibrosis score (Supplementary Fig. 1a, b). There was significantly higher concentration of Na$^+$ in lung tissue from TGFβ1 TG mice with lung fibrosis compared to TG mice without lung fibrosis and WT mice (Fig. 1a–c). These observations demonstrate that the lung fibrotic tissue is a salty microenvironment.

**Abnormal immune response in lung fibrotic tissue**. We separated lung immune cells from each WT mice without fibrosis, TGFβ1 TG mice without lung fibrosis and TGFβ1 TG mice with fibrosis and compared the percentage of cells between groups. There was a significant increase in the percentage of monocyte/macrophages and regulatory (CD4$^+$CD25$^+$) T cells in TGFβ1 TG mice with lung fibrosis compared to WT and TGFβ1 TG mice without lung fibrosis (Supplementary Fig. 2a, b; Supplementary Table 1). The percentage of total T cells was not different between groups, but the percentage of B cells was significantly decreased in TGFβ1 TG mice with lung fibrosis compared to WT and TGFβ1 TG mice without lung fibrosis (Supplementary Fig. 2c, d). These observations suggest impaired immune response in lung fibrotic tissue.

**Sodium, immune cells, fibrotic markers, and sodium channels**. As expected, the lung tissue relative mRNA expression of fibrotic markers (connective tissue growth factor, fibronectin 1, collagen I) and of pro-fibrotic cytokines (TGFβ1, tumor necrosis factor-α, interferon-γ), chemokines (monocyte chemoattractant protein-1), vascular endothelial growth factor or inducible nitric oxide synthase were significantly increased in TGFβ1 TG mice with lung fibrosis compared to WT and TGFβ1 TG mice without fibrosis (Supplementary Table 2). However, the lung tissue relative mRNA expression of the chloride (cystic fibrosis transmembrane conductance regulator) and sodium (Scnnγ, Scnnβ) channels were significantly decreased in TGFβ1 TG mice with lung fibrosis compared to WT and TGFβ1 TG mice without lung fibrosis (Supplementary Table 2). We evaluated correlation between variables in all WT and TGFβ1 TG mice with and without fibrosis. The tissue level of sodium was inversely and significantly correlated with the mRNA expression of chloride and sodium channels and with the number of B cells. In contrast, the tissue sodium level was proportionally and significantly correlated with fibrotic markers, pro-fibrotic cytokines and with the number of monocytes/macrophages and regulatory T cells (Supplementary Fig. 3). These findings support the detrimental role of a salty microenvironment in the process of tissue fibrosis and the implication of the tissue sodium level in the regulation of the immune response[14].

**Growth of bacteria from fibrotic lung tissue**. After confirming that the fibrotic tissue is a salty microenvironment, we posited that a hypersaline culture medium would best mimic the in vivo fibrotic tissue condition, and thus it would favor the growth of microbes implicated in disease pathogenesis. We incubated lung fibrotic tissue specimens from TGFβ1 TG and WT mice (Fig. 2a, b) for 48 h in a medium containing 8% NaCl. Bacterial growth in medium inoculated with lung fibrotic specimens from TGFβ1 TG mice, but not from WT mice, was detected. We then performed streak plating to isolate bacterial colonies, and by using phase-contrast microscopy, a bacteria morphology compatible with *Staphylococcus* spp. was observed (Fig. 2c). The identities of the bacterial strains were confirmed by sequencing of their 16S rRNA genes, amplified by polymerase chain reaction. Determining the whole genome sequences, however, revealed that while one of the colonies (strain 8) corresponds to a strain of *Staphylococcus nepalensis*, another colony (strain 6) was a mixture of *Staphylococcus* spp. The whole genome sequences of the cultures designated

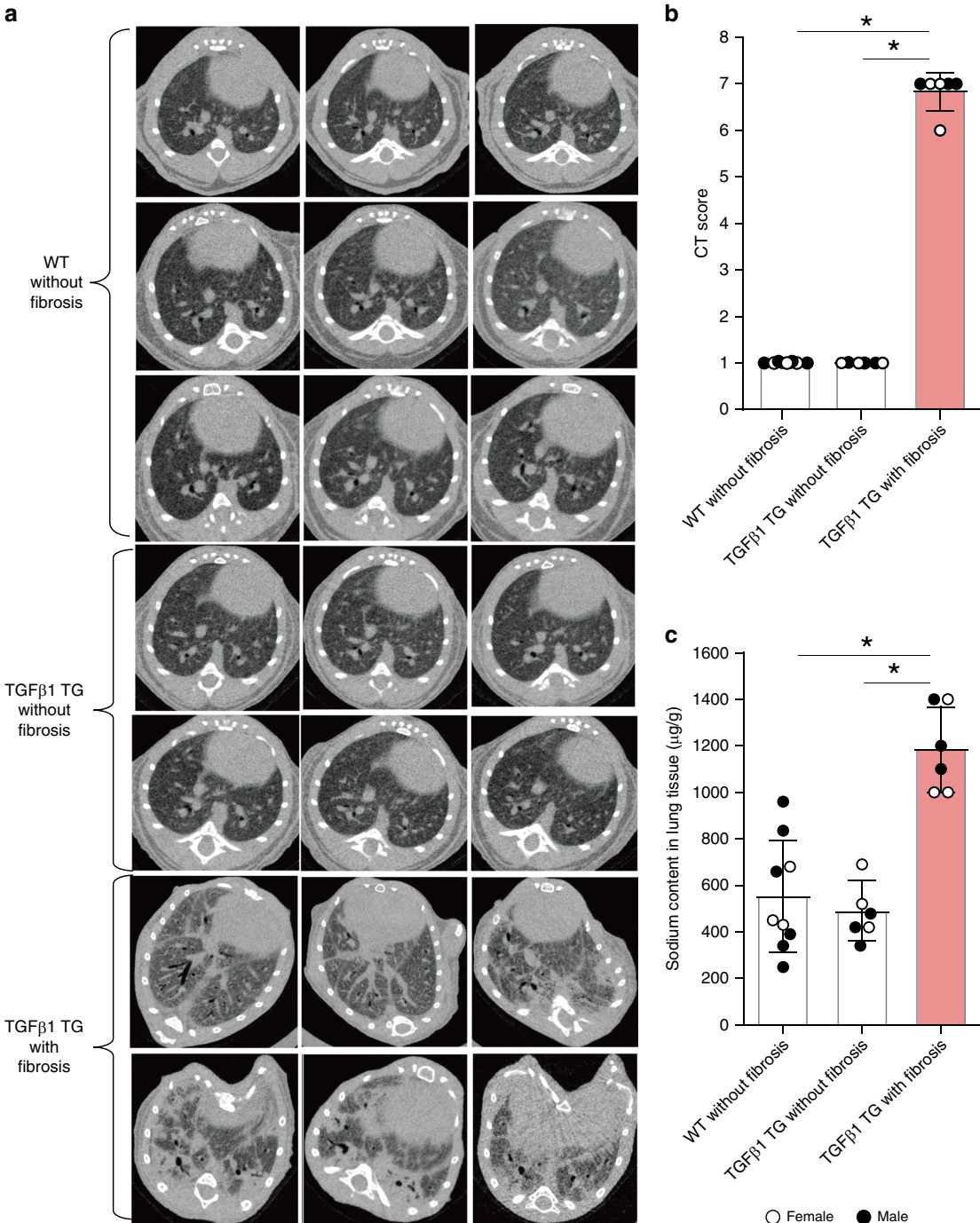

**Fig. 1 The fibrotic lung tissue is a salty microenvironment. a**, **b** Chest computed tomography (CT) findings and CT scores of wild-type and TGFβ1 transgenic (TG) mice. $n = 9$ in WT without fibrosis, $n = 6$ in TGFβ1 TG without fibrosis and TGFβ1 TG with fibrosis. Bars indicate the means ± S.D. Statistical analysis by ANOVA with Tukey's test. *$p < 0.001$. **c** The lung tissue $Na^+$ was measured by microwave analysis/inductively coupled plasma mass spectrometry. Fibrotic lung tissues excised from wild-type and transforming growth factor (TGF)β1 TG mice with and without ($n = 6$) lung fibrosis. $n = 9$ in WT without fibrosis, $n = 6$ in TGFβ1 TG without fibrosis and TGFβ1 TG with fibrosis. Bars indicate the means ± S.D. Statistical analysis by Mann–Whitney $U$ test. Statistical analysis by ANOVA with Tukey's test. *$p < 0.001$. The source data underlying Fig. 1b, c are provided in the Source Data file.

strain 6 and strain 8 have been deposited at the Genbank database with the accession number PRJNA544423. To further confirm the identity of strain 8, we compared its whole genome sequence with that of other *Staphylococcus nepalensis* strains in the Genbank database, and for strains JS9, SNUC4337, DSM15150, JS11, and JS1 the identities were 99.52%, 99.61%, 99.60%, 99.53%, and 99.50%, respectively. Thus, based on the purity of strain 8 and its very high genomic homology to other *Staphylococcus nepalensis* strains, we

named the bacterium *Staphylococcus nepalensis* with a strain designation of CNDG.

**Apoptosis of lung cells induced by culture supernatants.** To assess the potential implication of these fibrotic tissue-derived bacterial isolates in disease pathogenesis, we cultured normal human bronchial epithelial (NHBE) cells and A549 alveolar

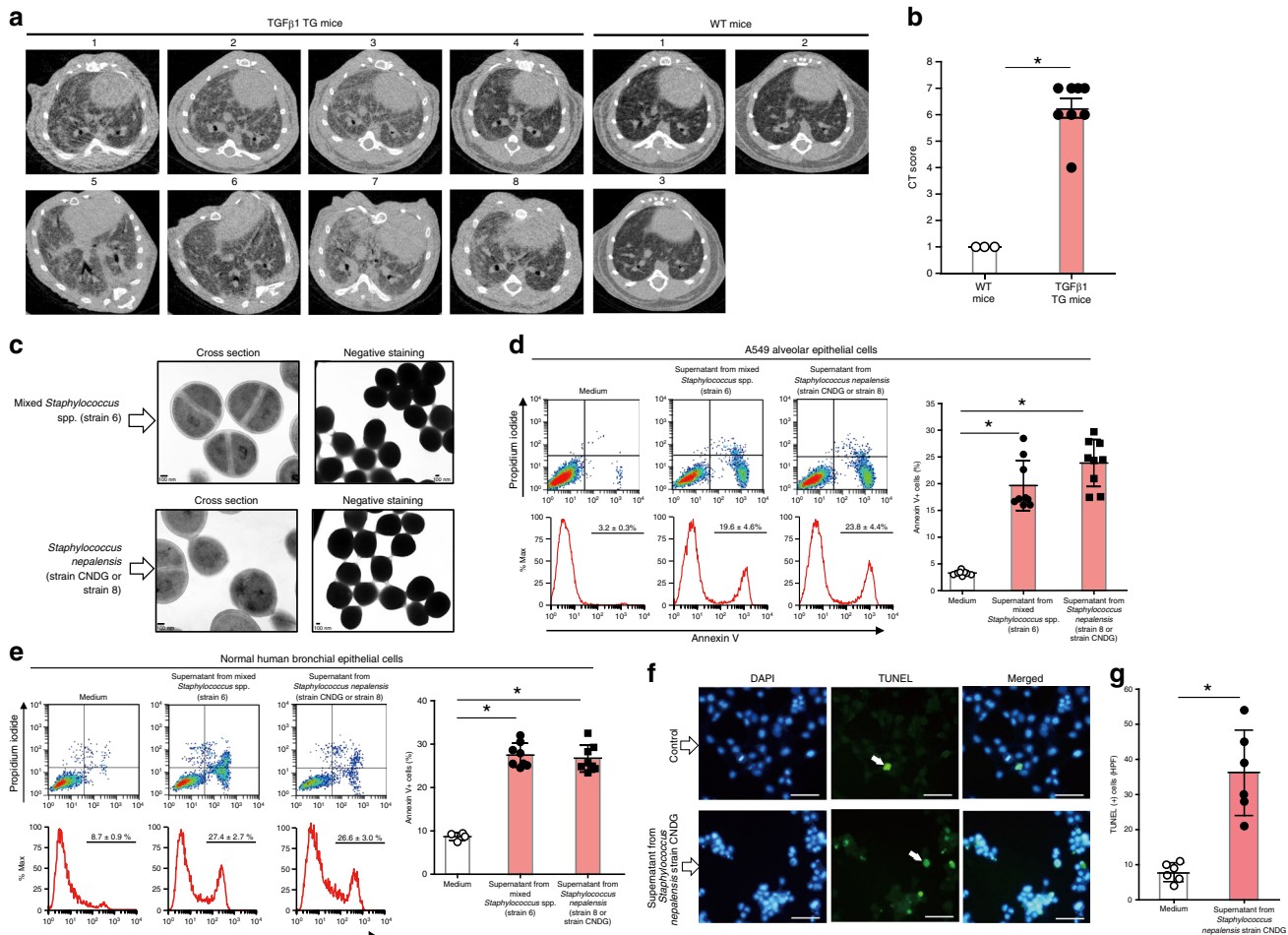

**Fig. 2 Growth of bacteria from fibrotic lung tissue cultured under hypersaline conditions and apoptosis induced by their culture supernatants.**
**a**, **b** Computed tomography (CT) and CT fibrosis scoring of wild-type (WT) mice ($n = 3$) and TGFβ1 transgenic (TG) mice ($n = 8$). Bars indicate the means ± standard error of the means. Statistical analysis by two-tailed Wilcoxon signed rank test. *$p = 0.01$. **c** Fibrotic lung tissues excised under sterile conditions from wild-type ($n = 3$) and TGFβ1 transgenic ($n = 8$) mice were cultured in hypersaline culture media for 48 h. Analysis of bacterial colonies by transmission electron microscope. Scale bars indicate 100 nm. **d** Flow cytometry analysis of A549 alveolar epithelial cells cultured for 48 h in DMEM medium containing 1/10 diluted spent culture supernatant of the mixture of *Staphylococcus* spp. (strain 6; $n = 9$), *Staphylococcus nepalensis* strain CNDG ($n = 9$), or control medium ($n = 9$). Bars indicate the means ± S.D. Statistical analysis by two-tailed Mann–Whitney $U$ test. *$p < 0.0001$. **e** Flow cytometry analysis of normal human bronchial epithelial cells cultured for 48 h in DMEM medium containing 1/10 diluted spent culture supernatant of the mixture of *Staphylococcus* spp. (strain 6; $n = 8$), *Staphylococcus nepalensis* strain CNDG ($n = 8$), or control medium ($n = 4$). Bars indicate the means ± S.D. Statistical analysis by two-tailed Mann–Whitney $U$ test. *$p < 0.005$. **f**, **g** TUNEL assay was performed as described under methods after culturing A549 alveolar epithelial cells in the presence of medium ($n = 6$) or supernatant of *Staphylococcus nepalensis* strain CNDG ($n = 6$). Scale bars indicate 20 μm. Bars indicate the means ± S.D. Statistical analysis by two-tailed Mann–Whitney $U$ test. *$p < 0.01$. Abbreviations are defined as HPF, high-power field; TUNEL, terminal deoxynucleotidyl transferase dUTP nick end labeling; DAPI, 4′,6-diamidino-2-phenylindole. The source data underlying 2**b**, **d**, **e**, **g** are provided in the Source Data file.

epithelial cells in the presence of the bacterial culture supernatant and evaluated cell survival. Cells cultured in the presence of supernatants from *Staphylococcus nepalensis* CNDG and the mixed bacteria showed significant levels of apoptosis, caspase-3 activation and DNA fragmentation compared to cells cultured in control medium (Fig. 2d–g).

**Culture supernatant with the highest apoptotic activity.** The culture supernatants from the mixed *Staphylococcus* spp. (strain 6; Fig. 3a–d) and *Staphylococcus nepalensis* CNDG (strain 8; Fig. 3e–h) were separated into several fractions using a Sephadex column, and the peak of the protein concentrations matched well with the nadir of cell viability of the MTT assay and with the sub-G1 fraction peak of the cell cycle analysis.

**Apoptosis depends on the bacterial medium salt concentration.** We cultured *Staphylococcus nepalensis* CNDG and the mixed *Staphylococcus* spp. in media containing 0%, 2% or 8%NaCl and used the culture supernatant to assess apoptosis by flow cytometry. The apoptotic activity was significantly dependent on the salt concentration of the medium used to culture both isolates in vitro (Fig. 3i, j).

**The apoptotic factor is a heat-stable peptide.** The culture supernatant from bacteria was incubated at 85 °C for 15 min before assessing its pro-apoptotic activity on A549 alveolar epithelial cells. The apoptotic activity of the culture supernatant from both *Staphylococcus nepalensis* CNDG and the mixed *Staphylococcus* spp. remained stable after heating, and the activities were significantly stronger than unheated culture supernatant

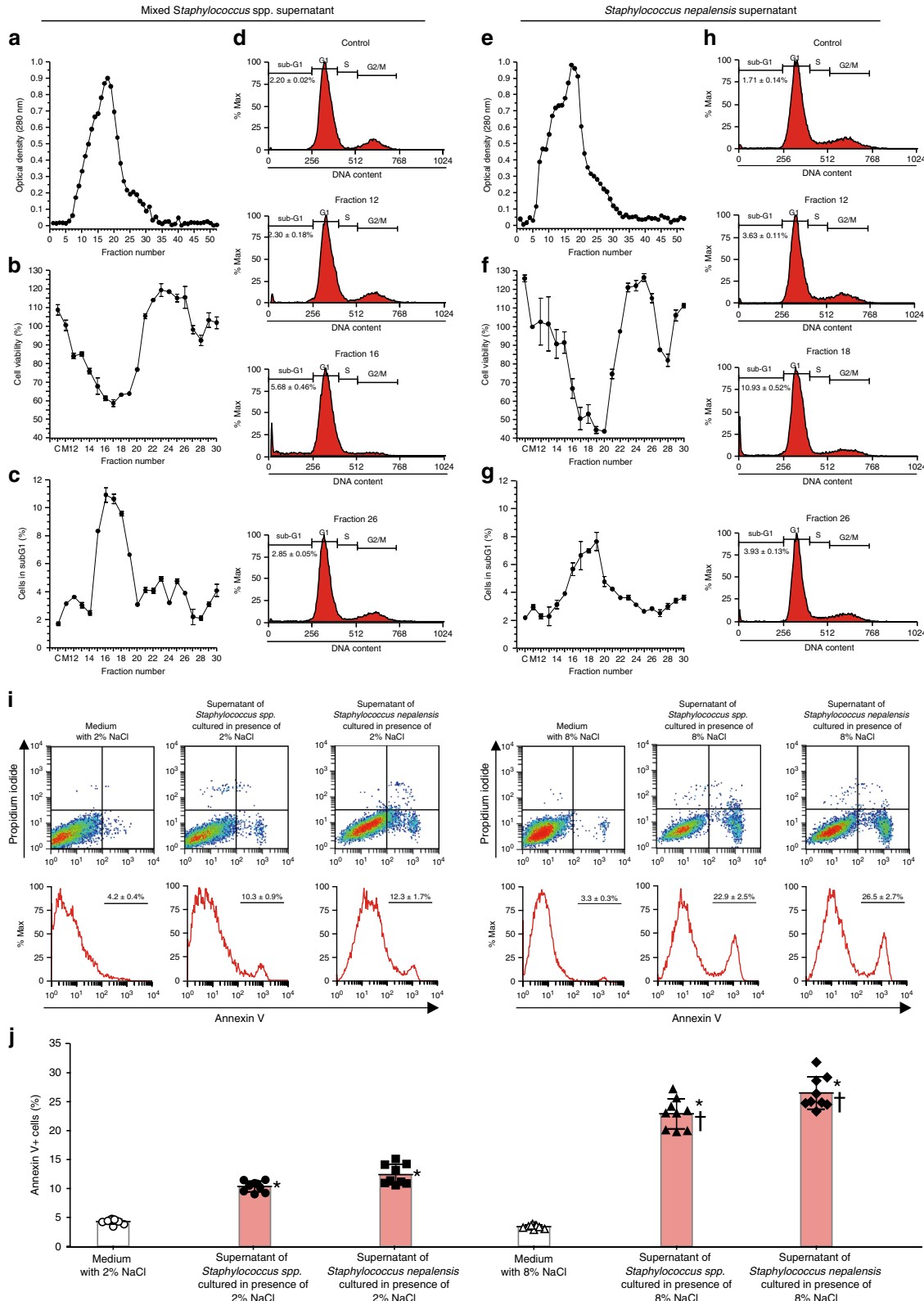

(Supplementary Fig. 4a–d). To gain insight into the identity of the pro-apoptotic factor, we fractionated the proteins of the bacterial supernatants into low (<10 kDa) and high (>10 kDa) molecular weight proteins, repeated the experiments, and found that the fraction with low-molecular-weight proteins has a potent and significant apoptotic activity compared to the fraction with high-molecular-weight proteins (Fig. 4a, b). These observations suggested that the apoptosis-inducing factor is a protein of low molecular weight, and that this soluble factor released by the bacteria enriched from the fibrotic tissue contributes to the mechanism of lung fibrosis by sealing the fate of lung epithelial cells.

**Fig. 3 Fractions of culture supernatant with high protein concentration and spent medium with high salt concentration induce potent apoptotic activity. a** Absorbance of fractions from the culture supernatant of the mixture of *Staphylococcus* spp. after gel filtration using Sephadex G25 column. **b** Cell viability after treating A549 alveolar epithelial cells with the culture supernatant of the mixture of *Staphylococcus* spp. (each fraction $n = 3$). **c** Cells in sub-G1 phase after treating A549 cells with culture supernatant of the mixture of *Staphylococcus* spp. (each fraction $n = 3$). **d** Representative histograms of A549 cells in sub-G1 phase after treatment with culture supernatant of the mixture of *Staphylococcus* spp. **e** Absorbance of fractions from the culture supernatant of *Staphylococcus nepalensis* strain CNDG after gel filtration. **f** Cell viability after treating A549 cells with culture supernatant of *Staphylococcus nepalensis* strain CNDG (each fraction $n = 3$). **g** Cells in sub-G1 phase after treating A549 cells with culture supernatant of *Staphylococcus nepalensis* strain CNDG (each fraction $n = 3$). **h** Representative histograms of A549 cells in sub-G1 phase after treatment with culture supernatant of *Staphylococcus nepalensis* strain CNDG. One mL of each sample was applied into the Sephadex G25 column. The material eluted was collected in 2 ml fractions and then absorbance was measured at 280 nm. Cell viability was evaluated by using a commercial cell counting kit and the percentage of cells in sub-G1 by flow cytometry. **i, j** Bacteria were cultured in medium containing 2% or 8% salt and the culture supernatant of the mixture of *Staphylococcus* spp. ($n = 9$), *Staphylococcus nepalensis* CNDG strain ($n = 9$) or medium ($n = 9$) was prepared by centrifugation and added to the culture medium of A549 alveolar epithelial cells at 1/10 dilution. Flow cytometry of A549 cells was performed after staining with propidium iodide and annexin V. Bars indicate the means ± S.D. Statistical analysis by ANOVA with Tukey's test. *$p < 0.001$ vs medium control; †$p < 0.0001$ vs supernatant of bacteria cultured in medium with 2% NaCl. The source data underlying **a–c**, **e–g**, and **i** are provided in the Source Data file.

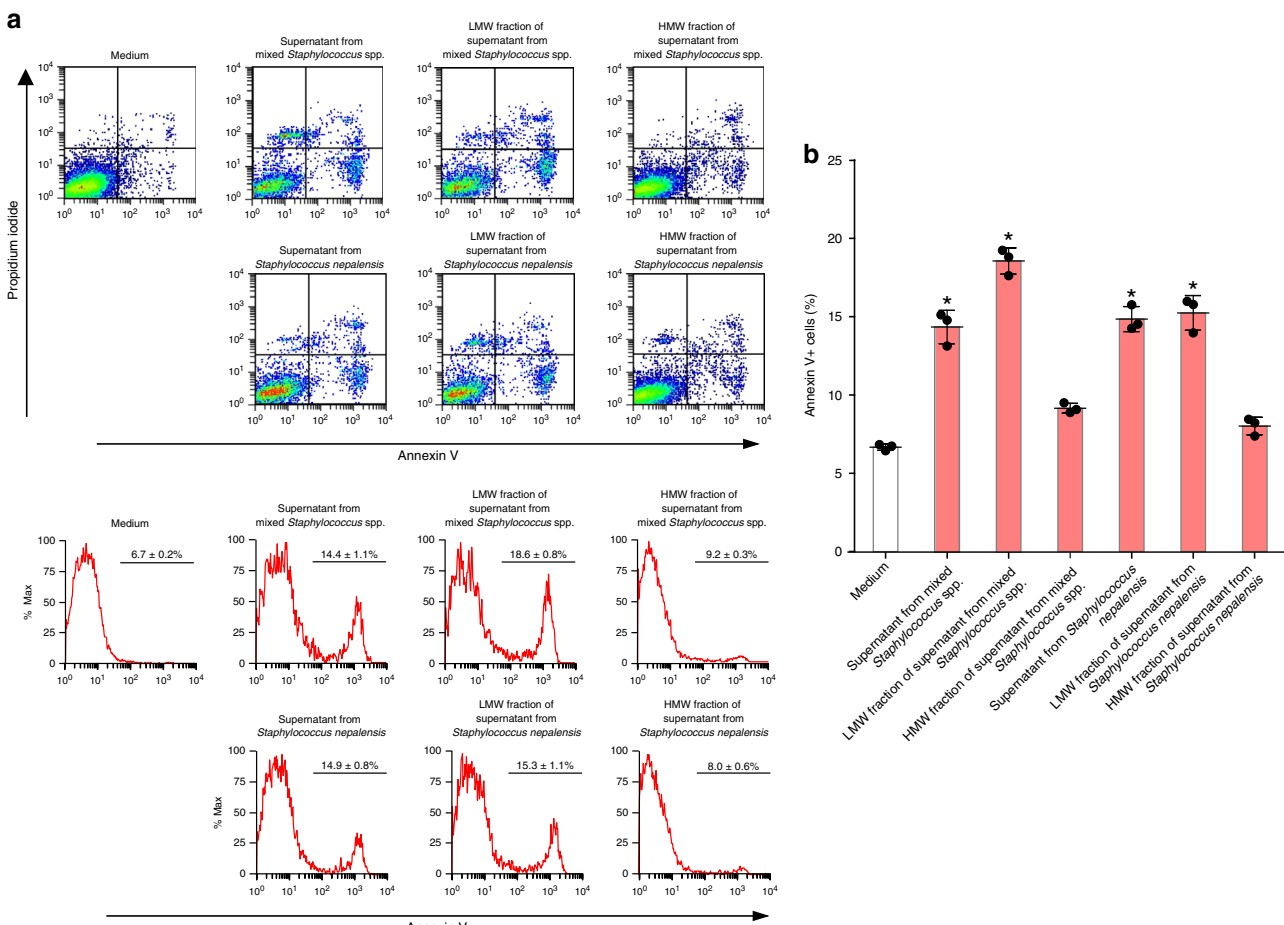

**Fig. 4 The apoptotic factor in culture supernatant from bacteria is a low molecular weight protein. a, b** Culture supernatant from bacteria was separated into fractions of <10 kDa and >10 kDa by filtration and each fraction was added to A549 alveolar epithelial cells after 1/10 dilution to determine apoptosis by flow cytometry. For each group, $n = 3$. Bars indicate the means ± S.D. Statistical analysis by ANOVA and Tukey's test. *$p < 0.001$ vs medium control. LMW low molecular weight, HMW high molecular weight. The source data underlying **b** are provided in the Source Data file.

**Identification of the pro-apoptotic peptide.** We next proceeded to purify the soluble pro-apoptotic factor from the culture supernatant of *Staphylococcus nepalensis* strain CNDG. Successive extractions of the proteins in the supernatant were performed in *n*-hexane, water, ethyl acetate, ethanol and then fractionations using octadecyl-silane gel flash column chromatography and Sep-Pak followed by high-performance liquid chromatography (HPLC) (Supplementary Fig. 5; see HPLC data in Supplementary Information) to separate the biologically active protein (Supplementary

Figs. 6 and 7). The biological activity decreased significantly after treatment of the samples with proteinase K (Supplementary Fig. 8a, b). Silver staining, after gel electrophoresis of the sample, revealed a protein/peptide with an apparent molecular weight of 2 kDa (Supplementary Fig. 9). Subsequently, we analyzed the peptide by mass spectrometry and compared the raw data against a custom database of *Staphylococcus nepalensis* strain CNDG protein sequences, based on its closed genome sequence data (Genbank Accession number PRJNA544423). The mass

spectrometry analysis (see mass spectrometry data in Supplementary Information) identified a peptide of 19 amino acid residues (IVMPESSGNPNAVNPAGYR) that corresponded to a molecular mass of 1.94 kDa, in agreement with the purified biological activity in the culture supernatant. We named this newly discovered peptide "corisin". Homology search revealed that the corisin sequence corresponds to a segment of transglycosylase 351 IsaA (MW: 25.6 kDa) of *Staphylococcus nepalensis* strain CNDG.

**Structure prediction and apoptotic activity of corisin**. Structural alignment using a homology modeling server (https://swissmodel.expasy.org/) showed that corisin shares 46.88% identity with a segment of an endo-type membrane-bound lytic murein transglycosylase A (Fig. 5a–c). Two different commercial manufacturers prepared synthetic corisin (with the deduced amino acid sequence, Peptide Institute, Osaka, Japan and ThermoFisher Scientific, Waltham, MA, USA), and we used each peptide to treat A549 alveolar epithelial cells. Both synthetic peptides recapitulated the pro-apoptotic effect of the staphylococcal isolate supernatant in a dose dependent manner (Fig. 5d, e, Supplementary Fig. 10a, b) in A549 lung epithelial cells. The apoptotic activity of synthetic corisin was significantly more potent than equal protein concentrations of supernatant from *Staphylococcus nepalensis* strain CNDG and from the mixed *Staphylococcus* spp. (strain 6) (Supplementary Fig. 10c–e). Normal human bronchial epithelial cells also showed significantly enhanced apoptosis in the presence of corisin, but not in the presence of a synthetic peptide composed of its scrambled amino acid sequence (Supplementary Fig. 11a, b), in association with increased cleavage of caspase-3 and decreased Akt activation (Supplementary Fig. 11c–e). In additional experiments using A549 alveolar epithelial cells, the pro-apoptotic activity of synthetic corisin was heat-resistant (Supplementary Fig. 12a, b), as observed in the culture supernatant, and examination by transmission electron micrographs confirmed the apoptotic property of corisin (Fig. 5f). However, corisin showed no apoptotic activity on lung fibroblast, vascular endothelial cell or lymphocyte cell lines (Supplementary Fig. 13a–f).

**Anti-corisin antibody inhibits corisin-induced apoptosis**. We developed polyclonal antibody against corisin. The antibody could detect corisin in mouse lung tissue and in culture supernatant of *Staphylococcus nepalensis* (Supplementary Fig. 14a, b). We then stimulated A549 alveolar epithelial cells with corisin or with culture supernatant from *Staphylococcus nepalensis* strain CNDG in the presence of saline, control rabbit IgG or rabbit anti-corisin IgG and assessed apoptotic cells by flow cytometry. We found significant inhibition of lung epithelial cell apoptosis induced by synthetic corisin (Supplementary Fig. 15a, b) and by the culture supernatant of *Staphylococcus nepalensis* (Supplementary Fig. 15c, d) in the presence of polyclonal anti-corisin antibody compared to control IgG.

**The full-length transglycosylase has no apoptotic activity**. We prepared 6-Histidine-tagged (His-tagged) or Tag-free (the His-tag cleaved) recombinant full-length transglycosylase 351, expressed in *E. coli* cells, to evaluate apoptotic activity on A549 cells. The unheated or heated recombinant His-tagged transglycosylase 351 (Supplementary Fig. 16a, b) and the Tag-free recombinant transglycosylase 351 (Supplementary Fig. 16c–e) failed to induce apoptosis in lung epithelial cells, suggesting the need for polypeptide processing and corisin release for biological activity.

**Corisin exacerbates pulmonary fibrosis in hTGFβ1 TG mice**. To investigate whether corisin can exacerbate the lung fibrotic disease in vivo, we separated TGFβ1 TG mice into three groups with matched level of lung fibrosis (Supplementary Fig. 17a, b) and treated them with saline, scrambled peptide or corisin by intra-tracheal route once daily for two days before euthanasia on day 3 (Fig. 6a). TGFβ1 TG mice receiving corisin showed significantly increased infiltration of macrophages, lymphocytes and neutrophils, increased collagen deposition and concentration of inflammatory cytokines and chemokines, and enhanced apoptosis of epithelial cells in the lungs compared to control mice (Fig. 6b–g), indicating the detrimental effect of the pro-apoptotic activity of corisin in vivo.

***S. nepalensis* instillation exacerbates pulmonary fibrosis**. We evaluated in vivo whether bacteria that express transglycosylases containing the corisin sequence also exacerbate lung fibrosis. To this end, we intratracheally administered *Staphylococcus nepalensis* strain CNDG, which contains the corisin sequence, or *Staphylococcus epidermidis* [ATCC14990], as negative control, to germ-free TGFβ1 TG mice separated in three groups with matched lung fibrosis CT scores (Supplementary Fig. 18a, b). Before this in vivo experiment, we corroborated in vitro that a synthetic peptide (IIARESNGQLHARNASGAA) corresponding to the peptide sequence at the "corisin position" of the transglycosylase from *Staphylococcus epidermidis* exerts no pro-apoptotic effect on lung epithelial cells (Supplementary Fig. 19a, b). TGFβ1 TG mice instilled with *Staphylococcus nepalensis* strain CNDG showed significant worsening of lung radiological findings (Supplementary Fig. 20a, b), and significantly increased neutrophil infiltration, and enhanced alveolar epithelial cell apoptosis compared to mice receiving *Staphylococcus epidermidis* (Fig. 7), further corroborating the role of the pro-apoptotic peptide in acute exacerbation of pulmonary fibrosis.

**Detection of corisin in the lungs of mice and human patients**. We explored the presence of corisin in WT mice without fibrosis, TGFβ1 TG mice with and without fibrosis. We found significantly enhanced level of corisin in TGFβ1 TG mice with lung fibrosis compared to WT mice and TGFβ1 TG mice without fibrosis (Fig. 8a, b). To clarify the clinical relevance of this finding, we also evaluated corisin in human IPF patients. We collected bronchoalveolar lavage fluids from 34 IPF patients and 8 male healthy controls. The characteristics of the IPF patients are described in Supplementary Table 3. The level of corisin in bronchoalveolar lavage fluid was significantly increased in IPF patients with stable disease or with acute exacerbation compared to healthy controls (Fig. 8c, d). The BALF corisin level was also significantly elevated in IPF patients with acute exacerbation compared to patients with stable disease (Fig. 8c, d). The difference in the level of corisin was not statistically significant ($p = 0.07$) between males ($50.6 \pm 4.9$ pg/ml) and females ($58.8 \pm 10.7$ pg/ml). The corisin level was also not significantly correlated ($r = 0.1$, $p = 0.5$) with the age of the patients. These results suggest the clinical relevance of corisin in IPF. A dramatic increase of apoptotic epithelial cells occurs in the lung of IPF patients with acute exacerbation[15,16], and our results suggest that excessive release of the bacterial-derived pro-apoptotic corisin will contribute to this fatal disease complication.

**Phylogenetic analysis reveals conservation of corisin**. To unveil the evolutionary relationship of transglycosylases expressed by different bacteria, we constructed a phylogenetic tree based on the amino acid sequences of six transglycosylases identified in the genome of *Staphylococcus nepalensis* strain CNDG and their homologs in the publicly available database (https://www.ncbi.nlm.nih.gov/pubmed/). The topology of the phylogenetic tree shows that

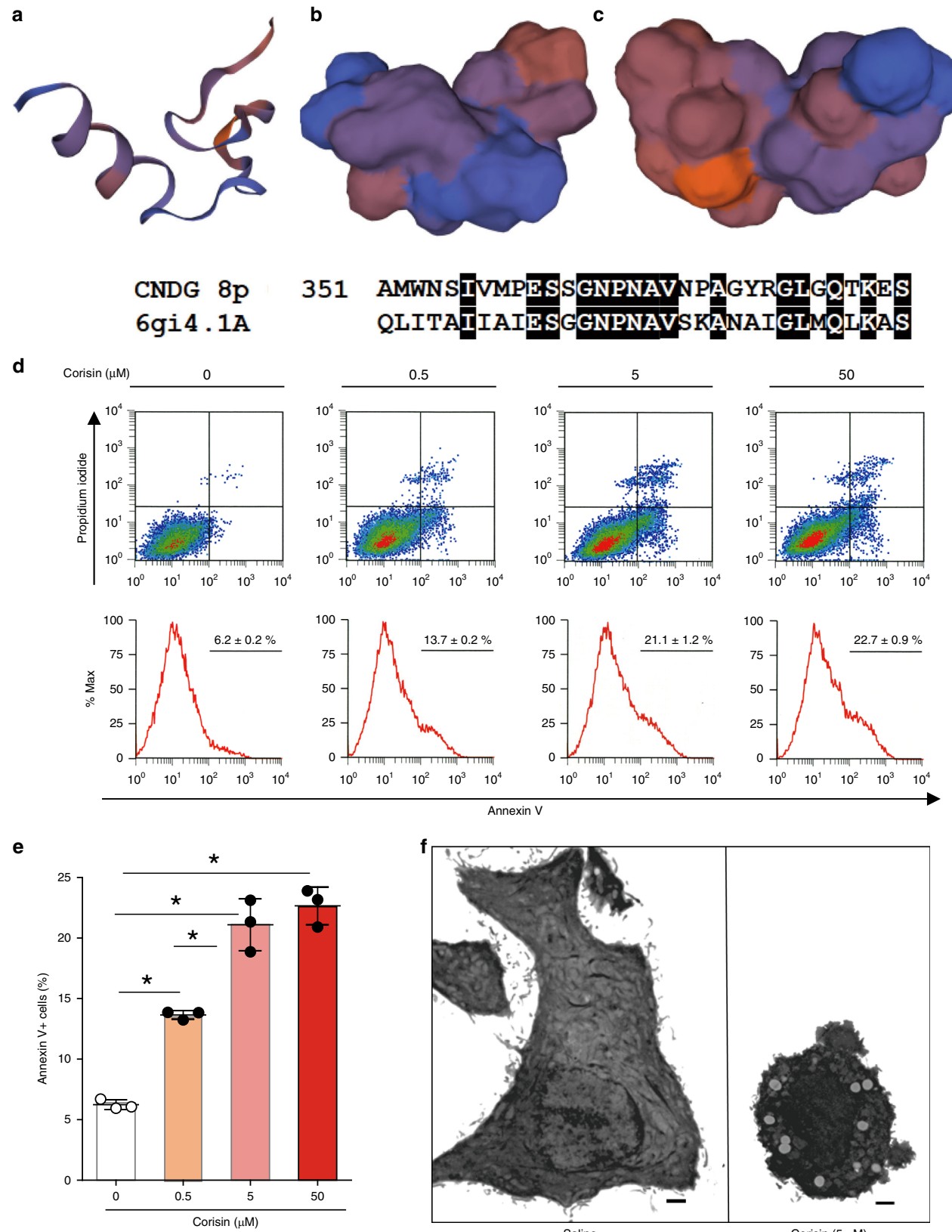

a derivative of the transglycosylases close to the ancestral sequence splits into the two IsaA clusters (IsaA-1 and IsaA-2) and from IsaA-1 related sequences, the proteins designated SceD members likely evolved (SceD-1, SceD-2, SceD-3, and SceD-4) (Supplementary Fig. 21). The multiple alignment of the IsaA and the SceD amino acid sequences revealed, in general, conservation of amino acid residues representing the pro-apoptotic corisin, and thus highlighting their functional significance (Supplementary Fig. 22). The

**Fig. 5 Structure prediction and apoptotic activity of corisin.** The full length of the transglycosylase 351 from *Staphylococcus nepalensis* strain CNDG was submitted to the protein structure prediction search engine at Swiss model (https://swissmodel.expasy.org/) but the full-length structure aligned with proteins of <20% identity. **a–c** However, using the pro-apoptotic peptide alone, a model was predicted by the search engine through a structural alignment with the entry 6gi4.1.A, an endo-type membrane-bound lytic murein transglycosylase A, with which it (peptide) shares 46.88% identity. **d, e** Flow cytometry analysis of A549 alveolar epithelial cells cultured for 48 h in DMEM medium containing increasing concentrations of the pro-apoptotic peptide (corisin from Peptide Institute Incorporation, Japan). Each concentration with $n = 3$ (triplicates). Bars indicate the means ± S.D. Statistical analysis by ANOVA and Tukey's test. *$p < 0.001$. **f** Electron micrographs of A549 cells cultured in the presence of saline or corisin. Representative microphotographs out of two experiments with similar results are shown. Scale bars indicate 1 μm. The source data underlying **e** are provided in the Source Data file.

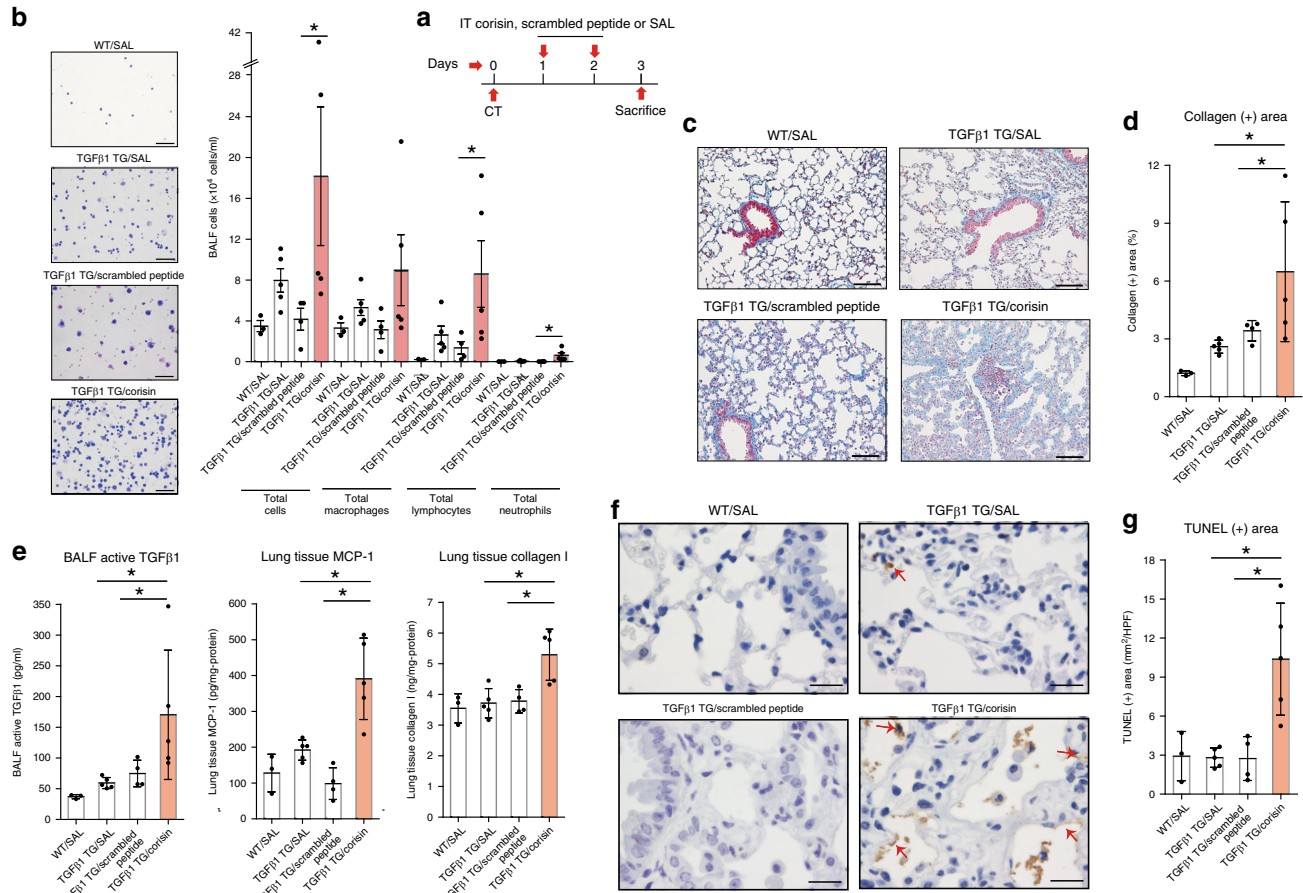

**Fig. 6 Corisin exacerbates lung fibrosis in transforming growth factor β1 transgenic mice. a** Transforming growth factorβ1 transgenic mice (TGFβ1 TG) mice with matched lung fibrosis based on computed tomography (CT) score received intra-tracheal corisin ($n = 5$) or scrambled peptide ($n = 4$) or 0.9% NaCl solution (SAL; $n = 5$) on days 1 and 2 and sacrificed on day 3 to evaluate changes in lung inflammation and fibrosis. Wild-type (WT) mice without lung fibrosis ($n = 3$) treated with 0.9% NaCl solution were used as controls. $n = 3$ in WT/SAL, $n = 4$ in TGFβ1 TG/scrambled peptide, $n = 5$ in TGFβ1 TG/ SAL and TGFβ1 TG/corisin groups. **b** Counting of bronchoalveolar lavage fluid cells. Scale bars indicate 100 μm. $n = 3$ in WT/SAL, $n = 4$ in TGFβ1 TG/ scrambled peptide, $n = 5$ in TGFβ1 TG/SAL and TGFβ1 TG/corisin groups. Statistical analysis by two-tailed Mann–Whitney U test. *$p < 0.05$. **c, d** Quantification of collagen area by WinROOF software. Scale bars indicate 100 μm. $n = 3$ in WT/SAL, $n = 4$ in TGFβ1 TG/scrambled peptide, $n = 5$ in TGFβ1 TG/SAL and TGFβ1 TG/corisin groups. Bars indicate the means ± S.D. Statistical analysis by ANOVA with Newman-Keuls test. *$p < 0.05$. **e** The concentrations of TGFβ1, monocyte chemoattractant protein (MCP)-1 and collagen I were measured by enzyme immunoassays. $n = 3$ in WT/SAL, $n = 5$ in TGFβ1 TG/SAL and TGFβ1 TG/corisin, and $n = 4$ in TGFβ1 TG/scrambled peptide groups. Bars indicate the means ± S.D. Statistical analysis by ANOVA with Newman-Keuls test. *$p < 0.05$. **f, g** DNA fragmentation was evaluated by staining through terminal deoxynucleotidyl transferase dUTP Nick-End Labeling (TUNEL). Scale bars indicate 50 μm. $n = 3$ in WT/SAL, $n = 5$ in TGFβ1 TG/SAL and TGFβ1 TG/corisin, and $n = 4$ in TGFβ1 TG/scrambled peptide groups. Bars indicate the means ± S.D. Statistical analysis by ANOVA with Newman-Keuls test. *$p < 0.01$. The source data underlying **b, d, e, g** are provided in the Source Data file.

amino acid sequence identity of corisin homologous transglycosylases from *Staphylococcus xylosus*, *Staphylococcus cohnii*, and *Staphylococcus nepalensis* was 100%. Furthermore, these staphylococci shared more than 98% identity with the corresponding corisin regions of transglycosylases from other members of the IsaA-1 and IsaA-2 clusters, and 60% identity with the corresponding regions in members of the SceD clusters (Supplementary Fig. 22). The genomic context of genes clustering around the transglycosylase (synteny) tended to be conserved in *Staphylococcus cohnii* and *Staphylococcus nepalensis* (Supplementary Fig. 23a).

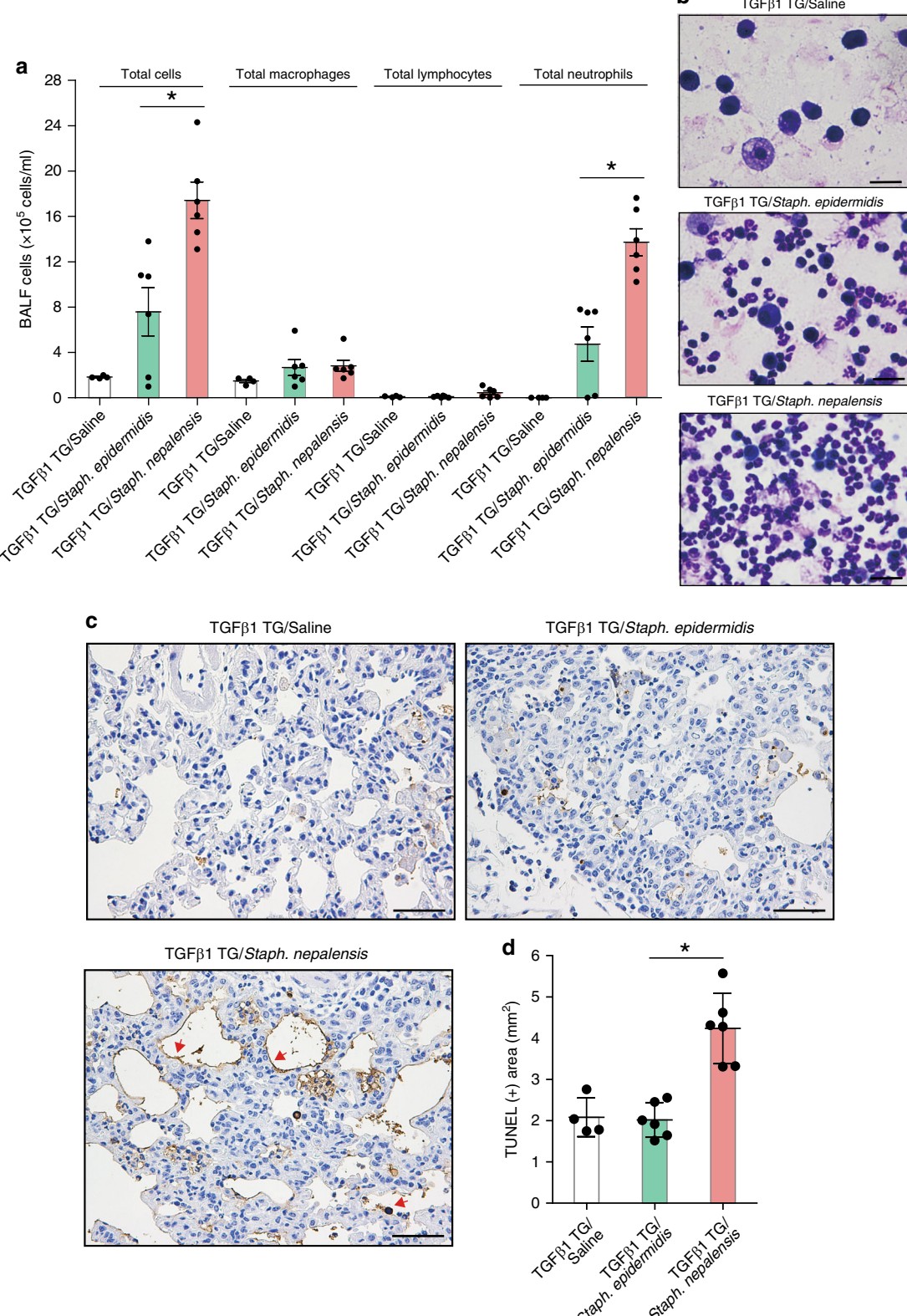

**Horizontal gene transfer of the corisin-encoding gene**. Sequence alignment and comparative genome analysis revealed that a pathogenic strain of *Streptococcus*, i.e., *Streptococcus pneumoniae* strain N, implicated in respiratory tract disease, contains a transglycosylase (COE35810) with a peptide sequence almost identical (a single amino acid change) to corisin. A further examination of the genome of this bacterium unveiled a second homolog (COE67256) of the corisin-containing polypeptide (Supplementary Fig. 22). To understand how *Streptococcus pneumoniae* strain N might have acquired the corisin-encoding gene, since its polypeptide sequence is highly conserved only in diverse *Staphylococcus* spp., we performed a search in the

**Fig. 7 Intra-pulmonary Instillation of the corisin-containing *Staphylococcus nepalensis* strain CNDG exacerbates pulmonary fibrosis in hTGFβ1 TG mice.**
Transforming growth factor (TGF)β1 transgenic (TG) mice with matched lung fibrosis based on computed tomography (CT) score received intra-tracheal instillation of saline ($n = 4$), *Staphylococcus epidermidis* ATCC14990 ($n = 6$), or *Staphylococcus nepalensis* strain CNDG ($n = 6$) as described under methods. **a, b** The number of cells in bronchoalveolar lavage fluid (BALF) was counted and then stained with Giemsa on the second day after intra-tracheal instillation of saline or each bacterium. Scale bars indicate 100 μm. Bars indicate the means ± standard error of the means. Representative microphotographs out of two experiments with similar results are shown. Statistical analysis by ANOVA with Tukey's test. *$p < 0.01$. **c, d** DNA fragmentation was evaluated by staining with terminal deoxynucleotidyl transferase dUTP Nick-End Labeling (TUNEL), and then quantified using the image WinROOF software. Representative microphotographs out of two experiments with similar results are shown. Scale bars indicate 50 μm. Bars indicate the means ± S.D. Statistical analysis by ANOVA with Tukey's test. *$p < 0.001$. The source data underlying **a**, **d** are provided in the Source Data file.

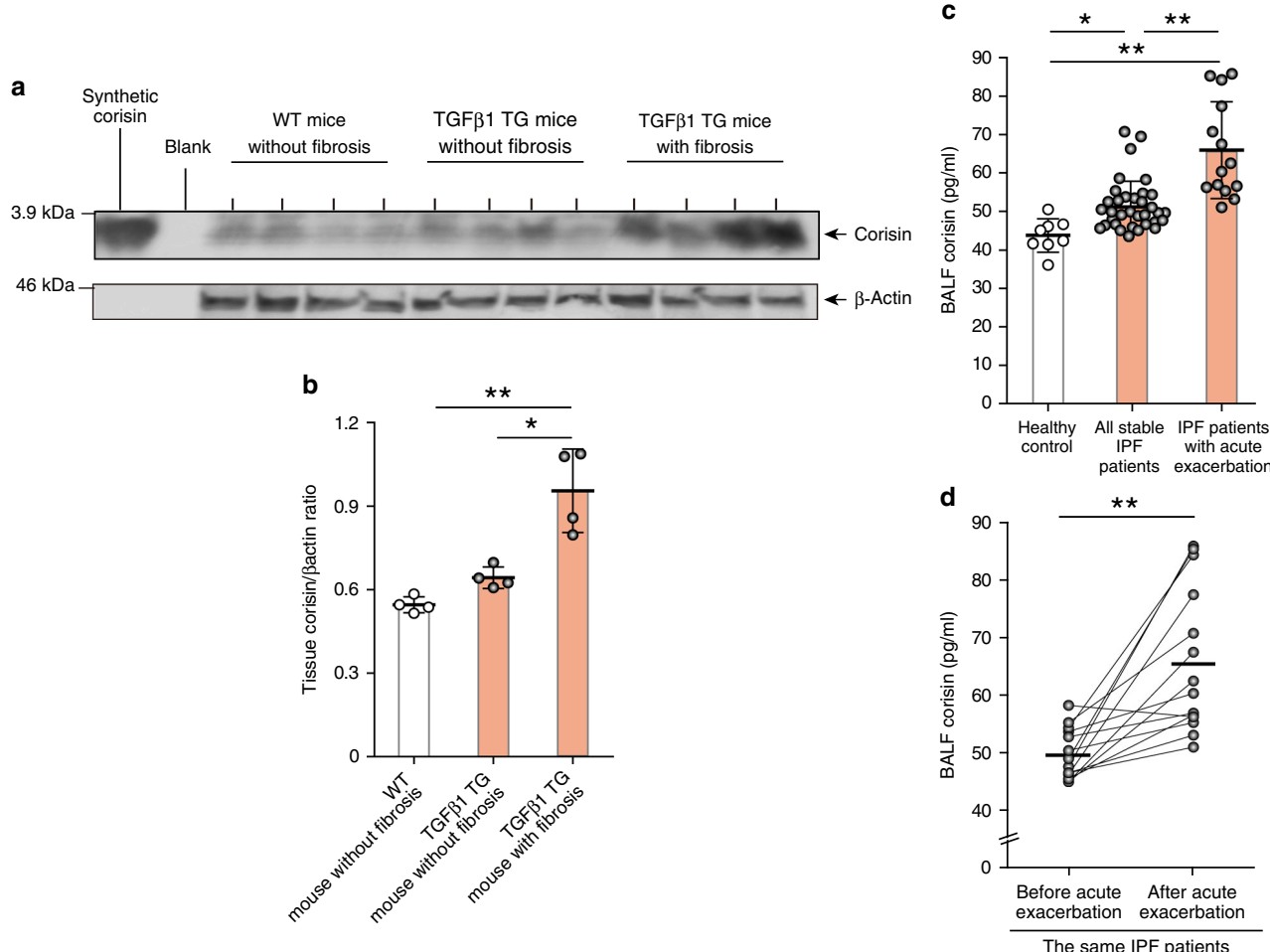

**Fig. 8 Significant increase in corisin level in TGFβ1 TG mice with lung fibrosis and in patients with idiopathic pulmonary fibrosis. a, b** Western blotting of corisin in lung tissue from wild-type and transforming growth factor (TGF)β1 transgenic (TG) mice performed as described under methods and the ratio of corisin to β-actin. Quantification was performed using ImageJ, a public domain image processing program. Each mouse group with $n = 4$. Bars indicate the means ± S.D. Statistical analysis by ANOVA with Tukey's test. *$p < 0.01$, **$p < 0.001$. **c** Corisin was measured using a competitive enzyme immune assay as described under methods. Healthy controls, $n = 8$; all stable idiopathic pulmonary fibrosis (IPF) patients, $n = 34$; IPF patients with acute exacerbation, $n = 14$. Bars indicate the means ± S.D. Statistical analysis by ANOVA with Newman-Keuls test. *$p < 0.05$, **$p < 0.001$. **d** Analysis of bronchoalveolar lavage fluid levels of corisin in the same IPF patients before and after acute exacerbation, $n = 14$. Bars indicate the means. Statistical analysis by two-tailed Mann–Whitney *U* Test. **$p < 0.001$. The source data underlying **a–d** are provided in the Source Data file.

Genbank database and found that the polypeptide (COE35810) yields 98–100% identity with transglycosylases in different strains of *Staphylococcus warneri* (WP_002467055, WP_050969398, WP_126403073, and WP_107532308) (Supplementary Fig. 23b, c). Despite the one or two changes in amino acids at the N-terminal region of the polypeptides, the corisin peptide sequences within these transglycosylases are invariant. We further examined the genomic context of these genes in *Streptococcus pneumoniae* strain N in comparison with a *Staphylococcus warneri* strain, and found a clear conservation of synteny, despite some differences in annotation (Supplementary Fig. 23d). We, therefore, hypothesized that the transglycosylase gene and other genes linked to it in *Streptococcus pneumoniae* strain N were acquired from a *Staphylococcus warneri* strain or a related species. Significantly, strains of another pathogenic bacterium known to inhabit the human lung, i.e., *Mycobacterium [Mycobacteroides] abscessus* harbors a variant of the transglycosylase (SKT99287) and based on a similar analysis, as described above for *Streptococcus pneumoniae* strain N, we inferred that the transfer was from *Staphylococcus hominis* or related species (Supplementary Fig. 23e, f). We

then performed an experiment that confirmed that the synthetic corisin from the transglycosylase of *Streptococcus pneumoniae* (contains 1 amino acid change from *Staphylococcus nepalensis* derivative) also induces apoptosis of A549 alveolar epithelial cells (Supplementary Fig. 22 and Supplementary Fig. 24a, b). From these observations, it appears that non-*Staphylococcus* organisms that have the genes encoding transglycosylases with very high homology to the *Staphylococcus nepalensis* transglycosylase 351 are lung-associated, suggesting a case of horizontal gene transfer from *Staphylococcus* strains inhabiting the lung.

## Discussion

TGFβ1 is a pleiotropic cytokine with a pivotal role in the pathogenesis of pulmonary fibrosis by its potent stimulatory activity on extracellular matrix synthesis, activation, differentiation and migration of myofibroblasts, epithelial-to-mesenchymal transition, and production of pro-fibrotic factors and apoptosis of alveolar epithelial cells[17,18]. The development of pulmonary fibrosis in our mouse overexpressing TGFβ1 is a proof-of-concept for the critical role of this cytokine in tissue fibrosis[11]. In addition, TGFβ1 may promote exacerbation of pulmonary fibrosis by directly suppressing both the innate and adaptive immune systems leading to enhanced host susceptibility to infection[19–21]. Previous studies have shown that high salt concentration impairs host defense mechanisms by suppressing the activity of antimicrobial peptides or by altering the population of immune cells[22–24]. Therefore, TGFβ1 may also indirectly affect the host immune response by favoring the accumulation of salt in the extracellular space[25,26]. Abnormal extracellular storage of salt may result from TGFβ1-mediated negative regulation of the surface expression of epithelial sodium and chloride channels leading to decreased transport of $Na^+$ and $Cl^-$ ions from the alveolar airspaces across the epithelium[25–29]. Consistent with these findings, in the present study, we found in lung tissue a significant increase of sodium level in TGFβ1 TG mice with lung fibrosis compared to WT mice, a significant positive correlation of sodium level with fibrotic markers and pro-fibrotic cytokines, and a significant negative correlation of sodium level with lymphocyte count and sodium and chloride channels. A recent single-cell RNA sequencing study showing that expression of several cell membrane sodium and chloride transporters is significantly altered in alveolar epithelial cells from IPF patients, suggests that ion transmembrane trafficking is disrupted in pulmonary fibrosis and favors the accumulation of salt in this fibrotic disease[30]. Sodium storage appears to require the presence of fibrotic matrix, because we found no difference in the lung sodium level between TGFβ1 TG mice without fibrosis and WT mice. In this connection, previous studies have shown that sodium is stored in extracellular spaces in an osmotically inactive form by binding to negatively charged glycosaminoglycans, which are abundant in the extracellular matrix of fibrotic tissues[31–35]. Overall, these observations suggest that the fibrotic tissue is a salty microenvironment (see model in Supplementary Fig. 25) with abnormal immune and healing responses. The identification of halophilic bacteria in the lung of IPF patients by previous studies support these findings[8,9].

Acute exacerbation is a devastating complication of IPF[36]. Nearly 50% of patients dying from IPF have a prior history of acute exacerbation and the life expectancy of patients with a previous acute exacerbation is only 3–4 months[37–41]. There is currently no optimal therapy for acute exacerbation of IPF[36]. An international working group in 2016 proposed to classify this complication into triggered (identified event: post-procedure, drug toxicity, infection, aspiration) or idiopathic (unidentified inciting event) acute exacerbation[36]. Recent data associating acute

exacerbation with the lung microbiome and with the host immunosuppressive states, and retrospective studies showing the preventive effect of antibiotic therapy suggest the role of infection in the pathogenesis of acute exacerbation and progression of pulmonary fibrosis[7,42–45]. Further, a double-blind, randomized, placebo-controlled study showing improvement of symptoms and exercise capacity in progressive IPF patients treated with co-trimoxazole, and a subsequent double-blind follow-up and multicenter study showing significant reduction of mortality with better quality of life and less respiratory tract infections in IPF patients treated with co-trimoxazole also support the pathogenic role of bacteria in lung fibrosis[46,47]. A previous report showed that bacteria of the *Staphylococcus* and *Streptococcus* genera worsen the clinical outcome of IPF patients, suggesting their implication in the disease progression and pathogenesis[7]. Studies showing the relative abundance of *Staphylococcus* or *Streptococcus* genera in the fibrotic lung and its significant correlation with the host immune response in IPF patients further support the contribution of these bacteria genera in the pathogenesis of pulmonary fibrosis[6,42,48–52]. However, the precise mechanism remains unclear.

In the present study, we hypothesized that a salty culture medium would mimic the in vivo salty fibrotic tissue and thus would favor the growth of bacteria involved in the pathogenesis of lung fibrosis. We detected growth of bacteria of the genus *Staphylococcus* in the hypersaline media inoculated with fibrotic tissues from hTGFβ1 TG mice with advanced fibrosis, and the whole genome sequence of a pure bacterial culture revealed that it corresponds to *Staphylococcus nepalensis* that we categorized as "strain CNDG". The culture supernatant of this bacterium induced apoptosis of alveolar epithelial cells, and subsequent chromatography, mass spectrometry and gene sequence analysis showed that apoptosis was induced by a peptide we called corisin that corresponds to a segment of transglycosylase 351 from *Staphylococcus nepalensis* strain CNDG. The higher apoptotic activity of supernatants from bacteria cultured under high-salt conditions may be due to salt-dependent stimulation of bacteria growth or increased bacterial expression of the corisin-containing transglycosylase, a related protein that has been reported to be enhanced in expression in *Staphylococcus aureus* under similar condition[53].

In additional experiments, we detected the peptide in the lung from hTGFβ1 TG mice with progressive lung fibrosis and from patients with IPF and found that intratracheal instillation of synthetic corisin or *Staphylococcus nepalensis* strain CNDG induces acute exacerbation of pulmonary fibrosis in association with extensive apoptosis of alveolar epithelia cells (see model in Supplementary Fig. 25). Accelerated apoptosis of alveolar epithelial cells plays a central role in the pathogenesis of acute exacerbation in pulmonary fibrosis[16,54]. Therefore, based on these observations, corisin emerges as a strong candidate in the microbial factors that might trigger acute exacerbation in patients with idiopathic pulmonary fibrosis.

We found that the sequence of corisin has high homology with a region in a membrane-bound lytic transglycosylase. Lytic transglycosylases are bacterial enzymes reported to cleave the peptidoglycan component of the bacterial cell wall[55] and further perform other essential cellular functions, such as cell-wall synthesis, remodeling, resistance to antibiotics, insertion of secretion systems, flagellar assembly, release of virulence factors, sporulation, and germination[55]. Transglycosylases are ubiquitous in bacteria and an individual species may produce multiple transglycosylases with functional redundancy, to compensate in case of loss or inactivation of any member[56,57]. In our current study, the complete genome sequence showed that *Staphylococcus nepalensis* strain CNDG produces six transglycosylases, of which

the transglycosylase 351, a member of the IsaA-1 cluster, harbors the corisin sequence. The full-length transglycosylase 351 did not induce apoptosis of lung epithelial cells, suggesting that the corisin peptide is active only after being released from the full-length protein. The mechanism of this peptide shedding is unknown but the genomic context of the *Staphylococcus nepalensis* CNDG strain showing the presence of peptidases surrounding the transglycosylase 351, suggests that they may be involved in the release of the deadly peptide.

We found that, in addition to *Staphylococcus nepalensis* strain CNDG, sequences similar to corisin are highly conserved in several transglycosylases from other *Staphylococcus* species and some members of the microbial community that inhabit the normal or fibrotic lungs, including strains of *Streptococcus pneumoniae* and *Mycobacterium abscessus*[51,58–60]. This observation suggests that a broad range of bacteria may be the source of corisin in pulmonary fibrosis. While the present study is the initial report on the pathogenicity of a peptide derived from an IsaA homolog in a strain of *Staphylococcus*, homologous proteins (i.e., IsaA and SceD) have been reported in *Staphylococcus aureus* to be involved in virulence. The *Staphylococcus aureus* IsaA in that report[53] corresponds to YP_501340 in the alignment shown in Supplementary Fig. 22, while the SceD, in the same report, has a variant of corisin similar to those in the SceD-1 to SceD-4 polypeptides (Supplementary Fig. 22). Thus, although relevant, the characterized transglycosylases in *Staphylococcus aureus* are quite different from the *Staphylococcus nepalensis* transglycosylase characterized in the present study. It is of note, however, that *Staphylococcus aureus* has an uncharacterized IsaA transglycosylase with a highly conserved corisin sequence (Supplementary Fig. 21, IsaA-2, SUK04795.1), which may suggest that a similar mechanism as the corisin processing described in the present study exists in *Staphylococcus aureus*.

*Streptococcus pneumoniae* and *Staphylococcus* species also frequently cause severe pulmonary infections with high in-hospital mortality rate in IPF patients[20,58,61]. Given the growing evidence that alveolar cell apoptosis plays a central role in the pathogenesis and exacerbation of IPF[62], it is reasonable to speculate that shedding of deadly peptides constitutes an important contribution to the loss of functional lung alveolar cells and to the poor clinical outcome in patients with complications of microbial infection. Another mechanism that may further contribute to bacterial virulence and invasiveness is horizontal transfer of bacterial genes[63]. Here we found that strains of *Streptococcus pneumoniae*, *Mycobacterium [Mycobacteroides] abscessus* and several *Staphylococcus* species shared highly similar genome context (synteny) and sequence homology of transglycosylases containing the corisin sequence, suggesting involvement of horizontal gene transfer in the acquisition of this virulence factor. *Staphylococcus* and *Streptococcus* genera are common members of the human microbiota[64]. Therefore, if determined that the corisin related peptides identified in the present study have similar apoptotic impact on human cells from other sites or organs, such as the kidney and liver, our view of infections by these bacteria will require re-assessment.

In light of the increasing evidence indicating the participation of the lung microbial population in the pathogenesis of IPF, the identification of corisin as a disease exacerbator substantiates the role of apoptosis in fibrotic diseases, provides a novel diagnostic marker and therapeutic target in IPF, and opens a new avenue for investigating the role of microbiomes in organ fibrosis.

## Methods

**Reagents**. The human lung epithelial cell line A549 and hypersaline media (ATCC media 1097, 2168) were from the American Type Culture Collection (Manassas, VA), Dulbecco's Modified Eagle Medium (DMEM) from Sigma-Aldrich (Saint Louis, MO) and fetal bovine serum (FBS) from Bio Whittaker (Walkersville, MD). L-glutamine, penicillin and streptomycin were from Invitrogen (Carlsbad, CA). Normal human bronchial epithelial (NHBE) cells were from Clonetics (Walkersville, MD). Synthetic peptides were prepared and provided by Peptide Institute Incorporation (Osaka, Japan) and by ThermoFisher Scientific (Waltham, MA, USA).

**Subjects**. This study comprised 34 Japanese patients with stable idiopathic pulmonary fibrosis (IPF; mean age: 71.7 ± 6.6 years-old, males: 29, females: 5) and eight Japanese male healthy volunteers (38.3 ± 6.1 years old). Supplementary Table 3 describes the characteristics of the patients. Diagnosis of idiopathic pulmonary fibrosis was done following accepted international criteria[65,66]. Bronchoscopy study was performed following guidelines of the American Thoracic Society and bronchoalveolar lavage fluid (BALF) samples were collected from all 34 IPF patients and 8 healthy volunteers[65]. BALF samples during acute exacerbation of the disease were available in 14 out of the 34 participant IPF patients. Aliquots of unprocessed bronchoalveolar lavage fluid (BALF) collected into sterile tubes were stored at −80 °C until analysis.

**Animals**. We used transgenic (TG) mice in a C57BL/6J background with lung-specific overexpression of the latent form of human TGFβ1 that have been previously characterized[8,11]. These TGFβ1 TG mice spontaneously develop pulmonary fibrosis from 10-weeks of age, and showed similarity to the disease in humans[8,11]. C57BL/6J wild-type mice were used as controls. In some of the experiments, TGFβ1 TG mice without lung fibrosis were used as controls; however, the number of mice born with the human TGFβ1 transgene positive but with no phenotype (lung fibrosis) is extremely scarce or rare and thus it was very difficult to include them in all experiments. All mice were maintained in a specific pathogen-free environment under a 12-h light/dark cycle in the facility for experimental animals of Mie University. Genotyping of TG mice were carried out by standard PCR analysis using DNA isolated from the tail of mice and primer pairs (Supplementary Table 5)[11].

**Computed tomography**. We performed radiological evaluation of the chest using a micro-CT (Latheta LCT-200, Hitachi Aloka Medical, Tokyo, Japan). Mice received isoflurane inhalation as anesthesia and were placed in a prone position for data acquisition[67]. Six specialists in respiratory diseases blinded to the treatment groups scored the chest CT findings based on the following criteria: score 1, normal lung findings; 2, intermediate findings; 3, slight lung fibrosis; 4, intermediate findings; 5, moderate lung fibrosis; 6, intermediate findings; and 7, advanced lung fibrosis (Supplementary Fig. 1a)[67]. We used the Ashcroft lung fibrosis score and the hydroxyproline content of the lungs to validate the CT findings (Supplementary Fig. 1b).

**Evaluation of pulmonary fibrosis in mice**. Under profound anesthesia, we collected bronchoalveolar lavage fluid for biochemical analysis and cell counting. Briefly, bronchoalveolar lavage fluid was performed by cannulating the trachea with a 20-gauge needle and infusing saline solutions into the lungs[68]. The samples were centrifuged and the supernatants were stored at −80 °C until analysis. The cell pellets were re-suspended in physiological saline solution and the number of cells was counted. A nucleocounter from ChemoMetec (Allerød, Denmark) was used for cell counting and the cells were stained with May–Grünwald–Giemsa (Merck, Darmstadt, Germany) to count differential cells. We then sacrificed mice by anesthesia overdose, and resected the lungs to fix in formalin, embed in paraffin and prepare for hematoxylin and eosin staining. The severity of lung fibrosis was quantitated based on the Ashcroft criteria[67]. The level of TGFβ1 was measured using commercial enzyme immunoassay kit from BD Biosciences Pharmingen (San Diego, CA).

**Ethical statement**. All subjects participating in the clinical investigation provided written informed consent and the study protocol was approved by the Ethical Committees for Clinical Investigation of Mie University (approval No: H2019064, date: 25/04/2019), Matsusaka Municipal Hospital (approval date: 11/06/2014), and Chuo Medical Center (approval No 2014-6, date: 02/09/2014) and conducted following the Principles of the Declaration of Helsinki. The Recombinant DNA Experiment Safety Committee (approval No: I-614 (henko1); date: 2013/15/12; approval No: I-708, date: 13/02/2019) and the Committee for Animal Investigation of Mie University approved the experimental protocols (approval No: 25-20-hen1-sai1, date: 23/07/2015; approval No: 29-23, date: 15/-01/2019) and all procedures were performed in accordance with internationally approved principles of laboratory animal care published by the National Institute of Health (https://olaw.nih.gov/).

**Lung sampling for in vitro culture**. Under sterile conditions, we excised the left and right lungs after euthanasia of mice by intraperitoneal injection of an overdose of pentobarbital and placed the tissue into sterile tubes and immediately stored them at −80 °C until use.

**Measurement of lung tissue Na$^+$.** We removed the lungs from TGFβ1 TG mice with or without lung fibrosis and from WT mice. The samples were sent to Shimadzu Techno-Research, Incorporation (Kyoto, Japan) for the measurement of tissue sodium content by using microwave analysis/inductively coupled plasma mass spectrometry (ICP-MS), the microwave ashing system ETHOS-TC (Milestone General) and the ICP-MS system 7700× (Agilent Technologies, Santa Clara, CA)[69,70].

**Evaluation of lung tissue immune cells.** To isolate lung immune cells, after sacrificing mice by anesthesia overdose, we incised and minced the lung tissue with scissors into 2–3 mm pieces, incubated in 0.5 mg/ml collagenase solution for 30 min at 37 °C, and then filtered through a stainless steel mesh. Lung cells were separated and purified using isotonic 33% Percoll (Sigma-Aldrich, St. Louis, MO) solution. We then detected the lung immune cells by flow cytometry using the antibodies described in Supplementary Table 4.

**Evaluating the effect of the pro-apoptotic corisin in mice.** Three groups of TGFβ1 TG mice (each $n = 5$ or $n = 4$) with matched grade of lung fibrosis as assessed by CT score underwent intratracheal instillation of corisin or scrambled peptide or 0.9% NaCl solution on days 1 and 2 and killed on day 3 to evaluate changes in lung inflammation and fibrosis. WT mice ($n = 3$) without lung fibrosis treated with 0.9% NaCl solution were used as controls.

**Intra-tracheal instillation of *Staphylococcus nepalensis*.** We administered by oral gavage 200 μl of a solution containing a cocktail of antibiotics including vancomycin (0.5 mg/ml), neomycin (1 mg/ml), ampicillin (1 mg/ml), metronidazole (1 mg/ml) and gentamycin (1 mg/ml) once a day for 4 days to three groups of TGFβ1 TG mice. All mice had a matched grade of lung fibrosis as assessed by CT score. On the 5th day, one group of mice received intra-tracheal instillation of $1 \times 10^8$ colony forming units (75 μl) of *Staphylococcus nepalensis* strain CNDG or *Staphylococcus epidermidis* ATCC14990 and sacrificed after 2 days. Germ-free TGFβ1 TG mice treated with 0.9% NaCl solution were used as controls.

**Bacteria isolation, culturing, and spent medium preparation.** Lungs from TGFβ1 TG mice with lung fibrosis and from WT mice were used for in vitro microbial culture. The lung tissue specimens were washed with PBS and inoculated into ATCC medium 1097 (8% NaCl) and cultured at 37 °C with shaking at 220 rpm until growth was visible. Bacterial colonies were isolated by plating the liquid medium-cultured organisms on an ATCC medium 1097 agar plates. Each single colony was inoculated into liquid ATCC medium 1097 (8% NaCl) and cultured at 37 °C at 220 rpm for 24 h. The cultures were centrifuged for 5 min at 4000 rpm at 4 °C to pellet the cells, and the resulting supernatant was filtered through a MILLEXGP filter unit (0.22 μm, Millipore) to remove any remaining cells and used as the spent bacterial medium.

**Phase-contrast microscopy.** We harvested bacterial cells from a single colony in exponential phase growth, immersed in a fixative overnight at 4 °C and collected microphotographs using phase contrast microscopy (Frederick Seitz Materials Research Lab, UIUC)[71].

**Genomic DNA sequencing and genome annotation.** Genome sequencing was carried out with a combination of Oxford Nanopore Sequencing and Illumina Miseq nano sequencing that produced 6.3 Gbases and 1.6 million ($2 \times 250$) nucleotides with perfect Qscores. Briefly, genomic DNA from the bacterial strain (400 ng) was converted into a Nanopore library with the Rapid Barcoding library kit SQK-RAD004. The library was sequenced on a SpotON R9.4.1 FLO-MIN106 flowcell for 48 h on a GridION sequencer. Base-calling was performed with Guppy 1.4.3, and demultiplexing was done with Porechops 0.2.3. The majority of the reads were 6–30 kb in length, although reads as long as 94 kb were also obtained. The Illumina Miseq sequencing was carried out by preparing shotgun genomic libraries with the Hyper Library construction kit from Kapa Biosystems (Roche). The library was quantitated by qPCR and sequenced on one MiSeq Nano flowcell for 251 cycles from each end of the fragments using a MiSeq 500-cycle sequencing kit version 2. Fastq files were generated and demultiplexed with the bcl2fastq v2.20 Conversion Software (Illumina).

A workflow was developed to perform four assemblies as follows, primarily to assess quality using different assembly strategies to find the best overall assembly. Initial assembly of the Oxford Nanopore data was carried out using Canu[72], followed by polishing using Nanopolish[73] and Pilon (utilizing the Illumina MiSeq reads)[74], and finally the genome was re-oriented using Circlator[75]. Another hybrid genome assembly was carried out using SPAdes[76], followed by reorienting the genome using Circlator. A hybrid genome assembly was also carried out using Unicycler[77]. The final hybrid genome assembly was generated using Unicycler, with the Canu assembly above as the assembly backbone.

All assemblies were quality-assessed using BUSCO[78] and QUAST[79] and compared to a relevant reference genome using MUMmer[80]. Assemblies were then followed by an annotation run using the tool Prokka[81]. After evaluation, the best overall assembly was determined using the best overall BUSCO scores in combination with overall assembly metrics.

**Assessment of the molecular weight of the apoptotic factor.** Bacterial culture supernatants were prepared from cultures grown in *Halomonas* medium (8% NaCl, 0.75% casamino acids, 0.5% proteose peptone, 0.1% yeast extract, 0.3% sodium citrate, 2% magnesium sulfate heptahydrate, 0.05% potassium phosphate dibasic, 0.05% ammonium iron (II) sulfate hexahydrate) with shaking at 37 °C. Bacterial cells were removed by centrifugation (17,000 × g, for 10 min at 4 °C) and filtration through 0.2 μm filters (Corning). Supernatants were size fractionated into high molecular weight (HMW) and low molecular weight (LMW) fractions by ultra-filtration with Ultracel-10K filters (Amicon), separated into aliquots and frozen at −20 °C. In some experiments, bacterial culture supernatants were heat-treated (85 °C, 15 min) before size fractionation. Equal volumes of supernatants were separated by 17.5% Tricine-sodium dodecyl sulfate-polyacrylamide gel electrophoresis (SDS-PAGE) and silver-stained using the Daiichi 2-D Silver Staining Kit (Daiichi, Tokyo, Japan).

**Cell culture.** The A549 and NHBE cells were cultured in DMEM supplemented with 10% fetal calf serum, 0.03% (w/v) L-glutamine, 100 IU/ml penicillin and 100 μg/ml streptomycin in a humidified, 5% CO$_2$ atmosphere at 37 °C. We used A549 cell lines in most experiments because they have higher potential growth and mimic more the phenotype of alveolar type II cells than primary NHBE cells;[82,83] and in addition, these primary cells usually easily change phenotype or become senescent after a short period of culture.

**Fractionation of the bacterial supernatant.** The bacterial culture supernatant (2 liters) was successively partitioned between *n*-hexane and water, and then ethyl acetate and water (2 L each, two times) (Supplementary Fig. 5). The concentrated proteins were further concentrated under reduced pressure and then extracted with ethanol (2 liters each, two times). The ethanol-soluble portion (7.96 g) was fractionated by octadecyl silane gel flash column chromatography (5%; 10%, 20%, 50% methanol and methanol, 0.5 liter each) to obtain 42 fractions (fractions 1–42). Fraction 42 (185.3 mg of proteins) was further separated by Sep-Pak (80% acetonitrile, methanol, and chloroform). Fraction 42–80% acetonitrile (75.6 mg of proteins) was separated by reverse-phase HPLC (C8, 80% methanol) to afford 22 fractions (fractions 42–80% acetonitrile-1–22) (see HPLC data in Supplementary Information).

**Mass spectrometry.** Dried samples were suspended in 0.1% formic acid (FA) in 5% acetonitrile (ACN), and 2 μg of peptides were injected into a Thermo UltiMate 3000 UHPLC system. Reversed phase separation of sample peptides was accomplished using a 15-cm Acclaim PepMap 100 C18 column with mobile phases of 0.1% FA in water (A) and 0.1% FA in ACN (B). Peptides were eluted using a gradient of 2% B to 35% B over 60 min followed by 35% to 50% B over 5 min at a flow rate of 300 μl/min. The UHPLC system was coupled online to a Thermo Orbitrap Q-Exactive HFX (Biopharma Option) mass spectrometer operated in the data dependent mode. Precursor scans from 300 to 1500 $m/z$ (120,000 resolution) were followed by collision induced dissociation (CID) of the most abundant precursors over a maximum cycle time of 3 s (3e4 AGC, 35% NCE, 1.6 $m/z$ isolation window, 60 s dynamic exclusion window).

The raw data were analyzed using Mascot 1.6 against a custom database consisting of the protein library of the *Staphylococcus nepalensis* CNDG genomic DNA, and the large and small plasmids encoded polypeptides (total of 3541 protein sequences). No enzyme was specified. Peptide mass tolerance and fragment mass tolerances were set to 10 ppm and 0.1 Da, respectively. Variable modifications included oxidation of methionine residues (see mass spectrophotometry data in Supplementary Information).

**Apoptosis assay.** A549 and NBHE cells ($4 \times 10^5$ cells/well) were seeded into 12-well plates, cultured to sub-confluency, washed and then cultured in serum free medium containing 10% of each bacterial supernatant for 48 h. Non-inoculated hypersaline medium was used as control. The cells were analyzed for apoptosis by flow cytometry (FACScan, BD Biosciences, Oxford, UK) after staining with fluorescein-labeled annexin V and propidium iodide (FITC Annexin V Apoptosis Detection Kit with PI, Biolegend, San Diego, CA). Flow cytometry gating strategy used in the experiments is described in Supplementary Fig. 26. Under physiological conditions, phosphatidylcholine is exposed externally while phosphatidylserine (PS) is located on the inner surface of the lipid bilayer of cellular membranes[84]. During apoptosis, PS is translocated from the cytoplasmic face of the plasma membrane to the cell surface[84]. Annexin V shows a strong affinity in binding to phosphatidylserine in a Ca$^{2+}$-dependent manner and thus it is generally used as a probe for detecting apoptosis[85].

**Western blotting.** The cells for Western blot analysis were washed twice with ice-cold phosphate-buffered saline and then lysed in radioimmunoprecipitation assay (RIPA) buffer (10 mM Tris-Cl (pH 8.0), 1 mM EDTA, 1% Triton X-100, 0.1% sodium deoxycholate, 0.1% SDS, 140 mM NaCl, 1 mM phenylmethylsulfonyl fluoride) supplemented with protease/phosphatase inhibitors (1 mM orthovandate, 50 mM β-glycerophosphate, 10 mM sodium pyrophosphate, 5 μg/mL leupeptin, 2 μg/mL aprotinin, 5 mM sodium fluoride). The suspensions were centrifuged (17,000 × g, 10 min at 4 °C), and the protein content was determined using Pierce

BCA protein assay kit (Thermo Fisher Scientific Incorporation, Waltham, MA). Equal amounts of cellular lysate protein were mixed with Laemmli sample buffer and separated by SDS-PAGE. Western blotting was then performed after electrophoretic transfer of proteins from sodium dodecyl sulfate-polyacrylamide gels to nitrocellulose membranes and using anti-phospho-Akt, anti-Akt, anti-cleaved caspase-3 or anti-β-actin antibody (Cell Signaling, Danvers, MA)[67]. The intensity of the bands was quantified by densitometry using the public domain NIH imageJ program (wayne@codon.nih.gov; Wayne Rasband, NIH, Research Service Branch).

**Immunohistochemistry.** Staining of terminal deoxynucleotidyl transferase dUTP Nick-End Labeling (TUNEL) was performed at the Biopathology Institute Corporation (Kunisaki, Oita, Japan) by using Alexa Fluor 594 goat anti-rabbit IgG and slow-fade gold-antifade reagent with 4′,6-diamidino-2-phenylindole (DAPI) or by using ApopTag terminal deoxynucleotidyl transferase (Merck Millipore, Burlington, MA), anti-digoxigenin-peroxidase and 3,3′-diaminobenzidine. Quantification of apoptotic areas was performed using the WinROOF software (Mitani Corporation, Tokyo, Japan) and the values were averaged for each individual mouse.

**Evaluation of gene expression.** We extracted total RNA from cells or lung tissue using Sepasol RNA-I Super G reagent (Nacalai Tesque Inc., Kyoto, Japan), synthesized cDNA from 2 μg of total RNA with oligo-dT primer and ReverTra Ace Reverse Transcriptase (Toyobo Life Science Department, Osaka, Japan) and then performed standard PCR using primers described in Supplementary Table 5. PCR was performed with 26 to 35 cycles depending on the gene, denaturation at 94 °C for 30 s, annealing at 65 °C for 30 s, elongation at 72 °C for 1 min followed by a further extension at 72 °C for 5 min[67]. The expression of mRNA was normalized against the glyceraldehyde 3-phosphate dehydrogenase (GAPDH) mRNA expression.

**Transmission electron microscopy of apoptotic cells.** A549 cells ($10 \times 10^4$ cells/ml) were plated on a collagen-coated 8-well chamber slides (BD Bioscience, San Jose, CA) and cultured until semi-confluent. Cells were serum-starved for 6 h and stimulated with the pro-apoptotic peptide (5 μM) for 16 h. Cells were fixed with 2% fresh formaldehyde and 2.5% glutaraldehyde in 0.1 M sodium cacodylate buffer (pH 7.4) for 2 h at room temperature. After washing with 0.1 M cacodylate buffer (pH 7.4), they were postfixed with 1% $OsO_4$ in the same buffer for 2 h at 4 °C. The samples were rinsed with distilled water, stained with 1% aqueous uranyl acetate for 2 h or overnight at room temperature, dehydrated with ethanol and propylene oxide, and embedded in epon (Epon 812 resin, Nakalai). After removal of the cells from the glass, ultra-thin sections (94 nm) were cut, stained with uranyl acetate and Reynolds's lead citrate, and viewed with a transmission electron microscope (JEM-1010, JEOL, Tokyo, Japan).

**Cell cycle analysis and cell viability assay.** We performed DNA content/cell cycle analysis by flow cytometry after culturing the cells for 48 h in the presence or absence of the bacterial supernatant fraction. Cell cycle distribution was evaluated after treating the cells with propidium iodide. Cell viability was performed using a commercial cell counting kit (Dojindo, Tokyo, Japan). The samples used in the assays were fractionated after gel filtration using a Sephadex G25 column.

**Expression of S. nepalensis IsaA transglycosylases.** The genes encoding Staphylococcus nepalensis strain CNDG transglycosylase 351 and transglycosylase 531 were synthesized with E. coli optimized codons, amplified to add terminal A and cloned into the TA-cloning vector pGEM-T Easy (Promega, Madison, WI). The genes were then excised and cloned into a modified pET28a vector and transformed into E. coli BL21 DE3 cells and expressed and purified as 6-Histidine tagged (His-tag) proteins[86].

**Preparation of antibody against the pro-apoptotic peptide.** Protein A purified rabbit polyclonal antibody against the pro-apoptotic peptide (corisin) was developed by Eurofins Genomics (Tokyo, Japan) using the sequence NH2-C+IVM-PESSGNPNAVNPAGYR-COOH. A band at the corresponding molecular weight for the target peptide can be observed in Western blotting of mouse lung tissue samples and culture supernatant of Staphylococcus nepalensis strain CNDG (Supplementary Fig. 14a, b).

**Corisin detection and measurement in tissue and body fluids.** The purified anti-corisin IgG antibody was used at 1/1000 dilution for western blotting in lung tissue. We measured the concentration of corisin in body fluids using a competitive enzyme immune assay. Briefly, the purified corisin from transglycosylase 351 was coated on a 96-well plate at a final concentration of 2 μg/ml in phosphate-buffered saline at 4 °C overnight. After blocking and appropriate washing, the standards, samples and 5 ng/ml of anti-corisin were added to the wells and incubated at 4 °C overnight. The wells were then washed before adding horseradish peroxidase-conjugated goat anti-rabbit IgG (R&D System), as the secondary antibody, in a phosphate-buffered saline solution containing 5 μg/mL human IgG. After appropriate washing and incubation, substrate solution was added for color development

and absorbance read at 450 nm. Values were extrapolated from a standard curve prepared using several concentrations of the peptide.

**Phylogenetic analysis.** The five transglycosylase polypeptides (CNDG_8p_00351, CNDG_8p_00513, CNDG_8p_00157, CNDG_8p_00159, and CNDG_8p_00845) were used to search the Genbank protein database (https://www.ncbi.nlm.nih.gov/protein/) to retrieve homologous proteins. The protein sequences were aligned with the MUltiple Sequence Comparison by Log-Expectation (MUSCLE) program and the alignment was used in generating a phylogenetic tree based on the neighbor joining method with bootstrap value of 1000 replicates. All of these programs are available in Geneious Prime 2016 version (https://www.geneious.com/).

**Statistical analysis.** Data are described as the mean ± standard deviation of the means (S.D.) unless otherwise specified. The statistical difference between two variables was assessed by Mann–Whitney $U$ test and the difference between three or more variables by analysis of variance using Tukey's test for post-hoc analysis. $P$ value < 0.05 was considered statistically significant. We performed the statistical analysis using GraphPad Prism vs 7 (GraphPad Software, Inc., San Diego, CA).

**Reporting summary.** Further information on research design is available in the Nature Research Reporting Summary linked to this article.

## Data availability

The authors declare that all data supporting the findings in this study are available within this manuscript and its supplementary files. The whole genome sequences of the cultures designated strain 6 and strain 8 have been deposited at the Genbank database with the accession number PRJNA544423. The raw mass spectral data have been deposited to the ProteomeXchange Consortium via the PRIDE partner repository (http://www.ebi.ac.uk/pride) with the dataset identifier PXD017818. All datasets generated during and/or analyzed during the current study are also available from the corresponding authors on reasonable request.

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

## Acknowledgements

This research was supported in part by the Ministry of Education, Culture, Sports, Science, and Technology of Japan (Kakenhi No 17K08442 and 18K08175), and in part by funding support for the Microbiome Metabolic Engineering Theme (Carl R. Woose Institute for Genomic Biology) and by the College of Agricultural, Consumer and Environmental Sciences Office of International Programs, University of Illinois at Urbana-Champaign. The funders had no role in study design, data analysis, decision to publish, or preparation of the manuscript. We would like to thank Lou Ann Miller (Frederick Seitz Material Research Lab, UIUC) and Miyuki Ieda (Mie University School of Medicine) for technical support.

## Author contributions

C.N.D.G: preparation of disease model and preparation of first manuscript draft. S.W. and A.M.: evaluation by transmission electron microscope. A.Ta., K.N., and Y.Y.: analysis of signal pathways and protein expression. M.F., J.O., and T.N.: preparation of recombinant proteins. T.K., H.F., O.H., T.O., A.To., K.F., Y.N., Y.K, and K.K.: clinical evaluation of subjects, provision of clinical data and samples. M.T., T.Y., V.F.D., and Y.O.: flow cytometry analyses, gene, and protein expression analysis. J.W. and H.Ka.: chromatography analysis. H.Ki., A.M.A.H., Y.R., G.V.P., and R.I.M.: bacterial culture, genome analysis. C.L.W., A.H., C.J.F., and P.M.Y.: gene sequencing and mass spectrometry analysis. I.C. and E.C.G.: study design, analysis and interpretation of the data, and preparation and correction of the manuscript.

## Competing interests

C.N.D.G: and E.C.G. have a patent on the TGFβ1 TG mice used in the present study. There is an invention disclosure by C.N.D.G:, E.C.G., and I.C. on the apoptotic peptides identified in this study. None of the other authors declared any conflict of interest regarding the present work.
