## [Peer Review File · Nature Communications]

Reviewers' comments:

Reviewer #1 (Remarks to the Author):

Please note that I have focused my review on the aspects mentioned here and not on the biology of IPF.

Summary: The authors isolate and characterize a short peptide, which they name corisin, from cultures of halophilic strains of *Staphylococcus nepalensis* isolated from fibrotic lung tissue in mice. The authors perform a series of extractions on the supernatant of this *S. nepalensis* culture to isolate corisin, which they identify using LC-MS/MS and confirmed that it possessed apoptotic activity. The authors then show that a synthesized form of corisin also contains this apoptotic activity on epithelial cells in culture and worsens fibrosis indicators within a mouse model. They also show that antibodies against corisin are reactive to lung tissue of TGF β 1 KO mice as well as the tissue of human patients with idiopathic fibrosis. Finally, the authors performed genomic search to show that this corisin peptide is located within a transglycosylase gene that is conserved in several *Staphylococcus* species as well as other lung pathogens.

General Remarks: Overall, I find this to be a very thorough paper that should be of interest to scientists and physicians work to treat idiopathic fibrosis, and perhaps a more general audience. The authors have been very systematic in their work to draw a link between *S. nepalensis* in the lungs of mice with fibrosis to corisin and from corisin to fibrotic symptoms in mouse models. The attempts to generalize the work using phylogenetic comparisons and the ties to the transglycosylase are somewhat less notable, but still useful. There are a few areas that I suggest clarifying in order to make the work understandable to non-experts and increase enthusiasm for the paper. I also suggest a few of additional experiments to increase the impact of the work and more clearly link human fibrosis to the corisin peptide.

Major Comments:

- The authors use TGF β 1 KO mice as models for fibrosis throughout the work, but they also have controls of TGF β 1 KO mice without fibrosis, so there must be some other determining factor that causes fibrosis in these mice besides the KO. The authors should clarify early on why TGF β 1 KO mice are a proper model for fibrosis and how fibrosis is induced or not induced in these mice. Along these lines, it seems that a control of TGF β 1 KO mice without fibrosis should be included in Extended data Fig. 18b.

- One of the best ways to increase the impact of this work would be to show that a *S. nepalensis* strain that contains corisin in its genome can be isolated from IPF patient tissues, currently used in Figure 4. This would more clearly show the role that the microbe plays in the human, not just the mouse, form of the disease.

- Because the identification of corisin by LC-MS in the fractions showing apoptotic activity is so important to the work, the authors should show the MS and MS/MS data that led to this identification with full annotation and e values. It would also be helpful to note what other peptides were found in this fraction (raw data wouldn't be needed for these, only peptide tags and protein identifications with e values).

- Because the authors were able to isolate relatively pure amounts of corisin, it would be helpful to compare the apoptotic activity per μg of the synthesized corisin and the protein fraction isolated from cells. If the apoptotic activity can be entirely placed on corisin, then the synthesized corisin should be more potent than the material isolated from cells. This data should be added to Extended Fig. 13a,b.

- The authors raise antibodies against corisin. Can the authors show by WB that this reacts with the correct MW in the SUP of *S. nepalensis* bacterial cultures? The whole blot should be shown to demonstrate specificity.

- In extended figure 6 b, the authors show that after heating the SUP of the bacterial cultures can still induce Annexin V+ cells, in fact this response appears to be greater than before heating. The authors should show an unheated control under those same conditions.

Minor Comments:

- The authors use Annexin V to show apoptotic activity of cell supernatants, etc. The authors should clarify why this marker was used and what it is measuring.

- The annotations for Extended data Fig. 7 b on the x-axis are unclear. Does -LMW indicate without LMW? If so, then why does the third column have such a strong response? Also, how does one "add" more HMW (as in +HMW). This may just be a problem with the annotations, but the authors should make this more clear.

- Since Staph are known to be contaminant with IPF, can the authors comment on if antibiotics are ever used to treat IPF and if they have been shown to be effective?

The authors mention that the corisin peptides are housed within transglycosylases. Can the authors comment on if activity has ever been observed from these transglycosylases and if they might play any role in pathogenesis? Or do the authors believe that the enzymatic function of these transglycosylases have been lost and the corisin peptide is the important part.

- Similarly, the authors show that the peptide must be cleaved to be active. Can the authors hypothesize how this cleavage occurs?

- The authors show that the peptide is more potent with higher Na concentrations, can they comment on why this might be?

- In Extended data Fig. 18 b, the authors show increases in macrophages and lymphocytes, but the error bars are pretty large. Can they show if these changes are significant?

- It is unclear how the authors determine the direction of horizontal gene transfer between species from extended figures 20-22. Can the authors clarify how they formed these hypotheses?

Reviewer #2 (Remarks to the Author):

Peer review of the manuscript with the number NCOMMS-19-17899-T at Nature Communications by D'Alessandro-Gabazza et al and with the title: "A pro-apoptotic peptide conserved in diverse staphylococci induces progression of lung fibrosis"

The authors provide an interesting manuscript with data supporting the hypothesis that the lung microbiome is directly involved in at least one of the pathogenic events during Idiopathic pulmonary fibrosis (IPF), an interstitial disease that possesses high rates of global morbidity and mortality. In a salty lung microenvironment, a pro-apoptotic peptide released by the haloduric bacteria *Staphylococcus nepalensis*, upon trans-glycosylase processing, directly targets alveolar epithelial cells. The secreted peptide, named corisin, was biochemically characterized by chromatographic methods and proved to induce local inflammation and disease progression by functional assays. The

authors validated their in vitro data using a mouse model that overexpresses human TGF β 1 and samples from IPF patients. This manuscript aims to integrate physiological impairments in the alveolar microenvironment (high salt concentrations), with the opportunistic infection by *Staphylococcus* and their further abnormal activation of repair mechanisms (epithelial injury and inflammation). Further, the isolation and characterization of the secreted peptide, which is heat-stable and conserved in its primary amino acid sequence among the trans-glycosylases of different bacterial strains, are indeed interesting and novel. Moreover, the authors succeeded the artificial synthesis of corisin by 2 different companies and validated their effects experimentally, as well as the production of specific antibodies to antagonize its effects. The clinical relevance of the manuscript is reflected in the data that support a therapeutic approach against IPF based on the manipulation of the direct effects of corisin or its release. Even though these are all strengths of the manuscripts, the current version of the manuscript is not suitable for publication at Nature Communications. There are various major concerns that have to be addressed in order to improve the quality of the manuscript, thereby achieving the standards for publications at Nature Communications.

Major concerns:

1. The general structure of the manuscript has to be improved. The current version of the manuscript with 4 main figures and 24 extended data figures is not optimal. The number of main figures has to increase, for example by moving data from the extended data figures to the main figures. In addition, the manuscript's results could better described and necessarily panel wise introduced, since the interpretation of results is misleading by adding after many sentences all the panels together. More important, the description of the data in the results section has to be more accurate, for example describing the statistical relevance of the data presented.

In the following points I will list corrections and changes to the figures that should be considered by the authors in order to increase the quality of the manuscript.

1.1 Extended data Fig. 2 could be Figure 1. A 2 fold Na⁺ increase induces 8 fold Ct score and this is not explained or discussed at all. Additional confirmation of these results are compulsory. For example, a detailed examination of an extracellular or intracellular changes on Na⁺ concentration. To investigate the cellular lineages involves, the use of inflammatory (iNOS, RNS/ROS), macrophage M1/M2 (CD11b⁺, CD14^{low}, CD16⁺), T cells (CD4⁺) or epithelial (sodium channel ENaC) markers altered by Na⁺ will be very informative. Moreover, a potential effect of this microenvironment should be an increase in angiogenesis, which could be assessed by immunofluorescence methods with VEGF, Akt/PI3k, known to be similarly upregulated in IPF.

1.2 In the same Extended Fig. 2 is referred that "increased TGF β 1 may explain the findings...". Nevertheless, there are some animals TGF β 1 without increase on the CT score (TGF β 1 TG, fibrosis -).

How do the authors then justify this sentence? Did the authors perform genotyping on their transgenic animals that correlated with the Ct and Ashcroft scores?

1.3 Extended data Fig. 3 should be Fig. 2 together with Fig. 1. TGFB1 TG, fibrosis (-) animals were not used for further experiments. If there is a rational explanation for their phenotype, they could have been the appropriate negative control for all further experiments using in vivo approaches. It is not specified neither clear, how the authors confirmed the purity of strain 8, what will be its homology to ID: 60894 in NCBI? If the strain was not absolutely pure, a better approach should have been to obtain commercial strains (BacDive ID 14652). Scale bars are missing in Fig. 1.

1.4 Did the authors perform the experiments on Fig. 1, panels d, e, f and g also on strain 6?

1.5 Extended data Fig. 4 and extended data Fig 5 could be together Fig. 3. In extended data Fig. 4, authors should add inside the graphs the percentages of cells in apoptosis, since the peak sub-G1 is not prominent.

1.6 The Western Blot results on extended data Fig. 4f are not intuitive, might authors have a better exposure time for this membrane.

1.7 Extended data Fig. 7 could be Fig 4.

1.8 The results in Extended data Fig. 9 should be verified by immunofluorescent detection of apoptotic markers.

1.9 In Extended data Fig. 11 are the authors presenting Fraction 3, it is not indicated on the panel neither on the figure legend.

1.10 Extended data Fig. 12 could be Fig. 5 together with Fig 2a,b,c.

1.11 In Extended data Fig. 14 it is not written the company source of corisin. The use of scramble and the effects on caspase cleavage and Akt activation should also be performed for Fig. 2.

1.12 In Fig. 2. validation for the specificity of the anti-corisin antibody is missing.

1.13 In Extended data Fig.16 the labels for the panels are not complete.

1.14 Fig. 2d,e,f,g,h should be a supplementary and not a main figure.

1.15 Fig. 3 could be Fig. 6.

1.16 Extended data Fig. 18 could be Fig. 7 together with Fig. 4. The corisin antibody is not convincing. Didn't it work for BALF samples or why the results were not included in Fig. 4?

1.17 In Extended data Fig. 20 the strains *S. xylosus* and *S. cohnii* are included for their comparison with *S. nepalis*, but not other strains that have been reported in IPF and are more relevant. The same applies to the strains of *Streptococcus* included on the analysis. The presented sequence alignment should be presented in the context of reported literature related to the endogenous processing, isoforms and crystal structure of trans-glycosylases (PMID: 17675373, 22493270).

1.18 Panel 23b has a mistake on the word "scramble".

1.19 Datasets for the mass spectrometry approach using HPLC should be publicly available or additional tables should be added in the supplementary material.

2. The manuscript focus into the alveolar cells response to corisin, without considering the effect that it might induce in matrix-producing fibroblasts and immune cells present and regulated by the same micro-environment.

3. Importantly, the manuscript lacks a substantial discussion of the main findings and their correlation with published reports. Indeed, a paper published recently (Tong et. al., PMID: 31165050) establishing molecular signatures in BALF samples from IPF patients cannot be ignored.

4. The baseline characteristics of patients are included neither in Materials and Methods, nor in Results section, giving weakness to the study design. It would be useful to include information about

patients' age, gender, smoking history, diagnose, co-morbidities (if existing) and treatment (if existing). Moreover, a crucial aspect would be to know their ethnical background. Authors should mention if the IPF patients have matched controls.

5. The genetic TGFB mouse model renders low efficiency to show a fibrotic phenotype, according to results presented in the corresponding Figures. With these preliminary observations, the authors could have used a more robust IPF in vivo model such the bleomycin mouse or an ex vivo model of IPF precision cut lung slices or ex vivo-induced fibrosis by growth factors cocktails (PMID: 31110176).

6. Some aspects of the final model in Extended data Fig 24 generated with the description of results are missing credibility. There is no experimental evidence for the gene transfer from Staphylococcus to Streptococcus. The authors do not explore a single aspect of myofibroblast or endothelial activation, among other general assumptions. There is a mistake on the word "endothelial".

7. The observations of a salty environment in an IPF animal model are not fully discussed in the context of other pulmonary diseases. Even if cystic fibrosis is known to display the same altered electrolyte course, for lung cancer it will be an innovative finding. The manuscript includes a numerous amount of panels with A549 cells, a model of lung adenocarcinoma. However, these cells are not representative of injured ATII cells during IPF. Rather they will display some metabolic effects (Warburg effect) inside a saline micro-environment. A good justification of selection of this cellular model is required or, in a better case, to substitute those panels exclusively with the healthy bronchial epithelial cell line, which shows consistent results with A549.

8. The authors obviated or ignored previous contributions from other groups where the specific bacterial strains for Streptococcus and Staphylococcus are included (PMID 30824326, 29486761, 28802277, 28157391). For example, in IPF Staphylococcus epidermidis and aureus, Streptococcus agalactiae, gallolyticus, pneumonie and gordonii have been reported in IPF patients. Intriguingly, this is the first report with Staphylococcus nepalensis and that makes even more relevant to include and discuss the ethnical background of the selected IPF and control patients. Their introduction statement that "specific bacterial strain...remain unknown" is only true for reference 7, where the authors use operational taxonomic units (OTU) rather than concrete strains.

The data presented in the manuscript will be a major contribution for the field of IPF. In addition, the manuscript has a strong translational potential suggesting therapeutic approached against IPF. I strongly believe that after addressing all the concerns, the manuscript will improve significantly achieving the standards for publication at Nature Communications.

Reviewer #3 (Remarks to the Author):

This was a very difficult manuscript to read. My apologies if the authors have followed journal instructions. However, as a reasonably experienced reviewer, I expect that the rationale of a study will be set up in an introduction, to be followed by a methods section and then successively by a results section and then a discussion. In this manuscript, the introduction, results and discussion have been merged and interspersed at the start of the manuscript followed by a methods section. I do not understand why the authors did not produce a traditional introduction closing with specific study aims to provide structure to the presentation, even if they have followed journal instructions in merging results and discussion.

All this said, the key question is whether the conclusions are robust. Without the usual manuscript structure, I had to resort to dissecting the abstract, as follows:

“Idiopathic pulmonary fibrosis (IPF) is a chronic, progressive and fatal disease of unknown etiology. Injury, followed by apoptosis of lung epithelial cells, and increased accumulation of collagen-secreting myofibroblasts play a critical role in the pathogenesis of this intractable disease.”

This is generally sound although the authors have chosen to highlight only selective pathogenic considerations.

“Clinical progression of IPF is associated with increased abundance of the bacterial genera *Staphylococcus* and *Streptococcus* in the lung, although the specific bacterial strain and the mediating factor remain unknown.”

However, it is not known that this linkage is causative. It may well be that a milieu of activated pathways driving progression also provides a predilection for growth of particular organisms. There is a danger of conflating cause and effect.

“Here, we report that *Staphylococcus nepalensis* strain CNDG, isolated from lung fibrotic tissue, releases a unique peptide, we named corisin, buried in a polypeptide, to induce apoptosis of lung epithelial cells and therefore accelerating progression of pulmonary fibrosis.”

The idea is interesting but the statement implies that the authors have observed a specific linkage between corisin and disease progression as opposed to a non-specific short term irritation due to flooding the lungs with an alien polypeptide

“The pro apoptotic peptide was significantly increased in the lungs from IPF patients compared to healthy controls.”

Does this observation actually help? Epiphenomena of fibrosis and, separately, epiphenomena of fibrosis progression should be present in disease and should not be present in healthy controls. The

key comparison, surely, is between corisin content and disease progression in IPF patients. Without this comparison, the relevance to IPF progression is wholly uncertain.

“Evolutionary analyses revealed that the polypeptide embedded with the apoptosis-inducing peptide is conserved in diverse staphylococci, with known and unknown pathogenicity, with pathogenic strains of *Streptococcus pneumoniae* and *Mycobacterium abscessus* acquiring the gene likely through acquisition of genetic material in the lung.”

This explores pathways for polypeptide conservation but does not establish linkage to disease progression.

“Our results suggest that bacteria carrying and shedding this pro apoptotic peptide are involved in progression of pulmonary fibrosis and provide the molecular basis underlying the association of strains of *Staphylococcus* and *Streptococcus* in the progression of lung fibrosis.”

The key observation is the paragraph that follows, again merging introduction, results and discussion

“To investigate whether corisin can exacerbate the lung fibrotic disease in vivo, we separated TGF β 1 TG mice into two groups with matched level of lung fibrosis (Extended data Fig. 17a,b) and treated them with saline or corisin by intra-tracheal route once daily 12 for two days before euthanasia on day 3. TGF β 1 TG mice receiving corisin showed significantly increased infiltration of neutrophils, collagen deposition, concentration of inflammatory cytokines and chemokines, and apoptosis of epithelial cells in the lungs compared to control mice, indicating the detrimental effect of the pro-apoptotic activity of corisin in vivo (Fig. 3a,b,c,d,e,f,g). We then explored the presence of corisin in mice and human samples. We found significantly enhanced level of corisin in TGF β 1 TG mice compared to WT mice (Extended data Fig 18a,b), and significantly increased concentration of corisin in the bronchoalveolar lavage fluids from IPF patients compared to healthy controls, suggesting the potential implication of corisin in IPF (Figure 4). A dramatic increase of apoptotic epithelial cells occurs in the lung of IPF patients with acute exacerbation,17,18 and our results suggest that excessive release of the bacterial-derived pro-apoptotic corisin will contribute to this fatal disease complication.”

The authors have shown that instillation of a polypeptide, as opposed to saline, causes a short term injurious effect but the relevance of this to human disease progression is unclear. The model used by the authors – the acute introduction of a polypeptide by intratracheal installation – does not simulate chronic colonisation in human disease and nor is it clear that what may be a highly concentrated irritant effect captures the level of corisin action in human disease. I suggest that this experiment would have had more plausibility had corisin been compared to selected control polypeptides, to establish a corisin-specific effect, and had corisin concentrations in BAL been compared between progressive and non-progressive IPF patients. It is not clear how the comparison with normal controls helps.

Response to Queries of Reviewer 1

We thank the reviewer for her/his time and appreciate the comments which were very constructive and have helped to improve the interpretation and the quality of our manuscript.

Query 1:

The authors use TGF β 1 KO mice as models for fibrosis throughout the work, but they also have controls of TGF β 1 KO mice without fibrosis, so there must be some other determining factor that causes fibrosis in these mice besides the KO. The authors should clarify early on why TGF β 1 KO mice are a proper model for fibrosis and how fibrosis is induced or not induced in these mice. Along these lines, it seems that a control of TGF β 1 KO mice without fibrosis should be included in Extended data Fig. 18b.

Response

As suggested, we have explained why the TGF β 1 TG mice we used in the study is a proper model for lung fibrosis. Please see on **page 8, lines 153 to 160 in the revised manuscript. The text on page 8 is also described below:**

“TGF β 1 is considered the most important mediator of IPF, therefore here we used TG mice with lung fibrosis induced by lung overexpression of human TGF β 1 as previously reported (reference No 8,10, 11 and 12). Similar to the disease in humans, these TGF β 1 TG mice spontaneously develop pulmonary fibrosis characterized by a predominant and progressive scarring process, fatal outcome and typical lung histopathological findings (diffuse collagen deposition, honeycomb cysts, fibroblast foci-like areas)(reference No 8 and 11).”

We also explained why there are TG mice with or without fibrosis. The TG mice without lung fibrosis express the transgene but not the protein.

Please see on page 8, lines 159 to 160 in the revised manuscript. The portion is also described below:

“We used a line of TGF β 1 TG mice without fibrosis that express the human transgene but not the protein as controls (reference No 11 and 13).”

As suggested, we also added TGF β 1 TG mice without fibrosis as control group in Extended data Fig. 18b, which is now **Figure 8 (a,b) (Revised figure 8 is based on reviewer 2's suggestions)** in the revised manuscript.

Query 2:

One of the best ways to increase the impact of this work would be to show that a *S. napalensis* strain that contains corisin in its genome can be isolated from IPF patient tissues, currently used in Figure 4. This would more clearly show the role that the microbe plays in the human, not just the mouse, form of the disease.

Response

We agree with the Reviewer that isolation of bacteria from lung tissue of IPF patients would be the best way to increase the impact of this work. However, under the current International and Japanese Guidelines for management of these patients (Eur Respir J 2018;52:1801485, PMID:30810336; Lancet Respir Med 2018;6:138, PMID:29154106; Am J Respir Crit Care Med 2018;198:e44, PMID: 30168753; Am J Respir Crit Care Med 2019;200:1089, PMID: 31498684), it would be ethically difficult to perform lung tissue sampling for experimental purposes, due to the tendency of diagnostic procedures to exacerbate and accelerate the progression of the disease (Respir Med 2012;106:436, PMID22138357; Tohoku J Exp Med 1994; 174:379, PMID7732520; Lung 2012, :373, PMID22543997). Therefore, it would be ethically very difficult to perform the experiments knowing that a large size of tissue sample is necessary for isolation purposes.

The Ethical committee of our institution also follows the guidelines of these societies when its members consider research projects.

For these ethical concerns, in our current study we performed the initial experiment in mice with fibrosis and then used human BALF samples to demonstrate that the antigen is also present and relevant in IPF patients (**Figure 8c**).

Query 3:

Because the identification of corisin by LC-MS in the fractions showing apoptotic activity is so important to the work, the authors should show the MS and MS/MS data that led to this identification with full annotation and e values. It would also be helpful to note what other peptides were found in this fraction (raw data wouldn't be needed for these, only peptide tags and protein identifications with e values).

Response

As suggested, we show the MS and MS/MS data that led to this identification with full annotation and e values.

Please see **Supplementary Data set** (Mass spectrometry) uploaded as separate file and for the convenience of the Reviewer, we also described the data below:

```
MS/MS Fragmentation of IVMPESSGNPNAVNPAGYR  
Found in Staph_8p_00351 in putative transglycosylase IsaAMatch to Query 3631:  
1987.945068 from (994.979810,2+) intensity (711086.6200) scans (4233)  
rawscans (sn4233) rtinseconds (1268.962) index(216)  
Title: 217: Scan 4233 (rt=21.1494) [E:\Data\HFX\Isaac Cann\18-154-IsaacCann-  
Trypsindigested.raw]  
Data file E:\Data\HFX\Isaac Cann\18-154-IsaacCann-Trypsindigested.raw
```

Monoisotopic mass of neutral peptide Mr(calc): 1987.9422

Variable modifications:

M3 : Oxidation (M), with neutral losses 0.0000(shown in table), 63.9983

Ions Score: 80 **Expect:** 1.6e-07

Matches : 9/260 fragment ions using 9 most intense peaks

#	b	b ⁺⁺	b [*]	b ^{*++}	b ⁰	b ⁰⁺⁺	Seq.	y	y ⁺⁺	y [*]	y ^{*++}	y ⁰	y ⁰⁺⁺	#
1	114.0913	57.5493					I							19
2	213.1598	107.0835					V	1875.8654	938.4363	1858.8388	929.9231	1857.8548	929.4310	18
3	360.1952	180.6012					M	1776.7970	888.9021	1759.7704	880.3889	1758.7864	879.8968	17
4	457.2479	229.1276					P	1629.7616	815.3844	1612.7350	806.8712	1611.7510	806.3791	16
5	586.2905	293.6489			568.2799	284.6436	E	1532.7088	766.8580	1515.6823	758.3448	1514.6982	757.8528	15
6	673.3225	337.1649			655.3120	328.1596	S	1403.6662	702.3367	1386.6397	693.8235	1385.6557	693.3315	14
7	760.3546	380.6809			742.3440	371.6756	S	1316.6342	658.8207	1299.6076	650.3075	1298.6236	649.8154	13
8	817.3760	409.1917			799.3655	400.1864	G	1229.6022	615.3047	1212.5756	606.7914			12
9	931.4190	466.2131	914.3924	457.6998	913.4084	457.2078	N	1172.5807	586.7940	1155.5541	578.2807			11
10	1028.4717	514.7395	1011.4452	506.2262	1010.4612	505.7342	P	1058.5378	529.7725	1041.5112	521.2592			10
11	1142.5146	571.7610	1125.4881	563.2477	1124.5041	562.7557	N	961.4850	481.2461	944.4585	472.7329			9
12	1213.5518	607.2795	1196.5252	598.7662	1195.5412	598.2742	A	847.4421	424.2247	830.4155	415.7114			8
13	1312.6202	656.8137	1295.5936	648.3005	1294.6096	647.8084	V	776.4050	388.7061	759.3784	380.1928			7
14	1426.6631	713.8352	1409.6366	705.3219	1408.6525	704.8299	N	677.3365	339.1719	660.3100	330.6586			6
15	1523.7159	762.3616	1506.6893	753.8483	1505.7053	753.3563	P	563.2936	282.1504	546.2671	273.6372			5
16	1594.7530	797.8801	1577.7264	789.3669	1576.7424	788.8748	A	466.2409	233.6241	449.2143	225.1108			4
17	1651.7744	826.3909	1634.7479	817.8776	1633.7639	817.3856	G	395.2037	198.1055	378.1772	189.5922			3
18	1814.8378	907.9225	1797.8112	899.4093	1796.8272	898.9172	Y	338.1823	169.5948	321.1557	161.0815			2
19							R	175.1190	88.0631	158.0924	79.5498			1

Peptide tag	Peptide matches	Protein identification	e-values
1. S.IVMPRESSGNPNAVNPAGYR.G + Oxidation (M)	1	Putative transglycosylase IsaA	1.6 e-07
2. T.PNAMANLDVITKKFGASPK.S + Oxidation (M)	1	Sodium/glutamate symporter	0.0025
3. M.FVHFLGLPLP.G	1	Copper-exporting P-type ATPase A	0.014
4. K.LTPPPVK.Q	1	Methionyl-tRNA formyltransferase	0.00031
5. E.PYQSLSELQ.S	1	Spermidine N(1)acetyltransferase	0.0089

6. L.LTIKTYLGG.L	1	Sodium/pantothenate symporter	0.02
7. D.VVIKGE.R	1	Putative ABC transporter ATP-binding protein YbiT	0.0074
8. N.DISIDSKLKGQV.N	1	Hypothetical protein	0.047
9. F.MFAGKDLIVYDDLTK + Oxidation (M)	1	ATP Synthase subunit alpha	0.031
10. L.IAVVLSSAAVSVAGAL.G	3	Iron-uptake system permease protein FeuC	0.044
11. K.PMLVVAFAIIMANTISVLL + Oxidation (M)	1	Hypothetical protein	0.022
12. S.VPEDAKGQKVFME+Oxidation (M)	1	Vitamin B12-binding protein	0.019
13. I.FMMIIGALIGGVTNMIAVRMLFHPFKT.Y + 3 Oxidation (M)	1	Hypothetical protein	0.041
14. A.DKVAKALNKKKSGGAGEGSYTYDMEA.F + Oxidation (M)	1	Macrolide export ATP binding permease MacB	0.037
15. V.SLALPTIRDDLNVTASISLL.F	1	Multidrug efflux pump SdrM	0.04
16. T.EMCKANNVEIAVMI.R + 2 Oxidation (M)	1	CutC-like protein	0.018
17. A.ISGKLPANYADAL.P	1	Transketolase	0.029
18. K.LEKHPYKNPI.D	1	p-aminobenzyol-glutamate hydrolase subunit B	0.044
19. F.LPKSTEEKHSVARQLNVSVSELEHYIASLN.E	1	Pyruvate, phosphate dikinase	0.027
20. L.LNVTFNFDNLHTLPPH.F	2	Homoserine dehydrogenase	0.019
21. P.HHEQFVNTTEDIGHQLS.I	1	Putative competence-damage inducible protein	0.029
22. L.YTIDIDGIDPSIAPGTG.T	1	Guanidinobutyrase	0.033
23. E.MTIFEPIKGLIVNK.L + Oxidation (M)	1	Penicillin acylase	0.039
24. M.KVEIGKIINTHG.I	1	Ribosome maturation factor RimM	0.012
25. D.IDGLEVILLVNNNY.K	1	Putative ring cleaving dioxygenase MhqA	0.047
26. Y.VILSDYRGYN.R	1	Hypothetical protein	0.041
27. T.QPVKKGMKEKGVTEAMAKSAE.E + 2 Oxidation (M)	1	Dihyrolipoyl dehydrogenase	0.022
28. G.VLGALEVVEHLNEH.H	1	Putative hydrolase	0.022
29. V.AFILILIHIG.L	1	Hypothetical protein	0.048
30. I.VPVLGPITGGML.G	1	Glycerol uptake facilitator protein	0.038
31. G.SRPIEQHIK.G.F	1	UDP-N-acetylglucosamine 1-carboxyvinyltransferase 1	0.034
32. S.ELTSTLPHAQDYLLR.N	1	Type II secretion system protein F	0.044
33. F.EQSIGFLRIINGSEPLDNTSIH.P	1	30S ribosomal protein S1	0.036
34. H.SDHIKGLGVLARKYGLPI.Y	1	Putative metallo-hydrolase YycJ	0.05
35. R.TGIYMAIDSTNGYMDADRSEWIHD.E + Oxidation (M)	1	C protein alpha-antigen	0.023
36. F.GGYKHS.G.I	1	Putative aldehyde dehydrogenase	0.021
37. M.HVTISHP.L	1	Aspartokinase 2	0.031
38. L.LLVSSLLSQTAMA.A	1	Bifunctional autolysin	0.049
39. K.GILTTIPRKEIDIVAKVKAQYNIKKVTQNL.Y.R	1	Hypothetical protein	0.028
40. V.FIMASILTFA.S + Oxidation (M)	1	Ktr system potassium uptake protein B	0.043
41. N.ieikdep	1	Hypothetical protein	0.025
42. L.EAAEEVGNTSFQVFMKT.N + Oxidation (M)	1	Oxygen sensor histidine kinase NreB	0.029
43. S.YMKSMQN.T + Oxidation (M)	1	Putative glycosyltransferase TagX	0.038
44. N.EALKMKMGVDGG.F+Oxidation (M)	1	Glycerophosphodiester phosphodiesterase	0.046
45. F.ltdNikf.Q	1	Amino-acid carrier protein AlsT	0.046

Query 4:

Because the authors were able to isolate relatively pure amounts of corisin, it would be helpful to compare the apoptotic activity per ug of the synthesized corisin and the protein fraction isolated from cells. If the apoptotic activity can be entirely placed on corisin, then the synthesized corisin should be more potent than the material isolated from cells. This data should be added to Extended Fig. 13a,b.

Response

We have done the experiment suggested by the reviewer and the results of the experiment showed that the synthesized corisin is significantly more potent than the bacterial growth supernatant.

Please see the results in Supplementary Fig. 10 (previously Extended Fig. 13a,b), and page 16, lines 297 to 301 in the revised manuscript.

Query 5:

The authors raise antibodies against corisin. Can the authors show by WB that this reacts with the correct MW in the SUP of *S. nepalensis* bacterial cultures? The whole blot should be shown to demonstrate specificity.

Response

As recommended, we performed Western blotting of the supernatant of *Staphylococcus nepalensis* and showed the complete blot and detection of the antigen (corisin) by the anti-corisin antibody.

Please see Supplementary Fig.14, and the revised text on page 17, lines 314 to 316 in the revised manuscript.

Query 6:

In extended figure 6 b, the authors show that after heating the SUP of the bacterial cultures can still induce Annexin V+ cells, in fact this response appears to be greater than before heating. The authors should show an unheated control under those same conditions.

Response

We have done the suggested experiment and the results showed that heated supernatants are significantly more active than unheated controls. Heating likely denatures/precipitate heat-labile proteins in the supernatant and thus facilitate corisin interactions with its target. We hope to explore this observation in further detail in subsequent work.

Please see Supplementary Fig. 4a,b,c,d and the description of results on page 14, lines 253 to 257 in the revised manuscript.

Query 7:

The authors use Annexin V to show aptototic activity of cell supernatants, etc. The authors should clarify why this marker was used and what it is measuring.

Response

We added an explanation on why Annexin V was used to evaluate apoptosis in the present study.

Please see on pages 44 and 45, lines 804 to 810 in the revised manuscript. The explanation in the text is also described below for your convenience:

“Under physiological conditions, phosphatidylcholine is exposed externally while phosphatidylserine (PS) is located on the inner surface of the lipid bilayer of cellular membranes (reference No 83). During apoptosis, PS is translocated from the cytoplasmic face of the plasma membrane to the cell surface (reference No 83). Annexin V shows a strong affinity in binding to phosphatidylserine in a Ca²⁺-dependent manner and thus it is generally used as a probe for detecting apoptosis (reference No 84).”

Query 8:

The annotations for Extended data Fig. 7 b on the x-axis are unclear. Does -LMW indicate without LMW? If so, then why does the third column have such a strong response? Also, how does one “add” more HMW (as in +HMW). This may just be a problem with the annotations, but the authors should make this more clear.

Response

We are very sorry for the lack of clarity. We have amended the annotations of the labels and expanded the abbreviations to make them clearer.

Please see Figure 4 in the revised manuscript.

Query 9:

Since Staph are known to be contaminant with IPF, can the authors comment on if antibiotics are ever used to treat IPF and if they have been shown to be effective?

Response

As recommended by the reviewer we have mentioned the use and effectiveness of antibiotics in patients with IPF in the discussion section and cited previous works on this topic.

Please see page 27, lines 487 to 496 in the revised manuscript. The text is also described below:

“.....Recent data associating acute exacerbation with the lung microbiome and with the host immunosuppressive states, and retrospective studies showing the preventive effect of antibiotic therapy suggest the role of infection in the pathogenesis of acute exacerbation and progression of pulmonary fibrosis (references No 7, and reference No 42 to 45). Further, a double-blind, randomized, placebo-controlled study showing improvement of symptoms and exercise capacity in progressive IPF patients treated with co-trimoxazole, and a subsequent double-blind follow-up and multicenter study showing significant reduction of mortality with better quality of life and less respiratory tract infections in IPF patients treated with co-trimoxazole also support the pathogenic role of bacteria in lung fibrosis (references 46 to 47).....”

Query 10:

The authors mention that the corisin peptides are housed within transglycosylases. Can the authors comment on if activity has ever been observed from these transglycosylases and if they might play any role in pathogenesis? Or do the authors believe that the enzymatic function of these transglycosylases have been lost and the corisin peptide is the important part.

Response

We have briefly mentioned the multiple functions of transglycosylases described so far in the literature, and mentioned that, in our present study, only the corisin is apoptotic but not the full-length protein.

Please see the Discussion section on pages 29 and 30, lines 529 to 547, and the Supplementary Fig. 16, the description on page 18, lines 324 to 332 in the revised manuscript.

We provide further explanation below:

Corisin is housed in a transglycosylase belonging to the family known as Immunodominant Staphylococcal Antigen A or IsaA. In some staphylococci, such as *S. aureus*, these transglycosylases are expressed during infections and function as antigens, since high levels of antibodies against IsaA are detected in individuals that have experienced *S. aureus* infections (Infect Immun 2013, PMID:23208606; Immunol Med Microbiol 2000, PMID:11024354). They (IsaA) have been designated putative lytic transglycosylases as there is some evidence of peptidoglycan cleavage (J Bacteriol 2007, PMID:17675373). Lytic transglycosylases have been reported in other bacteria to carry out diverse functions, including cell-wall synthesis, cell wall remodeling and degradation and as virulence mechanism during infection in Gram negative bacteria; and in sporulation and germination in Gram positive spores (J Bacteriol 1975;124:1067, PMID:11024354; Crit Rev Biochem Mol Biol 2017, PMID:28644060). Furthermore, they have been reported to play a role in cell division, biofilm formation, and antibiotic resistance. They release peptidoglycan fragments that are thought to be recognized by host cells resulting in response of general symptoms of fever and also induction of inflammation through cytokine production (J Bacteriol 1985, PMID:3891732; J Bacteriol 1987, PMID:3301822; J Bacteriol 1993, PMID:8416911; Nat Rev Microbiol 2006, PMID:16894338; Antimicrob Agents Chemother 2019, PMID:31570396). In a recent publication (Antimicrob Agents Chemother 2019, PMID:31570396), it was shown in a strain of methicillin resistant *S. aureus* (MRSA) that IsaA was involved in biofilm formation. Deletion of the IsaA gene led to significant reduction in biofilm formation and in addition loss of beta-lactam resistance, suggesting that it could be a potential target for MRSA infections. This recent publication (Antimicrob Agents Chemother 2019, PMID:31570396) is one of the few exploring the function of the IsaA gene in *S. aureus*, in which its role remains largely unknown.

Although, several functions have been associated with homologs of IsaA and thus making them a family associated with diverse functions; it appears that their role in bacterial physiology are numerous and not completely understood. Our current report should open the door to the research community to further investigate this new and unique pathogenic function of a member of this family of proteins. Please, note that

there are variations in the peptide occupying the position of corisin in the IsaA member under investigation in the present report (**Supplementary Fig. 22**), and this may suggest diverse targets/functions of this group in the family of IsaA transglycosylases.

Query 11:

Similarly, the authors show that the peptide must be cleaved to be active. Can the authors hypothesize how this cleavage occurs?

Response

As suggested, we have presented our hypothesis on how the peptide (corisin) is cleaved from the full-length protein in the discussion section.

Please see page 30, lines 541 to 547 in the revised manuscript.

We are working diligently on the mechanism of cleavage, and we look forward to publishing in the future the mechanisms underlying the generation of this pro-apoptotic peptide, which should open new doors for addressing this condition.

Query 12:

The authors show that the peptide is more potent with higher Na concentrations, can they comment on why this might be?

Response

We speculate that the supernatant from bacteria cultured at high Na⁺ concentration is more significantly apoptotic than the supernatant from bacteria cultured at low Na⁺ concentration because of salt-dependent stimulation of bacteria growth or bacterial transglycosylase expression and peptide shedding. We provide this potential explanation in the discussion section.

Please see the Discussion section on page 28, lines 514 to 519 in the revised manuscript.

Query 13:

In Extended data Fig. 18 b, the authors show increases in macrophages and lymphocytes, but the error bars are pretty large. Can they show if these changes are significant?

Response

We re-assessed the statistical difference and found significant difference between the number of macrophages and lymphocytes. We added this explanation to the result section.

Please see Figure 6b, and the description on page 18, lines 338 to 343 in the revised manuscript.

Query 14:

It is unclear how the authors determine the direction of horizontal gene transfer between species from extended figures 20-22. Can the authors clarify how they formed these hypotheses?

Response

We have expanded the explanations on horizontal gene transfer between the different genera in the result section.

Please see on pages 22, 23 and 24, lines 404 to 440 in the revised manuscript.

In addition, we are also providing an expanded and detailed explanation below:

The gene coding for the *Staphylococcus nepalensis* IsaA-1 and its homologs are conserved in diverse members of the genus *Staphylococcus* (Please refer to **Supplementary Fig. 21** and **Supplementary Fig. 22** and analysis below). A search in the publicly available database (**Genbank Nucleotide and Protein Databases**) shows that this gene is not common to the genus *Streptococcus*. However, some organisms associated with the lung, including *Streptococcus pneumoniae* strain N and *Mycobacterium (Mycobacteroides) abscessus* subspecies *abscessus* strain 1000 appear to have acquired this gene. Again, please note that NOT ALL strains of either *Streptococcus pneumoniae* or *Mycobacterium abscessus* have this gene. Thus, the presence of the *Staphylococcus* type IsaA-1 is unique to the two strains of *Streptococcus pneumoniae* and *Mycobacterium abscessus*.

In the BLAST analysis below, using (**COE35810.1**), we show that if the protein for *Streptococcus pneumoniae* strain N is used to search the database, we do not see the conservation in other *Streptococcus*, but rather in the *Staphylococcus* and the highest homology is to a *Staphylococcus warneri* transglycosylase (**WP_050969398.1**). However, note that on line 5 of the hits (under accession) is another protein (**COE67256.1**) from *Streptococcus pneumoniae* strain N, suggesting there are two of such genes or proteins in this *Streptococcus pneumoniae* strain.

>**COE35810.1 transglycosylase protein [Streptococcus pneumoniae]**

```
MMKKTFIASLTALTLGATGYAVSGHEAHASETTNVDQAHLVDLAHNHPEQLNAAPVQEGAYDIHFVSGGF
EYNFTSDGTNFSWNVQEAGSTSAQTSNTAVQSADYTTSYNQEQAGTQSVSSNQSSNTNVEAVSAPTTSSNN
GSNHNYSKTKTTSYSAPSTSSASTGGSTKAQFLANGGTEEAWNAIVMPESGGNPNVNPAGYRGLGQTMES
WGTGSVASQTKGMLNYANSRYGSLSNIAIAFRQSHGWW
```

Select for downloading or viewing reports	Description	Max_Score	Total_Score	Query_CoverE	Value	Per_Ident	Accession
Select seq emb COE35810.1	transglycosylase protein [Streptococcus pneumoniae]	496	496	100%	2e-177	100.00%	COE35810.1
Select seq ref WP_050969398.1	transglycosylase [Staphylococcus warneri]	494	494	99%	2e-176	100.00%	WP_050969398.1
Select seq ref WP_015364674.1	MULTISPECIES: hypothetical protein [Staphylococcus]	492	492	99%	1e-175	99.59%	WP_015364674.1
Select seq ref WP_002467055.1	hypothetical protein [Staphylococcus warneri]	491	491	99%	5e-175	99.19%	WP_002467055.1
Select seq emb COE67256.1	transglycosylase protein [Streptococcus pneumoniae]	491	491	99%	5e-175	99.19%	COE67256.1
Select seq ref WP_126403073.1	transglycosylase [Staphylococcus warneri]	490	490	99%	8e-175	99.19%	WP_126403073.1
Select seq ref WP_031464076.1	transglycosylase [Staphylococcus warneri]	490	490	99%	9e-175	99.19%	WP_031464076.1
Select seq ref WP_002450152.1	hypothetical protein [Staphylococcus warneri]	489	489	99%	2e-174	98.78%	WP_002450152.1
Select seq ref WP_047211132.1	MULTISPECIES: transglycosylase [Staphylococcus]	488	488	99%	5e-174	98.78%	WP_047211132.1
Select seq tpg HBY83538.1	TPA: transglycosylase [Staphylococcus sp.]	485	485	99%	5e-173	97.97%	HBY83538.1
Select seq ref WP_107532308.1	transglycosylase [Staphylococcus warneri]	484	484	99%	1e-172	97.97%	WP_107532308.1
Select seq ref WP_023374149.1	MULTISPECIES: transglycosylase IsaA [Bacterial]	478	478	99%	7e-170	95.22%	WP_023374149.1
Select seq ref WP_107566252.1	transglycosylase [Staphylococcus warneri]	475	475	99%	9e-169	94.82%	WP_107566252.1
Select seq ref WP_117239240.1	transglycosylase [Staphylococcus pasteurii]	372	372	99%	4e-128	89.33%	WP_117239240.1
Select seq ref WP_108000303.1	transglycosylase [Staphylococcus pasteurii]	370	370	99%	2e-127	88.93%	WP_108000303.1
Select seq ref WP_017636593.1	MULTISPECIES: transglycosylase [Staphylococcus]	368	368	99%	2e-126	85.66%	WP_017636593.1

Select for downloading or viewing reports	Description	Max_Score	Total_Score	Query_Cover	E_value	Per_Ident	Accession
Select seq ref WP_070453111.1	transglycosylase [Staphylococcus sp. HMSC13A10]	368	368	99%	2e-126 85.66%		WP_070453111.1
Select seq ref WP_126567979.1	transglycosylase [Staphylococcus saccharolyticus]	367	367	99%	6e-126 84.50%		WP_126567979.1
Select seq gh QDWR83669.1	transglycosylase [Staphylococcus pasteurii]	367	367	99%	8e-126 85.27%		QDWR83669.1
Select seq ref WP_119623141.1	transglycosylase [Staphylococcus pasteurii]	364	364	99%	6e-125 84.88%		WP_119623141.1
Select seq ref WP_046466985.1	transglycosylase [Staphylococcus pasteurii]	363	363	99%	1e-124 84.50%		WP_046466985.1
Select seq ref WP_002444378.1	MULTISPECIES: hypothetical protein [Staphylococcus]	348	348	99%	9e-119 79.67%		WP_002444378.1
Select seq ref WP_070871703.1	transglycosylase [Staphylococcus sp. HMSC62A08]	321	321	99%	4e-108 79.67%		WP_070871703.1
Select seq ref WP_077700030.1	transglycosylase [Staphylococcus hominis]	310	310	99%	1e-103 67.60%		WP_077700030.1
Select seq ref WP_049432745.1	hypothetical protein [Staphylococcus hominis]	309	309	99%	2e-103 67.60%		WP_049432745.1
Select seq ref WP_002488225.1	MULTISPECIES: hypothetical protein [Staphylococcus]	308	308	99%	4e-103 67.20%		WP_002488225.1
Select seq ref WP_107622239.1	transglycosylase [Staphylococcus hominis]	308	308	99%	5e-103 67.20%		WP_107622239.1
Select seq ref WP_04934776.1	hypothetical protein [Staphylococcus hominis]	308	308	99%	5e-103 67.20%		WP_04934776.1
Select seq ref WP_087436311.1	MULTISPECIES: transglycosylase [Staphylococcus]	308	308	99%	5e-103 67.20%		WP_087436311.1
Select seq ref WP_145436998.1	transglycosylase [Staphylococcus hominis]	308	308	99%	6e-103 67.60%		WP_145436998.1
Select seq ref WP_054105900.1	MULTISPECIES: hypothetical protein [Staphylococcus]	308	308	99%	6e-103 67.20%		WP_054105900.1
Select seq ref WP_145453006.1	transglycosylase [Staphylococcus hominis]	307	307	99%	2e-102 66.80%		WP_145453006.1
Select seq ref WP_070715468.1	MULTISPECIES: transglycosylase [Staphylococcus]	306	306	99%	4e-102 66.80%		WP_070715468.1
Select seq ref WP_130547884.1	transglycosylase [Staphylococcus saccharolyticus]	305	305	99%	6e-102 69.51%		WP_130547884.1
Select seq ref WP_145421727.1	transglycosylase [Staphylococcus hominis]	305	305	99%	7e-102 66.80%		WP_145421727.1
Select seq ref WP_002449188.1	MULTISPECIES: hypothetical protein [Staphylococcus]	305	305	99%	9e-102 66.40%		WP_002449188.1
Select seq ref WP_142837310.1	transglycosylase [Staphylococcus hominis]						

Below is the protein from ***Streptococcus pneumoniae*** strain N (COE35810.1) and it's best hit in ***Staphylococcus warneri*** (WP_050969398.1).

>COE35810.1 transglycosylase protein [*Streptococcus pneumoniae*]
MMKKTFFIASTLALTLGATGYAVSGHEAHASET TNVDQAHLVDLAHNHPEQLNAAAPVQEGAYDIHFVSGGF
EYNFTSDGTNFSWNYQEAGSTSAQTSNTAVQSADYTTSYNQEAGTQSVSSNQSSNTNVEAVSAPTTSNNG
GSNHNYSTKTTTSYAPSTSSASTGGSTKAQFLANGGTEEAWNAI VMPESGGNPNAVNPAGYRGLGQTMES
WGTGVSASQTKGMLNYANSRYGSLSNIAIAFRQSHGWW

>WP_050969398.1 transglycosylase [*Staphylococcus warneri*]
MKKTFIASTLALTLGATGYAVSGHEAHASET TNVDQAHLVDLAHNHPEQLNAAAPVQEGAYDIHFVSGGFE
EYNFTSDGTNFSWNYQEAGSTSAQTSNTAVQSADYTTSYNQEAGTQSVSSNQSSNTNVEAVSAPTTSNNG
SNHNYSTKTTTSYAPSTSSASTGGSTKAQFLANGGTEEAWNAI VMPESGGNPNAVNPAGYRGLGQTMESW
GTGVSASQTKGMLNYANSRYGSLSNIAIAFRQSHGWW

In the alignment of the ***Streptococcus pneumoniae*** (COE35810.1 or Query 2) and ***Staphylococcus warneri*** (WP_050969398.1 or subject 1) transglycosylases shown below, it is observed that, other than the first methionine, the two proteins are a perfect match.

Query	2	MKKTFFIASTLALTLGATGYAVSGHEAHASET TNVDQAHLVDLAHNHPEQLNAAAPVQEGAY	61
		MKKTFFIASTLALTLGATGYAVSGHEAHASET TNVDQAHLVDLAHNHPEQLNAAAPVQEGAY	
Sbjct	1	MKKTFFIASTLALTLGATGYAVSGHEAHASET TNVDQAHLVDLAHNHPEQLNAAAPVQEGAY	60
Query	62	DIHFVSGGF EYNFTSDGTNFSWNYQEAGSTSAQTSNTAVQSADYTTSYNQEAGTQSVSSN	121
		DIHFVSGGF EYNFTSDGTNFSWNYQEAGSTSAQTSNTAVQSADYTTSYNQEAGTQSVSSN	
Sbjct	61	DIHFVSGGF EYNFTSDGTNFSWNYQEAGSTSAQTSNTAVQSADYTTSYNQEAGTQSVSSN	120
Query	122	QQSSNTNVEAVSAPTTSNNGSNHNYSTKTTTSYAPSTSSASTGGSTKAQFLANGGTEEAW	181
		QQSSNTNVEAVSAPTTSNNGSNHNYSTKTTTSYAPSTSSASTGGSTKAQFLANGGTEEAW	
Sbjct	121	QQSSNTNVEAVSAPTTSNNGSNHNYSTKTTTSYAPSTSSASTGGSTKAQFLANGGTEEAW	180
Query	182	NAI VMPESGGNPNAVNPAGYRGLGQTMESWGTGVSASQTKGMLNYANSRYGSLSNIAIAFR	241
		NAI VMPESGGNPNAVNPAGYRGLGQTMESWGTGVSASQTKGMLNYANSRYGSLSNIAIAFR	
Sbjct	181	NAI VMPESGGNPNAVNPAGYRGLGQTMESWGTGVSASQTKGMLNYANSRYGSLSNIAIAFR	240
Query	242	QSHGWW 247	
		QSHGWW	
Sbjct	241	QSHGWW 24	

Furthermore, the nucleotide sequences coding for the two transglycosylases, are also a match based on the alignment below. In other words, the two are the same gene.

Alignment of Sequence_1: [S.pneumoniae.xdna] with Sequence_2: [S.warneri.xdna]

```

Seq_1  1      ATGATGAAGAAGACATTTATCGCATCAACTTTAGCATTAAACATTAGGCGCAACAGGTTAC  60
          |||
Seq_2  1      --ATGAAGAAGACATTTATCGCATCAACTTTAGCATTAAACATTAGGCGCAACAGGTTAC  57

Seq_1  61      GCAGTATCAGGACACGAAGCACACGCTTCAGAACTACTAACGTAGATCAAGCACACTTA  120
          |||
Seq_2  58      GCAGTATCAGGACACGAAGCACACGCTTCAGAACTACTAACGTAGATCAAGCACACTTA  117

Seq_1  121     GTAGACTTAGCTCATAACCACCCAGAACAATTAACGCTGCACCAGTTCAAGAAGGCGCT  180
          |||
Seq_2  118     GTAGACTTAGCTCATAACCACCCAGAACAATTAACGCTGCACCAGTTCAAGAAGGCGCT  177

Seq_1  181     TATGACATTCACTTTGTAAGTGGTGGATTCGAATATAACTTTACTTCAGATGGTACTAAC  240
          |||
Seq_2  178     TATGACATTCACTTTGTAAGTGGTGGATTCGAATATAACTTTACTTCAGATGGTACTAAC  237

Seq_1  241     TTCTCTTGGAACTACCAAGAAGCTGGTTCTACTTCAGCTCAAACATCAAACACTGCTGTT  300
          |||
Seq_2  238     TTCTCTTGGAACTACCAAGAAGCTGGTTCTACTTCAGCTCAAACATCAAACACTGCTGTT  297

Seq_1  301     CAATCAGCTGACTACACAACCTTCTTACAATCAAGAAGCTGGTACTCAATCAGTAAGCTCT  360
          |||
Seq_2  298     CAATCAGCTGACTACACAACCTTCTTACAATCAAGAAGCTGGTACTCAATCAGTAAGCTCT  357

Seq_1  361     AACCAACAATCAAGCAACACTAATGTAGAAGCTGTTTCAGCTCCTCAACTACATCAAACAAT  420
          |||
Seq_2  358     AACCAACAATCAAGCAACACTAATGTAGAAGCTGTTTCAGCTCCTCAACTACATCAAACAAT  417

Seq_1  421     GGTTCAAACCACAACACTACAGCACTAAAACAACCTCATACTCAGCACCATCAACTTCAAGT  480
          |||
Seq_2  418     GGTTCAAACCACAACACTACAGCACTAAAACAACCTCATACTCAGCACCATCAACTTCAAGT  477

Seq_1  481     GCTTCAACAGGTGGATCAACTAAAGCACAAATCTTAGCTAATGGTGGTACTGAAGAAGCT  540
          |||
Seq_2  478     GCTTCAACAGGTGGATCAACTAAAGCACAAATCTTAGCTAATGGTGGTACTGAAGAAGCT  537

Seq_1  541     TGGAACGCTATCGTTATGCCAGAATCAGGTGGTAACCCCTAACGCAGTAAACCCAGCTGGT  600
          |||
Seq_2  538     TGGAACGCTATCGTTATGCCAGAATCAGGTGGTAACCCCTAACGCAGTAAACCCAGCTGGT  597

Seq_1  601     TACAGAGGTTTAGGACAAACTATGGAATCATGGGGAACGGTTTCAGTAGCTAGCCAAACT  660
          |||
Seq_2  598     TACAGAGGTTTAGGACAAACTATGGAATCATGGGGAACGGTTTCAGTAGCTAGCCAAACT  657

```

```

Seq_1  661  AAAGGTATGCTTAACTATGCTAATAGCCGTTACGGTTCATTAAGCAATGCAATTGCTTTC  720
        |||
Seq_2  658  AAAGGTATGCTTAACTATGCTAATAGCCGTTACGGTTCATTAAGCAATGCAATTGCTTTC  717

Seq_1  721  CGTCAAAGCCACGGTTGGTGGTAG  744
        |||
Seq_2  718  CGTCAAAGCCACGGTTGGTGGTAG  741

```

Here we show the alignment of the two *IsaA-1* homologs (**COE67256.1** and **>COE35810.1**) in *Streptococcus pneumoniae* strain N

```

>COE67256.1 transglycosylase protein [Streptococcus pneumoniae]
MKKTFIASTLALTLGAAGYAVSGHEAHASETTNVDQAHLVDLAHHNPEQLNAAPVQEGAYDIHFVSGGFE
YNFTSDGTFNFSWNYQEVGSTSAQTSNTAVQSADYTTTSYNQEAGTQSVSSNQSSNTNVEAVSAPTTSNNG
SNHNYSTKTTTSYSAPSTSSASTGGSTKAQFLANGGTEEAWNNAIVMPESGGNPNAVNPAGYRGLGQTMESW
GTGSVASQTKGMLNYANSRYGSLSNIAIAFRQSHGWW

```

```

>COE35810.1 transglycosylase protein [Streptococcus pneumoniae]
MMKKTFFIASTLALTLGATGYAVSGHEAHASETTNVDQAHLVDLAHHNPEQLNAAPVQEGAYDIHFVSGGF
EYNFTSDGTFNFSWNYQEAGSTSAQTSNTAVQSADYTTTSYNQEAGTQSVSSNQSSNTNVEAVSAPTTSN
GSNHNYSTKTTTSYSAPSTSSASTGGSTKAQFLANGGTEEAWNNAIVMPESGGNPNAVNPAGYRGLGQTMES
WGTGSVASQTKGMLNYANSRYGSLSNIAIAFRQSHGWW

```

```

COE67256.1      -MKKTFIASTLALTLGAAGYAVSGHEAHASETTNVDQAHLVDLAHHNPEQLNAAPVQEGA
COE35810.1      MMKKTFFIASTLALTLGATGYAVSGHEAHASETTNVDQAHLVDLAHHNPEQLNAAPVQEGA
                  *****:*****

COE67256.1      YDIHFVSGGFEYNFTSDGTFNFSWNYQEVGSTSAQTSNTAVQSADYTTTSYNQEAGTQSVSS
COE35810.1      YDIHFVSGGFEYNFTSDGTFNFSWNYQEAGSTSAQTSNTAVQSADYTTTSYNQEAGTQSVSS
                  *****.*****

COE67256.1      NQSSNTNVEAVSAPTTSNNGSNHNYSTKTTTSYSAPSTSSASTGGSTKAQFLANGGTEEA
COE35810.1      NQSSNTNVEAVSAPTTSNNGSNHNYSTKTTTSYSAPSTSSASTGGSTKAQFLANGGTEEA
                  *****

COE67256.1      WNNAIVMPESGGNPNAVNPAGYRGLGQTMESWGTGSVASQTKGMLNYANSRYGSLSNIAIAF
COE35810.1      WNNAIVMPESGGNPNAVNPAGYRGLGQTMESWGTGSVASQTKGMLNYANSRYGSLSNIAIAF
                  *****

COE67256.1      RQSHGWW
COE35810.1      RQSHGWW
                  *****

```

The alignment (above) of the two *IsaA-1* found in *Streptococcus pneumoniae* strain N shows that they are almost identical, other than the two changes in amino acids

highlighted in red. We predict that this is a gene duplication that occurred in this bacterium after acquisition from a *Staphylococcus*. However, since this genome seems to be a genome that is not closed, it is very difficult to carry out further analysis.

We also found several of the IsaA transglycosylases (5 different genes or proteins) in *Mycobacterium (Mycobacteroides) abscessus subspecies abscessus* and using these proteins to search the database, the best hits were also mostly from the *Staphylococcus*, and this finding supports our hypothesis that these genes are common to the *Staphylococcus* and are being horizontally transferred to other organisms in the lung. Please, see the analyses of the *Mycobacteroides* proteins in the subsequent pages below.

We, therefore, decided to investigate where the genes in ***Streptococcus pneumoniae strain N*** might have originated from, and by analyzing the gene synteny of the region containing one of the IsaA encoding genes (**COE35810.1**), we discovered that the synteny matches one in ***Staphylococcus warneri*** (**Supplementary Fig. 23d**). The finding is in agreement with the analysis above that the *Streptococcus pneumoniae* proteins are a perfect match of IsaA homologs found in ***Staphylococcus warneri***.

To summarize, the IsaA (**COE35810.1**) of *Streptococcus pneumoniae* strain N is an exact match of the homolog in *Staphylococcus warneri*, and the synteny or gene arrangement is conserved in both organisms, although the two bacteria belong to different genera (family) of bacteria. Furthermore, all the analyses above and below point to these IsaA proteins as *Staphylococcus* proteins (and not a protein of the genus *Streptococcus*). Hence, we hypothesize that the IsaA genes found in *Streptococcus pneumoniae* strain N were horizontally transferred from a *Staphylococcus*, most likely *Staphylococcus warneri* or its relative.

The explanation below is in regards to the analysis of horizontal gene transfer in MYCOBACTERIUM ABCESSUS

The *Staphylococcus* type IsaA genes are conserved in strains of *Mycobacteroides* (used to be *Mycobacterium*) *abscessus* subsp. *abscessus* isolated from the lung (respiratory system). We have done searches with each of the *Mycobacterium abscessus* subspecies *abscessus* IsaA proteins, and they mostly hit the *Staphylococcus* proteins and also the *Streptococcus pneumoniae* strain N proteins.

Please, see below for analyses.

```
>SIH37943.1 Probable transglycosylase isaA precursor [Mycobacteroides  
abscessus subsp. abscessus]  
MKKSI FALMTMLSLGAASLETGQAHAAEEVSTSPSQHQYQYNQSHTSNLNASSSNTTTTSSSTSSRTQSVYQ  
RFLAAGGTEEMWEKIVLPESGGNPNASNGQYHGLGQTNQSWGYGSVETQTKSMIQYAKERYGSIGAAIRF  
RESNGWW
```

>SKR69498.1 Probable transglycosylase isaA precursor [*Mycobacteroides abscessus* subsp. *abscessus*]

MKKTFIASLTALTLGATGYAVSGHEAHASET TNVDQAHLVDLAHNP EQLNAAAPVQEGAYDIHFVSGGFE
 YNFTSDGNTFSWNYQEAGSTSAQTSNTAVQSADYTTSYNQEAGTQSVSSNQSSNTNVEAVSAPTTSNNG
 SNHNYSTKTTSSAPSTSSASTGGSTKAQFLANGGTEEAWNAINVMPESGGNPNAVNPAGYRGLGQTMESW
 GTGSVASQTKGMLNYANSRYGSLSNIAIAFRQSHGWW

>SKT99287.1 Probable transglycosylase isaA precursor [*Mycobacteroides abscessus* subsp. *abscessus*]

MKKTVIASSLAVTLGLTGALYALTNDSAHASEQT TNYSHLADLAQNNPSELNAHPVQAGAYDISFVKDGFK
 YNFTSNGNTWSWNYTYTGADTAQSTTDYTESYNQASTQSVSSNQASTSNVKAVSAPVQRTSSYNNYSA
 RTTSSAPKTTSSYSTASTGGSVKAQFLANGGTEEAWNAINVMPESGGNPASNGQYHGLGQTNQSWGTSV
 ASQTQGMANYAKSRYGSWDAIAFRNANGWW

>SKR88156.1 Probable transglycosylase isaA precursor [*Mycobacteroides abscessus* subsp. *abscessus*]

MKKSILAI IATISIGATGMEAHQAHAENNOSSQSYSESNESTSSVYQEFIDAGGTKALWDSIVIPESG
 GNPNASNGQYHGLGQTNQSWGYSVENQTKGMINYAKERYGSIDKAISFREANGYW

>SLB62866.1 Probable transglycosylase isaA precursor [*Mycobacteroides abscessus* subsp. *massiliense*]

MKKTVIASLTAVSLGIAGYGLSGHEAHASET TNVDKAHLVDLAQHNPEELNAKPVQAGAYDIHFVDNGYQ
 YNFTSNGSEWSWSYAVAGSDADYTESSSNQEVSAANTQSSNTNVQAVSAPTSSERSYSTSTTSSAPSHN
 YSSHSSSVRLSNGNTAGSVGSYAAAQMAARTGVSASTWEHI IARESNQQLHARNASGAAGLFQTMPGWGS
 TGSVNDQINAAAYKAYKAQGLSAWGM

A search with >SIH37943.1 Probable transglycosylase isaA precursor

[*Mycobacteroides abscessus* subsp. *abscessus*]

MKKSIFALMTMLSLGAASLETGQAHAEEVSTSPSQHQYQYNQSHTSNLNASSTNTTSSSTSSRTQSVYQ
 RFLAAGGTEEMWEKIVLPESGGNPASNGQYHGLGQTNQSWGYSVETQTKSMIQYAKERYGSIGAAIRF
 RESNGWW

RESULTS

Sequences producing significant alignments:

Select for downloading or viewing reports	Description	Max Score	Total Score	Query Cover	E value	Per. Ident	Accession
Select seq emb SIH37943.1	Probable transglycosylase isaA precursor [Mycobacteroides abscessus subsp. abscessus]	298	298	100%	4e-102100.00%		SIH37943.1
Select seq ref WP_070813180.1	MULTISPECIES: transglycosylase [Staphylococcus]	296	296	100%	2e-10199.32%		WP_070813180.1
Select seq ref WP_145416769.1	transglycosylase [Staphylococcus hominis]	296	296	100%	3e-10199.32%		WP_145416769.1
Select seq ref WP_061815426.1	hypothetical protein [Streptococcus pneumoniae]	296	296	100%	3e-10199.32%		WP_061815426.1
Select seq gb RLY83101.1	transglycosylase [Staphylococcus hominis]	293	293	100%	4e-10098.64%		RLY83101.1

Select for downloading or viewing reports	Description	Max Score	Total Score	Query Cover	E value	Per. Ident	Accession
Select seq ref WP_103213471.1	transglycosylase [Staphylococcus caprae]	358	358	100%	3e-123	78.51%	WP_103213471.1
Select seq ref WP_141041852.1	transglycosylase [Listeria monocytogenes]	347	347	74%	9e-120	100.00%	WP_141041852.1
Select seq ref WP_147700201.1	transglycosylase [Staphylococcus aureus]	341	341	72%	3e-117	100.00%	WP_147700201.1
Select seq gb PPJ68958.1	transglycosylase [Staphylococcus aureus]	333	333	73%	3e-114	98.27%	PPJ68958.1
Select seq gb PDG69856.1	transglycosylase [Listeria monocytogenes]	332	332	71%	1e-113	99.40%	PDG69856.1
Select seq ref WP_075778488.1	MULTISPECIES: transglycosylase SLT domain-containing protein [Staphylococcus]	294	294	100%	1e-97	66.02%	WP_075778488.1
Select seq ref WP_118828089.1	transglycosylase SLT domain-containing protein [Staphylococcus sp. M0911]	291	291	100%	2e-96	65.62%	WP_118828089.1
Select seq ref WP_049416533.1	MULTISPECIES: transglycosylase SLT domain-containing protein [Staphylococcus]	289	289	100%	1e-95	65.23%	WP_049416533.1
Select seq ref WP_119547599.1	transglycosylase SLT domain-containing protein [Staphylococcus warneri]	286	286	100%	1e-94	64.84%	WP_119547599.1
Select seq ref WP_084944604.1	transglycosylase SLT domain-containing protein [Staphylococcus lugdunensis]	282	282	100%	4e-93	69.55%	WP_084944604.1
Select seq ref WP_002460510.1	MULTISPECIES: transglycosylase SLT domain-containing protein [Staphylococcus]	282	282	100%	6e-93	69.55%	WP_002460510.1
Select seq ref WP_085425274.1	transglycosylase SLT domain-containing protein [Staphylococcus lugdunensis]	281	281	100%	2e-92	69.14%	WP_085425274.1
Select seq ref WP_111443705.1	transglycosylase SLT domain-containing protein [Burkholderia multivorans]	281	281	100%	2e-92	63.67%	WP_111443705.1

Response to Queries of Reviewer 2

We thank the reviewer for the time taken to review our manuscript and appreciate the comments and suggestions, which were very constructive and have helped to improve the interpretation and the quality of our manuscript.

Query 1:

The general structure of the manuscript has to be improved. The current version of the manuscript with 4 main figures and 24 extended data figures is not optimal. The number of main figures has to increase, for example by moving data from the extended data figures to the main figures. In addition, the manuscript's results could better described and necessarily panel wise introduced, since the interpretation of results is misleading by adding after many sentences all the panels together. More important, the description of the data in the results section has to be more accurate, for example describing the statistical relevance of the data presented.

In the following points I will list corrections and changes to the figures that should be considered by the authors in order to increase the quality of the manuscript.

Response

We have followed the constructive suggestions of the Reviewer and re-structured the text and the figures of the manuscript as an Article, added the sections of **Introduction**, **Results** and **Discussion** and provided more explanation for the results obtained in the experiments as described below.

Query 1-1:

Extended data Fig. 2 could be Figure 1. A 2 fold Na⁺ increase induces 8 fold Ct score and this is not explained or discussed at all. Additional confirmation of these results are compulsory. For example, a detailed examination of an extracellular or intracellular changes on Na⁺ concentration. To investigate the cellular lineages involves, the use of inflammatory (iNOS, RNS/ROS), macrophage M1/M2 (CD11b⁺, CD14^{low}, CD16⁺), T cells (CD4⁺) or epithelial (sodium channel ENaC) markers altered by Na⁺ will be very informative. Moreover, a potential effect of this microenvironment should be an increase in angiogenesis, which could be assessed by immunofluorescence methods with VEGF, Akt/PI3k, known to be similarly upregulated in IPF.

Response

As suggested, we allocated the previous Extended data Fig. 2 as **Figure 1** in the revised manuscript.

We added a detailed description of the results. Measurement of the concentration of Na⁺ in the lung tissue was performed several times and the results were reproducible.

As suggested, we have performed additional experiments to evaluate lung immune cells, expression of sodium channels, inflammatory and fibrotic markers, and angiogenic factors in the lungs from WT type mice and in TGFβ1 TG mice with or without fibrosis. We have also evaluated correlation of the lung expression of these tissue parameters with the lung tissue concentration of sodium. As described in the revised manuscript, there are an impaired immune response, decreased expression of sodium channels, high concentration of inflammatory, fibrotic and angiogenic parameters in the lung fibrotic tissue compared to non-fibrotic lung tissues. The concentration of sodium was also significantly correlated with the impaired immune response, fibrotic parameters and interestingly with the expression of sodium channels in fibrotic tissues.

Please see the results described in the new Supplementary Fig. 2a,b,c,d, Supplementary Table 1, Supplementary Table 2 and Supplementary Fig. 3, and the description of the results in the revised manuscript on pages 9, 10 and 11, lines 169 to 202 in the revised manuscript.

We also added comments under the discussion section to discuss the high concentration of sodium in the lung fibrotic tissue and its correlation with the abnormal immune response, and abnormal expression of sodium channels and inflammatory and fibrotic parameters in lung tissue.

Please see pages 25 and 26, lines 452 to 478 in the revised manuscript.

Query 1-2:

1.2 In the same Extended Fig. 2 is referred that “increased TGFB1 may explain the findings...”. Nevertheless, there are some animals TGFB1 without increase on the CT score (TGFB1 TG, fibrosis -). How do the authors then justify this sentence? Did the

authors perform genotyping on their transgenic animals that correlated with the Ct and Ashcroft scores?

Response

In the revised manuscript, we have dedicated a section to discuss the mechanism by which TGF β 1 would be involved in the increased level of Na⁺ in the lung fibrotic tissue based on studies reported before.

Please see pages 25 and 26, lines 455 to 478 in the revised manuscript.

As described above we have also evaluated fibrotic markers including the lung tissue expression and the plasma level of TGF β 1 and found that they are correlated with the lung tissue level of sodium.

Please see the results described in the new Supplementary Fig. 2a,b,c,d, Supplementary Table 2 and Supplementary Fig. 3, and the description of the results in the revised manuscript on pages 10 and 11, lines 181 to 202 in the revised manuscript.

Query 1-3:

1.3 Extended data Fig. 3 should be Fig. 2 together with Fig. 1. TGF β 1 TG, fibrosis (-) animals were not used for further experiments. If there is a rational explanation for their phenotype, they could have been the appropriate negative control for all further experiments using in vivo approaches. It is not specified neither clear, how the authors confirmed the purity of strain 8, what will be its homology to ID: 60894 in NCBI? If the strain was not absolutely pure, a better approach should have been to obtain commercial strains (BacDive ID 14652). Scale bars are missing in Fig. 1.

Response

As suggested by the Reviewer, we have allocated Extended data Fig. 3 as **Figure 2** together with Fig. 1 in the revised manuscript. Unfortunately, the number of mice born with the human TGF β 1 transgene positive but with no phenotype (lung fibrosis) is extremely scarce or rare and thus it was very difficult to include them in all experiments.

We have added some comment on the model in the revised manuscript on page 8, lines 153 to 160.

To answer the question of the Reviewer on the purity of strain 8, please let us explain how the process of microbial isolation was undertaken:

The microbes were isolated in the Laboratories of Isaac Cann and Roderick Mackie at the University of Illinois Urbana–Champaign. Both expert investigators routinely isolate microbes from different environments for characterization and publication (Int J Syst Evol Microbiol 2001, PMID:11321073; Int J Syst Evol Microbiol 2008, PMID:18319475; Int J Syst Evol Microbiol 2009, PMID:19542122). Therefore, the labs are equipped and

have experience for isolation of microbes in their pure culture. TGFβ1 mice lung tissues were inoculated into a *Halomonas* salt medium [NaCl 80 g/L, Casamino acids 7.5 g/L, Peptone 5.0 g/L; Yeast extract 1.0 g/L, sodium citrate 3.0 g/L, Mg.SO₄.7H₂O 20 g/L, K₂HPO₄ 0.5 g/L; Fe(NH₄)₂(SO₄).6H₂O] and cultured under anaerobic conditions for 2 days. Aliquots of the microbial growth observed from the 2-day cultures were streaked on agar plates of the salt medium and several single colonies were picked and regrown in the liquid medium. Microscopy, DNA extraction and amplification of the 16S rRNA gene and its nucleotide sequencing were used for identification of the different colonies. The colonies picked from the culture derived from mouse TGFβ1 #8 were confirmed as *Staphylococcus nepalensis*, based on the sequence of their 16S rRNA gene sequences. One colony, designated *S. nepalensis* strain CNDG in the present study, was selected and the genome was sequenced to confirm the identity of the bacterium. The genome sequence was easily closed into a single circular chromosome of size 2.86895 Mb (very similar to *S. nepalensis* strains SNUC 4025, DSM15150, and JS9 with genome sizes of 2.8631Mb, 2.86023 Mb and 2.89287 Mb, respectively) and two circular plasmids of sizes 30,614 nucleotides and 4,619 nucleotides, respectively. (Please refer to text on page ?????). In contrast, we could not close the genome of a colony from the cultures derived from mouse TGFβ1 #6. The assembly of the genome sequence from the colony derived from the mouse designated TGFβ1 #6 suggested two potential genomes. Therefore, this culture was designated a mixed culture.

Regarding the homology of strain #8 to ID: 60894 in NCBI. We have compared the genome of strain CNDG with that of other strains in the Genbank database (**accessible from ID: 60894**) and for *S. nepalensis* strains JS9, SNUC4337, DSM15150, JS11, and JS1 the identities were 99.52%, 99.61%, 99.60%, 99.53% and 99.50%, respectively, confirming the bacterium designated strain CNDG (strain #8) and used in the present study as a *Staphylococcus nepalensis*. The plasmid sizes of strain CNDG are also different from those described for other *S. nepalensis* strains (i.e., JS1, and JS11). Therefore, strain CNDG can be distinguished from other *S. nepalensis* strains (<https://www.ncbi.nlm.nih.gov/genome/genomes/60894/>), using its chromosome and plasmid sequence sizes. Based on the analysis above, we are very confident of the purity of the strain (strain CNDG) used in the present research. The strain is stored on agar slants (and also glycerol stock) in our laboratory and all samples tested so far have shown the properties reported in the present report. Furthermore, all other colonies isolated from this work are still stored in our lab, and strains will be made available upon request to any lab.

We added the scale bars to **Figure 1**, and described them in the figure legends.

Query 1-4:

Did the authors perform the experiments on Fig. 1, panels d, e, f and g also on strain 6?

Response

The pro-apoptotic effect of the culture supernatant from strain 6 was also confirmed by Western blotting of cleaved caspase.

The Western blotting and quantification are shown in **Supplementary Fig. 4c,d**.

Please also see the description in the text of the revised manuscript on **page 14, lines 253 to 257 in the revised manuscript**.

Query 1-5:

Extended data Fig. 4 and extended data Fig 5 could be together Fig. 3. In extended data Fig. 4, authors should add inside the graphs the percentages of cells in apoptosis, since the peak sub-G1 is not prominent.

Response

As suggested, we combined Extended data Fig. 4 and extended data Fig 5 and it is now **Figure 3** in the revised manuscript.

The percentage of apoptotic cells were added as recommended.

Query 1-6:

The Western Blot results on extended data Fig. 4f are not intuitive, might authors have a better exposure time for this membrane.

Response

We repeated the experiment and showed a better blot. The panel is now described in **Supplementary Fig. 4c,d**.

Query 1-7:

Extended data Fig. 7 could be Fig 4.

Response

As recommended, Extended data Fig. 7 is **Figure 4** in the revised manuscript.

Query 1-8:

The results in Extended data Fig. 9 should be verified by immunofluorescent detection of apoptotic markers.

Response

We verified the results presented in the original Extended data Fig. 9 by TUNEL staining.

Please see Supplementary Fig. 7 and its description in the revised manuscript on pages 14 and 15, lines 268 to 273.

Query 1-9:

In Extended data Fig. 11 are the authors presenting Fraction 3, it is not indicated on the panel neither on the figure legend.

Response

We sorry for the lack of clarification. We added a label to show that it corresponds to **fraction 3**.

Please see the figure, which is Supplementary Fig. 9 in the new revised manuscript.

Query 1-10:

Extended data Fig. 12 could be Fig. 5 together with Fig 2a,b,c.

Response

As suggested, we combined Extended data Fig. 12 with Fig 2a,b,c and allocated as **Figure 5** in the revised manuscript.

Query 1-11:

In Extended data Fig. 14 it is not written the company source of corisin. The use of scramble and the effects on caspase cleavage and Akt activation should also be performed for Fig. 2.

Response

The company source was clarified in the legend of the figure, which is now **Supplementary Fig. 12 in the revised manuscript**.

Scrambled peptide and the effects of caspase cleavage and Akt activation were also evaluated and they are described in **Supplementary Fig. 11 and Supplementary Fig. 12 in the revised manuscript**.

Query 1-12:

In Fig. 2. validation for the specificity of the anti-corisin antibody is missing.

Response

We have done a Western blotting to show the specificity of the antibody.

Please see Supplementary Fig. 14 and its description on page 17, lines 314 to 316 in the revised manuscript.

Query 1-13:

In Extended data Fig.16 the labels for the panels are not complete.

Response

We have revised the labels in the panels of the figure, which is now **Supplementary Fig. 15**.

Query 1-14:

Fig. 2d,e,f,g,h should be a supplementary and not a main figure.

Response

As suggested, we allocated the original Fig. 2d,e,f,g,h as **Supplementary Fig. 15a,b,c,d**.

Query 1-15:

Fig. 3 could be Fig. 6.

Response

As recommended, Fig. 3 is now **Figure 6** in the revised manuscript.

Query 1-16:

Extended data Fig. 18 could be Fig. 7 together with Fig. 4. The corisin antibody is not convincing. Didn't it work for BALF samples or why the results were not included in Fig. 4?

Response

As suggested, Extended data Fig. 18 was changed to Figure 7 together with Figure 4. To evaluate the specificity of the corisin antibody, Western blotting using the antibody was performed to evaluate the presence of corisin in the culture supernatant of *Staphylococcus nepalensis* and lung tissue homogenate from mice. A band corresponding to the molecular weight of corisin was detected in both culture supernatant and lung tissue. **The result is described in Supplementary Fig 14a,b and on the text on page 17, lines 314 to 316.**

The level of corisin was detected in BALF samples from the IPF patient as described in **Figure 8** in the revised manuscript. In mice, corisin was detected in lung tissue homogenate. The level in BALF samples from mice was not very high, probably because of the high dilution of the sample and/or because the corisin rapidly binds to the surface of lung epithelial cells, masking its binding site from the antibody.

Query 1-17:

In Extended data Fig. 20 the strains *S. xylosus* and *S. cohnii* are included for their comparison with *S. nepalis*, but not other strains that have been reported in IPF and are more relevant. The same applies to the strains of *Streptococcus* included on the analysis. The presented sequence alignment should be presented in the context of reported literature related to the endogenous processing, isoforms and crystal structure of trans-glycosylases (PMID: 17675373, 22493270).

Response

Please, also see response to query 10 of reviewer 1

As pointed out by the Reviewer, other strains of *Staphylococcus* including *Staphylococcus aureus* have been reported in IPF patients. To discuss this, the following paragraph in bracket was added to the discussion section of the revised manuscript. **Please see pages 30 and 31, lines 555 to 568 in the discussion section of the revised manuscript.**

[... While the present study is the initial report on the pathogenicity of a peptide derived from an IsaA homolog in a strain of *Staphylococcus*, homologous proteins (i.e., IsaA and SceD) have been reported in *Staphylococcus aureus* to be involved in virulence. The *Staphylococcus aureus* IsaA in that report (reference No 53) corresponds to YP_501340 in the alignment shown in **Supplementary Fig. 21**, while the SceD, in the same report, has a variant of corisin similar to those in the SceD-1 to SceD-4 (**Supplementary Fig. 21**). Thus, although relevant, the characterized transglycosylases in *Staphylococcus aureus* are quite different from the *Staphylococcus nepalensis* transglycosylase characterized in the present study. It is of note, however, that *Staphylococcus aureus* has an uncharacterized IsaA transglycosylase with a highly conserved corisin sequence (**Supplementary Fig. 21**, IsaA-2, SUK04795), which may suggest that a similar mechanism as the corisin processing described in the present study exists in *Staphylococcus aureus*....]

We have additional comments on this below:

Using a mouse model, Stapleton and co-workers (J Bacteriol 2007, PMID17675373) showed that *isaA* and *sceD* mutants were slightly attenuated while the double mutant (*isaA/sceD*) was significantly attenuated in pathogenicity compared to the wild-type. By also showing that the two transglycosylases, which are paralogous in the bacterium, cleaved *S. aureus* peptidoglycan, they inferred that altered cell wall structure may be important in antibiotic resistance. Interestingly, the authors also found that SceD is highly upregulated under high NaCl conditions. IsaA in *S. aureus* has also been reported to be antigenic (FEMS Immunol Med Microbiol 2000, PMID11024354) and with both SceD and IsaA being shown to be required for normal growth (J Bacteriol 2007, PMID 17675373), they have been suggested as potential targets for antibody-based therapy against methicillin resistant *Staphylococcus aureus* (MRSA). However, note that the corisin corresponding sequence in both the *Staphylococcus aureus* IsaA and SceD are different, and we have also shown in the present report that changes in the sequence abrogates its apoptotic activity on the lung cells investigated in the present study, suggesting that the *Staphylococcus aureus* IsaA and SceD may not target these cells or function by a different mechanism.

In regards to relating our work (alignment) to the published literature on the crystal structure of transglycosylases (Proc Natl Acad Sci U S A. 2012, PMID 22493270), we have observed that this published structure is of very low similarity, with many gaps in the alignment. **Please, see below for the alignment of *Staphylococcus nepalensis* strain CNDG protein 0351 containing corisin and the *Staphylococcus aureus* MGT polypeptide.** The alignment suggests that despite our protein and the crystallized protein

in PMID 22493270 being members of the transglycosylase family, the two are very different, and therefore preventing a meaningful structural interpretation. Using SWISS-Model program (<https://swissmodel.expasy.org/interactive>) to model the structure, the best results came from using a *Ralstonia* sp enzyme (3w6e.1.A or lysozyme-like chitinolytic enzyme) that has homology to a transglycosylase SLT-containing protein from *Paenibacillus* (WP_009225716.1). The homology of these enzymes to the *Staphylococcus nepalensis* protein are too low. Thus, we prefer not to present such data in the current manuscript. Importantly, we have the facilities to crystallize either our protein or a relative from the genus *Staphylococcus*, and we look forward to publishing our insights on the structure in the nearest future.

PMID:22493270 (S. aureus transglycosylase MGT) crystal structure publication >EHS15546.1 monofunctional glycosyltransferase [*Staphylococcus aureus* subsp. *aureus* IS-55]

```
MKILLTILIIIIALFIGIMYFLSTRDNVDELRKIENKSSFVSADNMPEYVKGAFISMEDERFYNNHHGFDLK
GTTRALFSTISDRDVQGGSTITQQVVKNYFYDNDRSFTRKVKELFVAHRVEKQYNKNEILSFYLNNIYFG
DNQYTLLEGAANHYFGTTVNKNSTTMSHITVQLQSAILASKVNAPSVYNINNMSENFTQRVSTNLEKMKQON
YINETQYQQAMSOL
```

Alignment of *S. aureus* MGT transglycosylase and *S. nepalensis* 0351-containing corisin)

```
SaureusMGT      MK-ILLTILIIIIALFIGIMYFLSTRDNVDELRKIENKSSFVSADNMPEYVKGAFISMEDE 59
Snep0351        MKKTILA--SSLAVALGVTGYAATSDDNNQAHASE-----QNIDK----- 37
**  *:      *: *: : * ** : .          *: :

SaureusMGT      RFYNHHGFDLKGTTTRALFSTISDRDVQGGSTITQQVVKNYFYDNDRSFTRKVKELFVAHR 119
Snep0351        -----AHLAELALNGSAELDQQLHAG----- 59
          : :: :  :*: : : ** :

SaureusMGT      VEKQYNKNEILSFYLNNIYFGDNQYTLLEGAANHY-F----GTT-----V 158
Snep0351        ---AY-----NYNFVLDGNEFIFITSDGNTWSWGYHAAGTQASSSNTTQDVSSEVSV 107
          *          * : . . * : : . * : :          **          *

SaureusMGT      NKNSTTMSHITVQLQSAILASKV-----NAPSVYNINNMSENFTQR-VS 200
Snep0351        NTNEKSASEVRSQQSYATPVTVAAPKASASTNVRTTQTSVAPKAYNVAQTSAASTGGSVK 167
* . * . : * :      **          . *          ** . * . : : *      *      * .

SaureusMGT      TN-----LEKM-----KQQNYINETQYQQ----- 219
Snep0351        AQFLAAGGSEAMWNSIVMPESGPNPNAVNPAGYRGLGQTKESWGTGGSVADQTKGMLNYAK 227
: :          * *          : * : * : * :

SaureusMGT      -----AMSOL----- 224
Snep0351        QRYGSEEAALAFRASHGWW 246
          : *
```

>WP_009225716.1 transglycosylase SLT domain-containing protein [*Paenibacillus* sp. oral taxon 786]

MIRRKVSIKAASLLVALAVFLSTFLAVLPASAASRGAWAPNTSYAVNDTVTYNGSTYTCIQAHTSLVGWE
 PPVVPALWSLSSGGGGGGTTPSDPPTNPPTTVTKPAEVPSPRIWITYVMNADNAYGKGGDFALLLSAVIKK
 ESYFGDGLSGSPSAGDGLMQVEPNTRNAYLSQFSAKYGHAYNHSSEQDQVYMGALILNEKIVRFGNIYNG
 LLHYNGGDWYYPGATDSYGRSILADQYANAVAAGCGRDLEITGKWTIRPDRSLALVPTYFLFINFPLTI
 ERQNRKPD

Paenibacillus transglycosylase SLT domain-containing protein and *S. nepalensis* CNDG 0351

Paenibacillus-transglyc Snep0351	MIRRKVSIKAASLLVALAVFLSTFLAV-----LPASA ---MKKTILASSLAVALGVT--GYAATSDNNQAHASEQNIDKAHLAELALNGSAELDQQP * : * * : * * * * * : * . * .	32 55
Paenibacillus-transglyc Snep0351	ASRGAWAPNTSYAVNDTVTYNGSTYTCIQAHTSLVGWEPPVVPALWSLSSGGGGGGT-T LHAGAYNY-----NFVLDGNEFI FTSDGNTWSWGYHAAGTQASSSNT ** : . . . : * . : . * . : * . . . : *	91 97
Paenibacillus-transglyc Snep0351	P-----SDPPTNPPTTVTKPAEVPSPRIWITYVMNAD TQDVSSEVSVNTNEKSASEVRSQQSYATPVTVAAPKASASTNVRTTQTSVAPKAYNVAQT * : * . * * : * * * : * . : *	121 157
Paenibacillus-transglyc Snep0351	NAYGKGGDFALLLSAVIKKESYFGD---GLSGSPSAGDGLMQVEPNTRNAYLSQFSAKY SAASTGGSVKAQFLAAGGSEAMWNSIVMPESSEGNPNA-----VNPAGY-RGLGQTKESW . * . * * . . : * . * . : . . * * . * * * * * . * . . . :	177 210
Paenibacillus-transglyc Snep0351	GHAYNHSSEQDQVYMGALILNEKIVRFGNIYNGLLHYNGGDWYYPGATDSYGRSILADQY GTG----SVADQT---KGMLNYAKQRYGSE-EAALAFRASHGWW----- * . * * * . : * * * * * : * * . : * . * * :	237 246
Paenibacillus-transglyc Snep0351	ANAVAAGCGRDLEITGKWTIRPDRSLALVPTYFLFINFPLTIERQNRKPD -----	288 246

>pdb|3W6B|A Chain A, Lysozyme-like chitinolytic enzyme

MNHKVVHHHHHIEGRHMGTTTPSDPPTNPPTTVTKPAEVPSPRIWITYVMNADNAYGKGGDFALLLSAVIKKE
 SYFGDGLSGSPSAGDGLMQVEPNTRNAYLSQFSAKYGHAYNHSSEQDQVYMGSLILNEKIVRFGSIYSGL
 LHYNNGDYWYYPGATDSYGRPILADQYANTVYAQYKSYGGGRYSR

Alignment of *Ralstonia* lysozyme-like chitinolytic enzyme and *S. nepalensis* CNDG 0351 protein

pdb 3W6B Snep0351	MN-----HKV 5 MKKTILASSLAVALGVTGYAATSDNNQAHASEQNIDKAHLAELALNGSAELDQQPLHAGA 60 * :
pdb 3W6B Snep0351	HHHHHHIEGR-----H-----MGTTPSD 23 YNYNFVLDGNEFI FTSDGNTWSWGYHAAGTQASSSNTTQDVSSEVSVNTNEKSASEVRSQ 120 : : : : . : * . * . . * . . * :
pdb 3W6B Snep0351	PPTNPPTTVTKPAEVPSPRIWITYVMNADNAYGKGGDFALLLSAVIKKESYF 73 QSYATPVTVAAPKASASTNVRTTQTSVAPKAYNVAQTSAASTGGSVKAQFLAAGGSEAMW 180 * * * : * * * : * : * . * * . . : * . * * : :
pdb 3W6B Snep0351	GD---GLSGSPSAGDGLMQVEPNTRNAYLSQFSAKYGHAYNHSSEQDQVYMGSLILNEK 129 NSIVMPESSEGNPNA-----VNPAGY-RGLGQTKESWGTG----SVADQ---TKGMLNYA 226 . . * * . * * * : * * * . . : * . * * * . : * * . : * * :

```

pdb|3W6B|          IVRFGSIYSGLLHYNGDYPGATDSYGRPILADQYANTVYAQYKSYGGRYSR          183
Snep0351          KQRYGSEEAALAFRASHGWW-----
                  *:**  :.*  .  .  .:**

```

```

>sp|Q5HCY1.1|ISAA_STAAC  RecName:  Full=Probable  transglycosylase  IsaA;
AltName:  Full=Immunodominant staphylococcal antigen A; Flags:  Precursor Gene
locus tag:SACOL2584(Staphylococcus aureus)
MKKTI MASSLAVALGVTGYAAGTGHQAHAEEVNVDAQHLVDLAHNNHQDLNAAPIKDGAYDIHFVKDGFQ
YNFTSNGTTWSWSYEAANGQTAGFSNVAGADYTTSYNQGSNVQSVSYNAQSSNSNVEAVSAPTYHNYSTS
TTSSSVRLSNGNTAGATGSSAAQIMAQRTGVSASTWAAIIARESNGQVNAYNPSGASGLFQTMPGWGPTN
TVDQQINAAVKAYKAQQLGAWGF

```

```

>sp|Q5HDQ9.1|SSAA2_STAAC  RecName:  Full=Staphylococcal secretory antigen ssaA2;
Flags:  Precursor (Gene locus tag: SACOL2291) (Staphylococcus aureus)
MKKIATATIATAGFATIAIASGNQAHASEQDNYGYNPNPDTSSYSYTYTIDAQGNHYHTWKGNNWHPSQLNQ
DNGYYSYYYYNGYNNYNNYNNNGYSYNNYSRYNNYSNNNQSYNNYNNYNSYNTNSYRTGGLGASYSTSSNNV
QVTTTMAPSSNGRSISSGYTSGRNLYTSGQCTYYVFDRVGGKIGSTWGNASNWANAAARAGYTVNNTPKA
GAIMQTTQGAYGHVAYVESVNSNGSVRVSEMNYYGYPGVVTSRTISASQAAGYNFIH

```

```

>sp|Q5HEA4.1|SCED_STAAC  RecName:  Full=Probable  transglycosylase  SceD; Flags:
Precursor (Gene locus tag: SACOL2088) (Staphylococcus aureus)
MKKTL LASSLAVGLGIVAGNAGHEAHASEADLNKASLAQMAQSNQTLNPKPIEAGAYNYTFDYEGFTYH
FESDGFHFAWNYHATGTNGADMSAQAPTTNNVAPSAVQANQVQSQEVEAPQNAQTQQPQASTSNNSQVTA
TPTESKSSEGSSVNVNAHLKQIAQRESGGNIHAVNPTSGAAGKYQFLQSTWDSVAPAKYKGVSPANAPES
VQDAAAVKLYNTGGAGHWVTA

```

S. nepalensis CNDG IsaA containing corisin (Snep0351)

```

MKKTI LASSLAVGLGVTGYAATSDDNQAHAASEQNIDKAHLAELALNGSAELDQQPLHAGAYNYNFVLDGNEFI FTSDGNTWSWGYH
AAGTQASSSNTTQDVSSEVSVNTNEKSASEVRSQQSYATPVTVAAPKASASTNVRTTQTSVAPKAYNVAQTSAASTGGSVKAQFLA
AGGSEAMWNS IMPESSGNPNAVNPAGYRGLGQTKESWGTGSVADQTKGMLNYAKQRYGSEEAALAFRASHGWW

```

Staphylococcus aureus IsaA and S. nepalensis 0351 alignment

```

Snep0351          MKKTI LASSLAVGLGVTGYAATSDDNQAHAASEQNIDKAHLAELALNGSAELDQQPLHAGA          60
Saureus- IsaA    MKKTI MASSLAVALGVTGYAAGT-GHQAHAEEVNVDAQHLVDLAHNNHQDLNAAPIKDGGA          59
                  *****:*****: .:****.* *:***.:** * . :*: *.: **

Snep0351          YNYNFVLDGNEFI FTSDGNTWSWGYHAAGTQASSSNTTQDVSSEVSVNTNEKSASEV-RS          119
Saureus- IsaA    YDIHFVKDGFQYNFTSNGTTWSWSYEAANGQTAGFSNVAGADYTTSYNQGSNVQSVSYNA          119
                  *: .** ** :*: **.***.*.***. *:.. ... .. * * ..: * ..:

Snep0351          QQSYATPVTVAAPKASASTNVRTTQTSVAPKAYNVAQTSAASTGGSVKAQF--LAAGGSE          177
Saureus- IsaA    QSSNSNVEAVSAPTY--HNYSTSTT---SSSVRLSNGNTAGATGSSAAQIMAQRTGVSA          173
                  *.* :. :*:** . * *: * .: .::: .:*.: ** **: :* *

Snep0351          AMWNS IMPESSGNPNAVNPAGYRGLGQTKESWGTGSVADQTKGMLNYAKQRYGSEEAAL          237
Saureus- IsaA    STWAAIIARESNGQVNAYNPSGASGLFQTMPGWGPTNTVDQQINAAVK---AYKA-----          225
                  : * **: ***: ** **:* ** ** .** ...** . * :

Snep0351          AFRASHGWW--          246 (corisin is in blue)

```

Saureus-IsaA ---QGLGAWGF 233
 .**

Staphylococcus aureus SsaA2 and S. nepalensis 0351 alignment

Snep0351	MKKTILASSLAVALGVTGYAA--TSDNNQAHASEQNIDKAHLAELALNGSAELDQQPLHA	58
Saureus-SsaA2	MKKIATA-----TIATAGFATIAIASGNQAHASEQDN-----YGYNPNDP	40
	*** * :...*:*: :..*****: .:* .	
Snep0351	GAYNYNFVLDGNEFI FTSDGNTWSWGYHAAGTQASSS-----NTQDVSSEVSVN	108
Saureus-SsaA2	TSYSYTYTIDAQGNH---YTWKGNWHPSQLNQDNGYYSYYYYNGYNNYNNYNGYSYN	96
	:.*. *...*:*: . ** .:* : :... * :. :. * *	
Snep0351	T-----NEKSASEVRSQQSYATPV---TVAAPKASASTNVRTTQTSVAPKAYNVAQ	156
Saureus-SsaA2	NYSRYNNYSNNNQSYNNYNNYSYNTNSYRTGGLGASYSTSSNNVQV-TTMAPSSNGRSI	155
	. *::: : .. :** * .:* :*:**. *::*. : . :	
Snep0351	TSAAST-----GGSVKAQFLAAG-----GSEAMWNS IVMP -----	186
Saureus-SsaA2	SSGYTSGRNLYTSGQCTYYVDFRVGGKIGSTWGNASNWANAAARAGYTVNNTPKAGAIMQ	215
	:.* :. * .. * . * * . : * . .	
Snep0351	ESSGN ----- PNAVNPAGYR GLGQTKESWGTGSVADQTKGMLNYAKQRYGSEEAALAFR	240
Saureus-SsaA2	TTQGAYGHVAYVESVNSNGSVRSEMNYGPGVVTSRITISAS-----QAA--GYN	264
	:.* :*: * :. : .:* * *::* . : * ..	
Snep0351	ASHGWW 246 (Corisin is highlighted in blue)	
Saureus-SsaA2	FIH-- 267	

Staphylococcus aureus SceD and S. nepalensis 0351 alignment

Snep0351	MKKTILASSLAVALGVTGYAATSDNNQAHASEQNIDKAHLAELALNGSAELDQQPLHAGA	60
Saureus-SceD	MKKTLLASSLAVGLGIVAG---NAGHEAHASEADLNKASLAQMAQSNQDQTLNQKPIEAGA	57
	:**.**:.. . :*:*** :*: **: * .. *::*:***	
Snep0351	YNYNFVLDGNEFI FTSDGNTWSWGYHAAGTQASSNNTQDVSSEVS---VNTNEKSASEV	117
Saureus-SceD	YNYTFDYEGFTYHFESDGTTHFAWNYHATGTNGADMSAQAPTNNVAPSQANQVQSQEV	117
	.* :* : * *** :*::***:.. :. :. :*: *::*: :.***	
Snep0351	RSQQSYATPVTVAAPKASASTNVRTTQTSVAPK-----AYNVAQTSAASTGGSVKA	168
Saureus-SceD	EAPQNA---QTQQPQASTSNNSQVATPTESKSSSEGSSVNVNAHLKQIAQRESGGNIHA	173
	.: * . *::*: * .:* * . * : : * : :*:***:*	
Snep0351	QFLAAG-----SEAMWNS IVMP ESSGNPNAVNPAGYR GLGQTKESWGTGSVADQTKGM	222
Saureus-SceD	VNPTSGAAGKYQFLQSTWDSV-----APAKYKGVSPANAPES---VQD-----	213
	:.*. :*: * : ** **: . : . * *	
Snep0351	LNyakQRYGSEEAALAFRASHGWW--- 246 (corisin is highlighted in blue)	
Saureus-SceD	-----AAAVKLYNTGGAGHWVTA 231	
	: . * .. * *	

S. aureus IsaA and SceD alignment

Saureus-IsaA	MKKTIMASSLAVALGVTGYAAGTGHQAHAEEVNVVDQAHLVDLAHNHQDQLNAAPIKDGAY	60
Saureus-SceD	MKKTLLASSLAVGLGIVA--GNAGHEAHASEADLNKASLAQMAQSNQDQTLNQKPIEAGAY	58
	:**.**:.. . :*:***:***:***:***:***:***:***:***:***	
Saureus-IsaA	DIHFVKDGFQYNFTSNGTTWSWSYEAANGQTAGFSNVAGADY--TTSYNQGSNVQSVSYN	118
Saureus-SceD	NYTFDYEGFTYHFESDGTTHFAWNYHATGTNGADMSAQAPTNNVAPSQANQVQSQEVE	118
	: * :** *:* *:* * :*:*. * : : : * * :.*** . :	

Saureus-IsaA	AQSSNSNVEAVSAPTYHNYSTSTTSSSVRLSNGNTAGATGSSAAQIMAQRTGVSASTWAA	178
Saureus-SceD	APQN-AQTQQPQASTSNNSQVTATPTEKSSE-----GSSV-----NVNAHLKQ	161
	* .. ::: . * * : * . : : * : . : * : * * * . :	
Saureus-IsaA	IIARESNQVNAVNP-S-GASGLFQTM-PGWG-----PTNTV-DQQINA AVKAYK	224
Saureus-SceD	IAQRESGGNIHAVNPTSGAAGKYQFLQSTWDSVAPAKYKGVSPANAPESVQDAAAVKLYN	221
	* * * * . * : : * * * : * * : * * : * : * * : . * * * * * * :	
Saureus-IsaA	AQGLGAWGF- 233	
Saureus-SceD	TGGAGHWVTA 231	
	: * * *	

Query 1-18:

Panel 23b has a mistake on the word “scramble”.

Response

We have corrected the spelling of scramble in the figure, which is now **Supplementary Fig. 19** in the revised manuscript.

Query 1-19:

Datasets for the mass spectrometry approach using HPLC should be publicly available or additional tables should be added in the supplementary material.

Response

As recommended, Datasets for the mass spectrometry approach using HPLC were provided as supplementary material.

Please see Supplementary data set (Mass spectrometry), which was separately uploaded.

Query 2:

The manuscript focus into the alveolar cells response to corisin, without considering the effect that it might induce in matrix-producing fibroblasts and immune cells present and regulated by the same micro-environment.

Response

We have evaluated the effect of corisin on a lung fibroblast cell line, vascular endothelial cell line and T cell line but corisin did not induce apoptosis in any of them.

We presented these data in Supplementary Fig. 13 in the revised manuscript.

Please also see page 17, lines 309 to 311 in the revised text of the manuscript.

Query 3:

Importantly, the manuscript lacks a substantial discussion of the main findings and their correlation with published reports. Indeed, a paper published recently (Tong et. al.,

PMID: 31165050) establishing molecular signatures in BALF samples from IPF patients cannot be ignored.

Response

In the revised version of the manuscript we extensively discussed findings of the present study in relation to previous reports in the literature. As suggested, we also included in the discussion and cited the work of Tong et al (reference No 51).

Please see the discussion section on page 24 to page 32, lines 441 to 586 in the revised manuscript.

Query 4:

The baseline characteristics of patients are included neither in Materials and Methods, nor in Results section, giving weakness to the study design. It would be useful to include information about patients' age, gender, smoking history, diagnose, co-morbidities (if existing) and treatment (if existing). Moreover, a crucial aspect would be to know their ethnical background. Authors should mention if the IPF patients have matched controls.

Response

We have added the **Supplementary Table 3** to show the characteristics of IPF patients enrolled in the present study. Please see the description on **page 33, lines 605 to 614** in the revised manuscript.

We have also new (14) patients with stable disease and with acute exacerbation to compare difference in corisin between them. **Please see Figure 8 in the revised manuscript.**

Query 5:

The genetic TGFB mouse model renders low efficiency to show a fibrotic phenotype, according to results presented in the corresponding Figures. With these preliminary observations, the authors could have used a more robust IPF in vivo model such the bleomycin mouse or an ex vivo model of IPF precision cut lung slices or ex vivo-induced fibrosis by growth factors cocktails (PMID: 31110176).

Response

We have explained (**page 8, lines 153 to 159 in the revised manuscript**) why we used mice overexpressing the human TGF β 1 specifically in the lungs as models of lung fibrosis in the present study as described below:

“.... TGF β 1 is considered the most important mediator of IPF, therefore here we used TG mice with lung fibrosis induced by lung overexpression of human TGF β 1 as previously reported (references 8,10, 11 and 12). Similar to the disease in humans, these TGF β 1 TG mice spontaneously develop pulmonary fibrosis characterized by a predominant and progressive scarring process, fatal outcome and typical lung

histopathological findings (diffuse collagen deposition, honeycomb cysts, fibroblast foci-like areas) (reference 8 and 11).

Query 6:

Some aspects of the final model in Extended data Fig 24 generated with the description of results are missing credibility. There is no experimental evidence for the gene transfer from Staphylococcus to Streptococcus. The authors do not explore a single aspect of myofibroblast or endothelial activation, among other general assumptions. There is a mistake on the word “endothelial”.

Response

We have corrected the figure, which is now **Supplementary Fig. 25** in the revised manuscript. The drawing for gene transfer was deleted, and myofibroblast was replaced by fibroblast. We have kept fibroblast and endothelial cells because we have presented data on both cells (fibroblast and endothelial cells, **Supplementary Fig. 13**) in the present revised form of the manuscript.

We have also corrected the error in the word “endothelial”.

The lateral gene transfer hypothesis is based on genomic analysis (please, see explanation for Referee 1 (Response for Query 14 of Referee 1), and the revised text under the results section on pages 22, 23 and 24, lines 404 to 440.

Query 7:

The observations of a salty environment in an IPF animal model are not fully discussed in the context of other pulmonary diseases. Even if cystic fibrosis is known to display the same altered electrolyte course, for lung cancer it will be an innovative finding. The manuscript includes a numerous amount of panels with A549 cells, a model of lung adenocarcinoma. However, these cells are not representative of injured ATII cells during IPF. Rather they will display some metabolic effects (Warburg effect) inside a saline micro-environment. A good justification of selection of this cellular model is required or, in a better case, to substitute those panels exclusively with the healthy bronchial epithelial cell line, which shows consistent results with A549.

Response

As recommended, we have discussed in a paragraph the salty environment in the IPF animal model. The TGFβ1 mice with fibrosis often progress to show symptoms of lung cancer, and thus in the future this model may become useful to investigate the effect of the altered electrolyte on lung cancer. We think this is a very notable suggestion from the reviewer.

Please see pages 25 and 26, lines 452 to 478 in the discussion section of the revised manuscript.

As suggested, we have also presented a justification for the use of A549 in most experiments.

Please see the revised manuscript on page 42, lines 761 to 765 in the revised manuscript.

Query 8:

The authors obviated or ignored previous contributions from other groups where the specific bacterial strains for Streptococcus and Staphylococcus are included (PMID 30824326, 29486761, 28802277, 28157391). For example, in IPF Staphylococcus epidermidis and aureus, Streptococcus agalactiae, gallolyticus, pneumoniae and gordonii have been reported in IPF patients. Intriguingly, this is the first report with Staphylococcus nepalensis and that makes even more relevant to include and discuss the ethnical background of the selected IPF and control patients. Their introduction statement that “specific bacterial strain...remain unknown” is only true for reference 7, where the authors use operational taxonomic units (OTU) rather than concrete strains.

Response

As recommended, we have discussed and cited previous studies showing the participation of other specific bacteria including PMID 30824326 (Reference No 52), PMID29486761 (Reference No 50), and PMID28802277 (Reference No 48), PMID28157391 (Reference No 42) suggested by the Reviewer.

Please see pages 27 and 28, lines 496 to 503 in the revised manuscript.

We presented in **Supplementary Table 3**, the ethnical background of the patients.

The statement “specific bacterial strain...remain unknown” was removed in the present revised version of the manuscript.

Response to Queries of Reviewer 3

We thank the reviewer for his/her time in reviewing our manuscript and appreciate the comments, which were very constructive and have helped to improve the interpretation and the quality of our manuscript.

Query 1

This was a very difficult manuscript to read. My apologies if the authors have followed journal instructions. However, as a reasonably experienced reviewer, I expect that the rationale of a study will be set up in an introduction, to be followed by a methods section and then successively by a results section and then a discussion. In this manuscript, the introduction, results and discussion have been merged and interspersed at the start of the manuscript followed by a methods section. I do not understand why the authors did not produce a traditional introduction closing with specific study aims to provide

structure to the presentation, even if they have followed journal instructions in merging results and discussion.

Response

We have changed the format of the manuscript to fit the format of an Article, as suggested by Reviewer.

Query 2

All this said, the key question is whether the conclusions are robust. Without the usual manuscript structure, I had to resort to dissecting the abstract, as follows:

“Idiopathic pulmonary fibrosis (IPF) is a chronic, progressive and fatal disease of unknown etiology. Injury, followed by apoptosis of lung epithelial cells, and increased accumulation of collagen-secreting myofibroblasts play a critical role in the pathogenesis of this intractable disease.”

This is generally sound although the authors have chosen to highlight only selective pathogenic considerations.

“Clinical progression of IPF is associated with increased abundance of the bacterial genera *Staphylococcus* and *Streptococcus* in the lung, although the specific bacterial strain and the mediating factor remain unknown.”

However, it is not known that this linkage is causative. It may well be that a milieu of activated pathways driving progression also provides a predilection for growth of particular organisms. There is a danger of conflating cause and effect.

Response

Following the constructive comments of the Reviewer, we have made changes in the title, in the abstract and in the text to clarify that we have evaluated whether the pro-apoptotic peptide causes acute exacerbation of the lung fibrotic disease. For the same reason, we have also deleted the word “**progression**” to make much clearer that the purpose of the study was to evaluate **acute exacerbation of the disease** by the pro-apoptotic peptide.

Please see the title (page 1) and the Abstract (pages 5 and 6) in the revised manuscript.

Query 3

“Here, we report that *Staphylococcus nepalensis* strain CNDG, isolated from lung fibrotic tissue, releases a unique peptide, we named corisin, buried in a polypeptide, to induce apoptosis of lung epithelial cells and therefore accelerating progression of pulmonary fibrosis.”

The idea is interesting but the statement implies that the authors have observed a specific linkage between corisin and disease progression as opposed to a non-specific short term irritation due to flooding the lungs with an alien polypeptide.

Response

To determine whether the effect is specific to corisin or a side irritant effect, we performed additional experiments in which we used a scrambled peptide as negative control and the results showed that the scrambled peptide does not exacerbate the lung fibrotic disease, suggesting that the effect is specific to corisin.

The results are described in Figure 6 (from panel a to panel g) and on pages 18 and 19, lines 333 to 343 in the revised text of the manuscript.

In addition, we tested a variant (IVMPESGGNPNAVNPAGYR) of corisin (with a single amino acid change) present in *Streptococcus pneumoniae* strain N and show that this peptide also induces apoptosis, as observed for corisin. Thus, it is clear that corisin and its closely related sequences induce apoptosis, and that their effect is inherent to the peptide sequence.

The results are described in Supplementary Fig. 24a,b and on page 24, lines 431 to 440 in the revised text of the manuscript.

Query 4

“The pro apoptotic peptide was significantly increased in the lungs from IPF patients compared to healthy controls.”

Does this observation actually help? Epiphenomena of fibrosis and, separately, epiphenomena of fibrosis progression should be present in disease and should not be present in healthy controls. The key comparison, surely, is between corisin content and disease progression in IPF patients. Without this comparison, the relevance to IPF progression is wholly uncertain.

“Evolutionary analyses revealed that the polypeptide embedded with the apoptosis-inducing peptide is conserved in diverse staphylococci, with known and unknown pathogenicity, with pathogenic strains of *Streptococcus pneumoniae* and *Mycobacterium abscessus* acquiring the gene likely through acquisition of genetic material in the lung.”

This explores pathways for polypeptide conservation but does not establish linkage to disease progression.

Response

We compared the level of corisin in the lungs between healthy subjects and IPF with stable disease to clarify whether the level of corisin is higher under pathological conditions than under healthy states

Please see Figure 8c,d and page 20, paragraph 1, lines 5 to 14 in the revised version of the manuscript.

In addition, we evaluated the corisin content between IPF patients with and without acute exacerbation of the disease.

The results showed that the concentration of corisin is significantly increased in IPF patients with acute exacerbation compared to patients without exacerbation.

Please see Figure 8c,d and page 20, lines 367 to 376 in the revised version of the manuscript.

Query 5

The key observation is the paragraph that follows, again merging introduction, results and discussion

“To investigate whether corisin can exacerbate the lung fibrotic disease in vivo, we separated TGFβ1 TG mice into two groups with matched level of lung fibrosis (Extended data Fig. 17a,b) and treated them with saline or corisin by intra-tracheal route once daily 12 for two days before euthanasia on day 3. TGFβ1 TG mice receiving corisin showed significantly increased infiltration of neutrophils, collagen deposition, concentration of inflammatory cytokines and chemokines, and apoptosis of epithelial cells in the lungs compared to control mice, indicating the detrimental effect of the pro-apoptotic activity of corisin in vivo (Fig. 3a,b,c,d,e,f,g). We then explored the presence of corisin in mice and human samples. We found significantly enhanced level of corisin in TGFβ1 TG mice compared to WT mice (Extended data Fig 18a,b), and significantly increased concentration of corisin in the bronchoalveolar lavage fluids from IPF patients compared to healthy controls, suggesting the potential implication of corisin in IPF (Figure 4). A dramatic increase of apoptotic epithelial cells occurs in the lung of IPF patients with acute exacerbation,17,18 and our results suggest that excessive release of the bacterial-derived pro-apoptotic corisin will contribute to this fatal disease complication.”

The authors have shown that instillation of a polypeptide, as opposed to saline, causes a short term injurious effect but the relevance of this to human disease progression is unclear. The model used by the authors – the acute introduction of a polypeptide by intratracheal installation – does not simulate chronic colonisation in human disease and nor is it clear that what may be a highly concentrated irritant effect captures the level of corisin action in human disease. I suggest that this experiment would have had more plausibility had corisin been compared to selected control polypeptides, to establish a corisin-specific effect, and had corisin concentrations in BAL been compared between progressive and non-progressive IPF patients. It is not clear how the comparison with normal controls helps.

Response

The Reviewer is right in pointing out that in the current study we have not evaluated the effect of corisin on the “chronic progression” of pulmonary fibrosis. As also explained above, we have clarified in the title, in the Abstract and throughout the text of the revised version of the manuscript that, in this study, we have only evaluated whether

corisin can induce **acute exacerbation** of the lung fibrotic disease. We apologize for the lack of clarity in the original version of the manuscript.

Acute exacerbation is a common fatal complication that develops suddenly in patients with IPF for unknown etiology in most cases and that is characterized by a rapid and dramatic deterioration of the lung fibrotic disease.

In the present study, we found that intratracheal (intrapulmonary) instillation of corisin for two consecutive days induces this acute exacerbation of the lung fibrotic disease in our TGF β 1 overexpression-associated lung fibrosis mouse model.

Following the suggestion of the Reviewer, we also instilled intratracheally control **scrambled peptide** to the same mice but the control peptide did not induce acute exacerbation, suggesting that the effect is specific to corisin.

The results are described in Figure 6. Please also see the description of the results on pages 18 and 19, lines 333 to 343 in the revised manuscript.

In addition, to corroborate these findings, for the revised version of the manuscript, we performed an additional *in vivo* experiment in which we compared the acute effect of instilling intratracheally into mice with lung fibrosis, bacteria containing the pro-apoptotic corisin (*Staphylococcus nepalensis* strain CNDG) and bacteria lacking the corisin sequence (*Staphylococcus epidermidis* ATCC14990). We found that only the bacterium containing the corisin sequence induces acute exacerbation of the lung fibrotic disease, further suggesting the role of corisin in the development of this acute complication of lung fibrosis.

These results are described in Figure 7 and in Supplementary Fig. 20. Please also see pages 19 and 20, lines 344 to 360 in the revised manuscript

Also as described above, we compared the corisin level between IPF patients with and without acute exacerbation. Patients with acute exacerbation showed significantly increased level of corisin compared to patients without exacerbation. The results are described in **Figure 8**. Based on these observations, we believe that corisin is involved in the process of acute exacerbation in patients with IPF.

REVIEWERS' COMMENTS:

Reviewer #1 (Remarks to the Author):

the reviewers have addressed my concerns.

Reviewer #2 (Remarks to the Author):

Peer-review of the revised manuscript with the number NCOMMS-19-17899-T at Nature Communications by D'Alessandro-Gabazza et al and with the title: "A pro-apoptotic peptide conserved in diverse staphylococci induces acute exacerbation of pulmonary fibrosis"

The authors addressed and/or discussed almost all the concerns that I have raised in my previous peer-review. In addition, the authors also addressed most of the concerns of the other two Reviewers. The work became more intuitive to follow and accurate to interpret after the changes done to the structure of the manuscript. For example, in the current version of the manuscript, it is evident that the increase of Staphylococcus and Streptococcus in IPF lungs contributes to the exacerbation of the disease, rather than to the progression. Further, the authors added a comprehensive analysis of the changes in the immune cells yields in their animal model, as well as a correlation between the expression of sodium channels with conventional fibrotic factors. The new version of the manuscript confirms my original, positive opinion on the manuscript. Nevertheless, I would like to suggest addressing the following minor concerns:

1. The explanation on the validation of the phenotype on their TGFB TG mice w/ and w/o fibrosis (Rebuttal letter, pages 21-22) is worth to include in the Material and methods section. Otherwise, the rationale for selecting specific animals as negative controls is not obvious.
2. Of great interest for the reproducibility of the results is the inclusion of raw HPLC data in the new version of the Supplementary data set.
3. The clinical information of the selected patients and the contributions of previous groups should be accessible to the readers.

4. The labelling of the samples has to be uniform on each panel. For instance, Fig1B and SupFig2a-d (right), contain the same samples and the graphs are not matching in terms of labelling and colors.

The inclusion of an ex vivo validation with human tissue would have been an elegant contribution to the manuscript. However, I accept the explanation provided by the authors, despite a nice example of how to achieve this aim was provided in PMID: 31110176. In any case, the data presented in the manuscript will be a major contribution for the field of IPF. In addition, the manuscript has a strong translational potential suggesting therapeutic approaches against IPF. I strongly believe that after addressing the minor concerns described above, the manuscript will achieve the standards for publication at Nature Communications.

Reviewer #3 (Remarks to the Author):

The authors have done well to restructure their manuscript in a much clearer presentation. I still suggest that they refer always to "acute exacerbation" rather than variably to "acute exacerbation" and "exacerbation" as the latter term includes both acute exacerbation and more chronic progression

In the title, suggest "Pro-apoptotic peptide in lung fibrosis acute exacerbation."

This apart, I was impressed by the definitive responses and further experiments, all convincing, prompted by reviewer enquiries. I have no further suggestions.

RESPONSE TO REVIEWERS' COMMENTS:

Response to Reviewer #1 :

Comment 1

the reviewers have addressed my concerns.

Reply

We are very thankful for the constructive comments of the Reviewer that have substantially improved the manuscript.

Response Reviewer #2 :

Comment 1

Peer-review of the revised manuscript with the number NCOMMS-19-17899-T at Nature Communications by D'Alessandro-Gabazza et al and with the title: "A pro-apoptotic peptide conserved in diverse staphylococci induces acute exacerbation of pulmonary fibrosis"

The authors addressed and/or discussed almost all the concerns that I have raised in my previous peer-review. In addition, the authors also addressed most of the concerns of the other two Reviewers. The work became more intuitive to follow and accurate to interpret after the changes done to the structure of the manuscript. For example, in the current version of the manuscript, it is evident that the increase of Staphylococcus and Streptococcus in IPF lungs contributes to the exacerbation of the disease, rather than to the progression. Further, the authors added a comprehensive analysis of the changes in the immune cells yields in their animal model, as well as a correlation between the expression of sodium channels with conventional fibrotic factors. The new version of the manuscript confirms my original, positive opinion on the manuscript.

Reply

We are very thankful for the constructive comments of the Reviewer that have substantially improved the manuscript.

Comment 2

Nevertheless, I would like to suggest addressing the following minor concerns:

The explanation on the validation of the phenotype on their TGFB TG mice w/ and w/o fibrosis (Rebuttal letter, pages 21-22) is worth to include in the Material and methods section. Otherwise, the rationale for selecting specific animals as

negative controls is not obvious.

Reply

As recommended by the Reviewer we have added the explanation on the validation of the phenotype on the TGF β 1 TG mice with and without lung fibrosis. Please see pages 34 (lines 618 to 622) in the revised MS (validated merged manuscript pdf file).

Comment 3

Of great interest for the reproducibility of the results is the inclusion of raw HPLC data in the new version of the Supplementary data set.

Reply

We have added the raw chromatogram of the HPLC data as suggested. It included in the Supplementary information and mentioned in the main text. Please see page 15 (line 268) in the revised MS (validated merged manuscript pdf file).

Comment 4

The clinical information of the selected patients and the contributions of previous groups should be accessible to the readers.

Reply

The clinical information of the selected patients are available in Supplementary Table 1, and also in the Source Data file.

Comment 5

The labelling of the samples has to be uniform on each panel. For instance, Fig1B and SupFig2a-d (right), contain the same samples and the graphs are not matching in terms of labelling and colors.

Reply

Labeling of in each panel of the figures were made uniform.

Comment 6

The inclusion of an ex vivo validation with human tissue would have been an elegant contribution to the manuscript. However, I accept the explanation provided by the authors, despite a nice example of how to achieve this aim was provided in PMID: 31110176. In any case, the data presented in the manuscript will be a major contribution for the field of IPF. In addition, the manuscript has a

strong translational potential suggesting therapeutic approaches against IPF. I strongly believe that after addressing the minor concerns described above, the manuscript will achieve the standards for publication at Nature Communications.

Reply

We are very thankful for the constructive comments of the Reviewer that have substantially improved the quality of the manuscript..

Response to Reviewer #3:

Comment 1

The authors have done well to restructure their manuscript in a much clearer presentation. I still suggest that they refer always to "acute exacerbation" rather than variably to "acute exacerbation" and "exacerbation" as the latter term includes both acute exacerbation and more chronic progression

Reply

As suggested, we changed "exacerbation" to "acute exacerbation" in the new revised version of the manuscript.

Comment 2

In the title, suggest "Pro-apoptotic peptide in lung fibrosis acute exacerbation."

Reply

We have changed the title of the manuscript. Please see page 1 of the MS(validated merged manuscript pdf file).

Comment 3

This apart, I was impressed by the definitive responses and further experiments, all convincing, prompted by reviewer enquiries. I have no further suggestions.

Reply

We are very thankful for the constructive comments of the Reviewer that have substantially improved the quality of the manuscript.